

# The Value of Citizen Science for Flood Risk Reduction: Cost-benefit Analysis of a Citizen Observatory in the Brenta-Bacchiglione Catchment

Michele Ferri[1], Uta Wehn[2], Linda See[3], Steffen Fritz[3]

[1]Alto-Adriatico Water Authority (AAWA), Cannaregio 4314, 30121 Venice, Italy
[2]IHE Delft Institute for Water Education, Westvest 7, 2611 AX Delft, The Netherlands
[3]International Institute for Applied Systems Analysis (IIASA), Schlossplatz 1, 2361 Laxenburg, Austria

*Correspondence to*: Michele Ferri (michele.ferri@distrettoalpiorientali.it)

**Abstract.** Citizen observatories are a relatively recent form of citizen science, which involve citizens in making
environmental observations over a period of time. These observations can help to inform the decision making of local authorities and other stakeholders, creating a platform for two-way interaction between citizens and public agencies. Although citizen observatories can clearly generate many different benefits, they also have an associated cost. There are currently no examples of quantifying the costs and benefits of citizen observatories in the literature, yet this type of analysis is critical if there is to be real uptake of citizen observatories by public agencies more generally. This paper presents and
applies a generic methodology for capturing the value of a citizen observatory for flood risk reduction in the Brenta-Bacchiglione catchment using a cost-benefit analysis. The results show that the benefits of implementing a citizen observatory approach outweigh the costs by approximately 2 to 1 and can reduce the annual expected damage to a greater degree than a much more costly structural approach.

## 1 Introduction

For the past three years, extreme weather events (including flooding) are the top risk in terms of likelihood and among the top three risks in terms of impact, where this combination makes it the top risk in 2019 (WEF, 2017, 2018, 2019). Between 1995-2015, flooding alone accounted for 47% of all weather-related disasters, affecting 2.3 billion people globally (CRED and UNISDR, 2015). Continuing an upward trend, financial losses in 2017 due to global weather-related disasters exceeded US$300 billion (Swiss Re, 2017). Hurricane Harvey, in particular, caused US$125 billion damage in 2017, led to the death
of 88 people, destroyed more than 12,700 homes and resulted in a rise in gas prices due to the impacts on oil production (Amadeo, 2019). However, economic losses can go well beyond damage to infrastructure and assets, e.g., disruption to businesses and supply chains can equal or exceed the costs of infrastructure damage (Hallegatte, 2008; Jongman, 2018). Moreover, developing countries and small island states affected by tropical storms are likely to suffer greater losses. In 2017,





Hurricane Maria caused an estimated total damage and loss of US$1.3 billion to Dominica while US$5.4 billion in damage
and loss was estimated for other islands due to the combined effects of Hurricanes Maria and Irma (Asariotis, 2018).

Accurate predictions are crucial for flood risk management (FRM), e.g., to control river structures and water levels, in
order to reduce risks and damages from flooding, particularly in densely populated urban areas (Mazzoleni et al., 2017b).
However, weather patterns are local in nature, not easily captured or predicted by existing in-situ and remote sensing-based
modelling approaches, and are likely to be intensified by climate change (Pachauri et al., 2014; Tol, 2014). The data acquired
using these methods are often incomplete in terms of resolution and density (Lanfranchi et al., 2014). This translates into
variable accuracy in flood predictions (Werner et al., 2005).

The recent exponential growth in citizen science and crowdsourcing approaches, accelerated by the rapid diffusion of
information and communication technologies, is providing additional, complementary sources of data for hydrological and
hydraulic models. Citizen science refers to the involvement of the public in any step of the scientific method (Shirk et al.,
2012). Among the various forms of citizen science (Cooper et al., 2007; Bonney et al., 2009; Shirk et al., 2012), contributory
forms are of particular interest here, focusing on the observations that citizens can contribute (as opposed to their
collaboration in the entire research process or the co-design of the research). Citizen observatories (CO) are a particular form
of citizen science in so far as they involve citizens in environmental observations over an extended period of time (rather
than one-off exercises such as data collection 'Blitzes'), and hence contribute to improved temporal resolution of the data,
using dedicated apps, easy-to-use physical sensors and other monitoring technologies linked to a dedicated platform (Liu et
al., 2014; Mazumdar et al., 2016). COs must also include a public authority (e.g., a local, regional or national body) to enable
two-way communication between citizens and the authorities to create a new source of high quality, authoritative data for
decision making and for the benefit of society. This approach is increasingly being used in hydrology/water sciences and
management and in various stages of the FRM cycle, as reviewed and reported by e.g., Assumpção (2018), Etter et al.
(2018), Mazzoleni et al. (2017a), Buytaert et al. (2014), Wehn and Evers (2015) and Wehn et al. (2015).

The promising potential of the contribution of COs to improved FRM is paralleled by limited evidence of their actual
impacts and added value. Efforts are ongoing such as the consolidation of evaluation methods and empirical evidence by the
H2020 project WeObserve[1] Community of Practice on the value and impact of citizen science and COs and the development,
and the application of methods for measuring the impacts of citizen science by the H2020 project MICS[2]. However, the lack
of available, appropriate and peer-reviewed evaluation methods and of evidence of the added value of COs is holding back
the uptake and adoption of COs by policy makers and practitioners. The aim of this paper is to fill this gap by presenting and
applying a generic methodology for capturing the value of COs by means of a tailored, detailed cost-benefit analysis (CBA),
the COCBA. The proposed methodology is applied using primary empirical evidence from a CO pilot that was undertaken
by the WeSenseIt project in the town of Vicenza, Italy, and now extended to the wider catchment.


---

[1] https://www.weobserve.eu/
[2] https://mics.tools/





The paper is structured as follows. Section 2 presents the conceptual details of the COCBA as well as information about the Brenta-Bacchiglione catchment and the WeSenseIt CO pilot while section 3 presents the results from the analysis. Conclusions and limitations of the methodology as well as case-specific insights are provided in section 4.

## 2 Methodology

Starting with a description of the input data used (section 2.1), the proposed methodology is presented in terms of the calculation of risk (section 2.2) and the steps involved in evaluating the costs and benefits of COs for FRM (section 2.3). This is followed by the justification of the selected case study to which this methodology has been applied, i.e., the WeSenseIt CO in the Brenta-Bacchiglione catchment, together with information about the case study (section 2.4).

### 2.1 Input data

There are four main data sets used in the COCBA methodology. The first is Corine Land Cover 2006[3] produced by the European Environment Agency (Steemans, 2008). The second is the population of the catchment, which was obtained from ISTAT 2001[4]. The third data set is the pollutants affecting the basin[5] while the protected areas and cultural heritage is the final data set, obtained from the Italian Ministry of Property and Cultural Activities[6].

### 2.2 Calculation of risk

In this context, risk is the probability that a damaging event will occur from a natural phenomenon or due to human activities that can cause harmful effects to the surrounding population, assets and/or infrastructure, within a particular area and over a given period of time. Specifically, *Risk* is calculated as the combination of three components (Cutter, 1996):

*Risk = Hazard * Vulnerability * Exposure*                                                                                                    (1)


where *Hazard* is the probability that a phenomenon of a certain intensity will occur in a certain period of time in a given area; *Vulnerability* is the degree to which different elements (i.e., people, buildings, infrastructure, economic activities, etc.) will suffer damage as a consequence of the stresses induced by an event of a certain intensity; and *Exposure* is the number of units (or the "value") of each of these elements at risk present in a given area, such as human lives or assets. The potential

damage can then be calculated as the combination of the value of the exposed elements with the value of these elements with respect to an event of given intensity. If the impact of floods is assessed at a mesoscale, risk can be quantified in relative terms, i.e., a value between 0 and 1, where 0 represents the absence of risk and 1 is the maximum risk. Figure 1 depicts the

---

[3] http://www.centrointerregionale-gis.it/script/corinedownload.asp
[4] http://www.istat.it/it/archivio/44523
[5] https://prtr.eea.europa.eu/#/home
[6] http://vincoliinrete.beniculturali.it





different steps in calculating risk for the purpose of undertaking an integrated flood risk assessment. Each of the components of risk are described in more detail in the sections that follow.

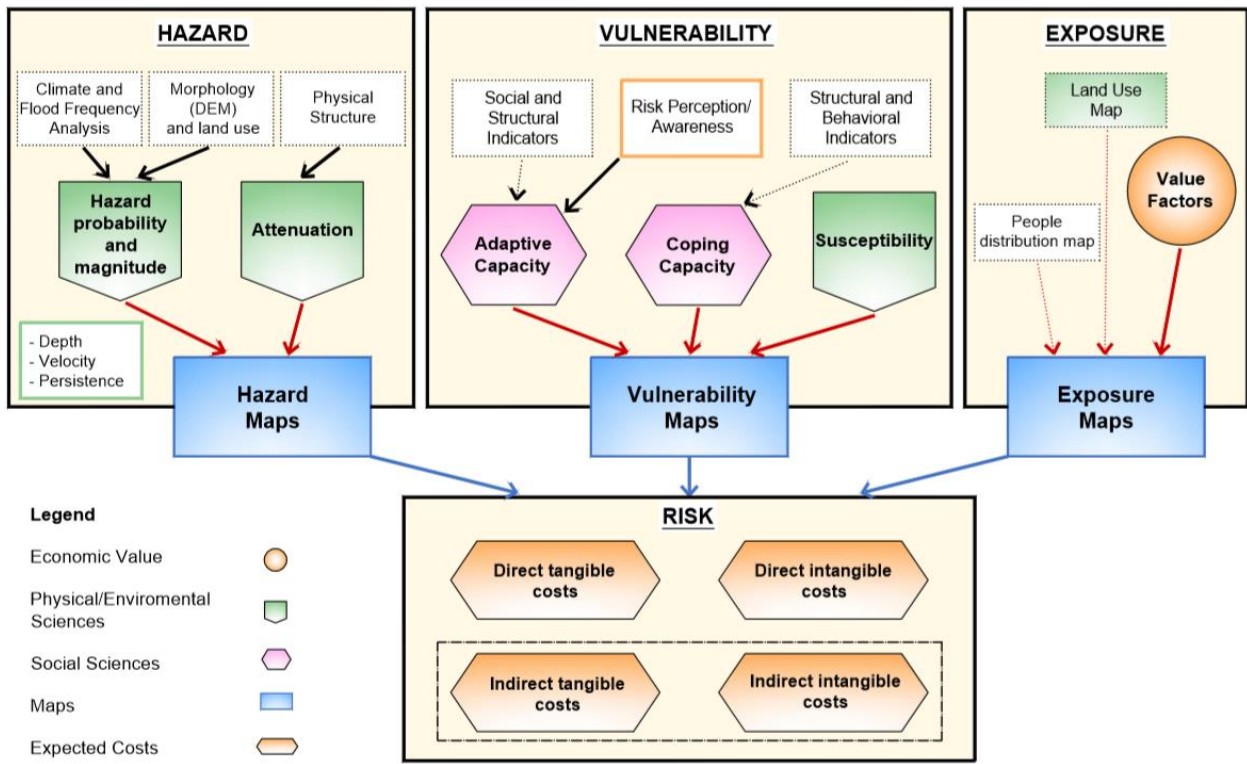

**Figure 1: Flowchart outlining the determination of risk in a flood risk assessment context.**

The exposed elements must be expressed in terms of the following macro-categories, which are set out in the EU 2007/60/CE Flood Directive (EU, 2007); these include: the population affected (art.6-5.a); the types of economic activities affected (art.6-5.b); and the environmental and cultural-archaeological assets affected (art.6.5.c). These three macro-categories can be characterized by the land use classes shown in Table 1, which are taken from the Corine Land Cover map 2006. The next three sections describe how the individual components of risk (Figure 1) are calculated in the context of this case study.

### 2.2.1 Hazard

According to Article 6 of the 2007/60/CE Flood Directive (EU, 2007), three hazard scenarios must be addressed, which can be calculated using a hydrological and hydraulic model:

1. A flood with a low probability, which is 300-year return period in this study area;
2. A flood with a medium probability, which is a 100-year return period; and





3. A flood with a high probability, which is a 30-year return period in the study area.


**Table 1: List of the land use classes used to characterize the three macro-categories from the EU 2007/60/CE Flood Directive.**

| ID | Description |
|---|---|
| 1 | Residential |
| 2 | Hospital facilities, health care, social assistance |
| 3 | Buildings for public services |
| 4 | Commercial and artisan |
| 5 | Industrial |
| 6 | Specialized agricultural |
| 7 | Woods, meadows, pastures, cemeteries, urban parks, hobby agriculture |
| 8 | Tourist-Recreation |
| 9 | Unproductive |
| 10 | Ski areas, Golf course, Horse riding |
| 11 | Campsites |
| 12 | Communication and transportation networks: roads of primary importance |
| 13 | Communication and transportation networks: roads of secondary importance |
| 14 | Railway area |
| 15 | Area for tourist facilities, Zone for collective equipment (supra-municipal, subsoil) |
| 16 | Technological and service networks |
| 17 | Facilities supporting communication/transportation networks (airports, ports, service areas, parking lots) |
| 18 | Area for energy production |
| 19 | Landfills, Waste treatment plants, Mining areas, Purifiers |
| 20 | Areas on which plants are installed as per Annex I of Legislative Decree 18 February 2005, n. 59 |
| 21 | Areas of historical, cultural and archaeological importance; cultural heritage |
| 22 | Environmental goods |
| 23 | Military zone |

The hazard associated with these scenarios was calculated in relative terms as a value between 0 and 1. A two-dimensional hydraulic model was used to generate the water levels and the water speeds at a resolution of 10 m (Ferri et al., 2010) for

these hazard scenarios. These model outputs were also used to calculate the vulnerability in this area (see section 2.2.3).

**2.2.2 Exposure**

Exposure is calculated for each of the macro-categories in the EU Flood Directive, i.e., based on the people, economic activities and environmental/cultural-archaeological assets affected, as described in more detail below.

**(i) People affected**

The exposure of the population is a function of two factors. The first is the number of people living in an area expressed by a four-class density factor ($F_d$) as outlined in Table 2. The second is the duration factor ($F_t$), which is calculated as the proportion of time spent in certain locations (e.g., houses, schools, etc. - see the land use types listed in Table 1) over a 24 hour day (Provincia Autonoma di Trento, 2006). The exposure of the population ($E_P$) is then calculated as:





$E_p = F_d * F_t$                                                                                  (2)

**Table 2: A factor characterizing the density of people ($F_d$) in relation to the number of people present.**

| Number of people | $F_d$ |
|:---:|:---:|
| 1 ÷ 50 | 0.90 |
| 51 ÷ 100 | 0.95 |
| 101 ÷ 500 | 0.98 |
| > 500 | 1 |

**(ii)   Economic activities affected**

The spatial distribution and types of economic activities in flood risk areas must be determined in order to assess the potential negative impacts from flooding. The exposure or impact on economic activities ($E_E$) is calculated from the restoration costs, and the costs resulting from losses in production and services. These are calculated for each of the land use categories provided in Table 1.

**(iii)   Environmental and cultural heritage assets affected**

The exposure of assets in the environmental and cultural heritage category ($E_{ECH}$) is calculated by land use type (Table 1), by considering the degree of potential damage caused by an adverse flood event (Provincia Autonoma di Trento, 2006). The relative values of exposure for each of the three macro-categories ($E_P$, $E_E$ and $E_{ECH}$) are provided in Table 3, listed by land use type.

**2.2.3 Vulnerability**

Vulnerability results from the interaction between physical-environmental and social components. To define vulnerability from a physical point of view, we use the concept of the susceptibility of an exposed element such as people or buildings, as outlined above (Balbi et al., 2012). Susceptibility is related to the context in which the event occurs and refers to a quantitative (or qualitative) assessment of the event type, the causal factors and the characteristics of the event. Social

vulnerability refers to the perception or awareness that an adverse event may occur. Greater awareness tends to correspond to greater preparation if an event takes place. Social vulnerability can be divided into:

- Adaptive Capacity: the combination of strengths, attributes and resources available to an individual, community, society or organization (ex-ante hazard) that can be used to prepare and/or implement actions aimed at reducing impacts or exploiting beneficial opportunities (IPCC, 2012; Torresan et al., 2012).






**Table 3: The relative values of exposure for people, economic activities and environmental/cultural assets by land use type.**

| ID | Description | $E_P$ | $E_E$ | $E_{ECH}$ |
|---|---|---|---|---|
| 1 | Residential | 1 | 1 | 1 |
| 2 | Hospital facilities, health care, social assistance | 1 | 1 | 1 |
| 3 | Buildings for public services | 1 | 1 | 1 |
| 4 | Commercial and artisan | 0.5 ÷ 1 | 1 | 0.8 |
| 5 | Industrial | 0.5 ÷ 1 | 1 | 0.3 ÷ 1 |
| 6 | Specialized agricultural | 0.1 ÷ 0.5 | 0.3 ÷ 1 | 0.7 |
| 7 | Woods, meadows, pastures, cemeteries, urban parks | 0.1 ÷ 0.5 | 0.3 | 0.7 |
| 8 | Tourist recreation | 0.4 ÷ 0.5 | 0.5 | 0.1 |
| 9 | Unproductive | 0.1 | 0.1 | 0.3 |
| 10 | Ski areas, Golf course, Horse riding | 0.3 ÷ 0.5 | 0.3 ÷ 1 | 0.3 |
| 11 | Campsites | 1 | 0.5 | 0.1 |
| 12 | Roads of primary importance | 0.5 | 1 | 0.2 |
| 13 | Roads of secondary importance | 0.5 | 0.5 ÷ 1 | 0.1 |
| 14 | Railway area | 0.7 ÷ 1 | 1 | 0.7 |
| 15 | Area for tourist facilities, Zone for collective equipment (supra-municipal, subsoil) | 1 | 0.3 | 0.3 |
| 16 | Technological and service networks | 0.3 ÷ 0.5 | 1 | 0.1 |
| 17 | Facilities supporting communication and transportation networks (airports, ports, service areas, parking lots) | 0.7 ÷ 1 | 1 | 1 |
| 18 | Area for energy production | 0.4 | 1 | 1 |
| 19 | Landfill, Waste treatment plants, Mining areas, Purifiers | 0.3 | 0.5 | 1 |
| 20 | Areas on which plants are installed as per Annex I of Legislative Decree 18 February 2005, n. 59 | 0.9 | 1 | 1 |
| 21 | Areas of historical, cultural and archaeological importance | 0.5 ÷ 1 | 1 | 1 |
| 22 | Environmental goods | 0.5 ÷ 1 | 1 | 1 |
| 23 | Military zone | 0.1 ÷ 1 | 0.1 ÷ 1 | 0.1 ÷ 1 |





- Coping Capacity (or ex-post adaptation capacity): the ability of people, organizations and systems to cope with
adverse conditions using available skills, resources and opportunities (IPCC, 2012; Torresan et al., 2012).

Vulnerability is quantified for each of the three macro-categories (i.e., people, economic activities and environmental/cultural-archaeological assets affected) as outlined below.

### (i) People affected

To characterize the vulnerability associated with human presence, we refer to flow velocity ($v$) and water depth ($h$) values that produce "instability" with respect to remaining in an upright position. Many authors have dealt with the instability of people in flowing water (see e.g., Chanson and Brown, 2018), and critical values derived from the product of $h$ and $v$ have been proposed. For example, Ramsbottom et al. (2004) and Penning-Rowsell et al. (2005) have proposed a semi-quantitative equation that links a flood hazard index, referred to as the Flood Hazard Rating (FHR), to $h$, $v$ and a factor

related to the amount of transported debris, i.e., the Debris Factor ($DF$), as follows:

$$FHR = h * (v + 0.5) + DF \qquad\qquad (3)$$

The values of $DF$ related to different ranges of $h$, $v$ and land use are reported in Table 4.


**Table 4: Debris Factor ($DF$) for different water depths ($h$), flow velocities ($v$) and land uses.**

| Values of $h$ and $v$ | Grazing/Agricultural land | Forest | Urban |
|---|---|---|---|
| 0 m < $h$ ≤ 0.25 m | 0 | 0 | 0 |
| 0.25 m < $h$ ≤ 0.75 m | 0 | 0.5 | 1 |
| $h$ > 0.75 OR $v$ > 2 m/s | 0.5 | 1 | 1 |

Based on the FHR, the vulnerability of the population, $V_P$, can be calculated. One assumption is that people are vulnerable at water heights greater than 0.25m. People located in "hospital and social assistance structures", whose vulnerability is

considered as 1 for an FHR > 0.75, represent an exception because the physical condition of people living in such structures makes them more vulnerable. These relationships are summarized in Figure 2.

The method to evaluate the adaptive and coping capacities is based on the hierarchical combination of indicators as shown in Figure 3, where the weights used in the calculation are reported in brackets. The data related to the social indicators have different units of measurement. Therefore, it is necessary to adopt a normalization procedure using value functions

(Mojtahed et al., 2013).





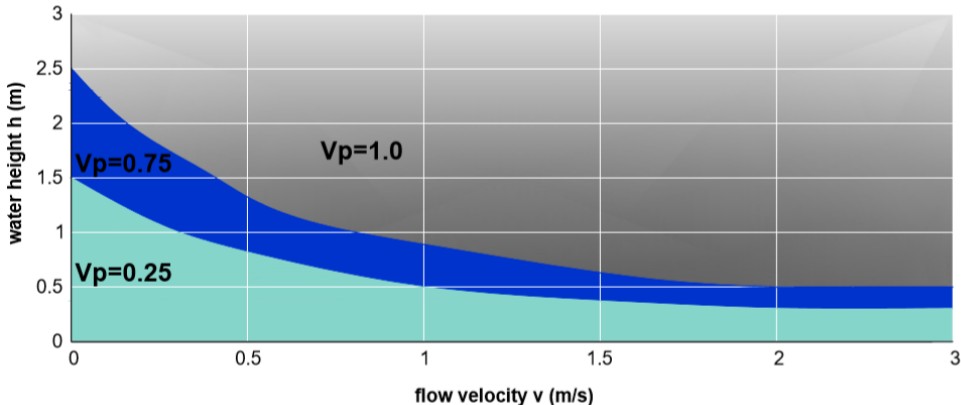

**Figure 2: Vulnerability values for the population ($V_p$) as a function of water depth ($h$) and flow velocity ($v$).**

To evaluate the Coping Capacity, four different variables are included (shown in Figure 4 along with their normalized functions):

- Dependency ratio: the number of citizens aged under 14 and over 65 compared to the total population. A population with a high value of this index implies a reduced ability to adapt to hazardous events.
- Foreigners: the number of foreigners as a percentage of the total population. An area with a high number of

immigrants may react with more difficulty after a flood event and during emergency situations, due to, e.g., language barriers and cultural habits.

- Number of people involved in emergency management: the number of operators who have been trained to manage an emergency in the region, expressed qualitatively as low, medium and high; and
- The frequency at which Civil Protection Plans are updated: Updating is measured in months to years and indicates

how often new hydraulic, urban and technological information is taken into account in Civil Protection Plans.

Similarly, for the Adaptive Capacity, the variables and normalized functions (shown in Figure 5) are described below:

- Gini Index: a measure of the inequality of income distribution within the population. A value of 0 means perfect equality while 1 is complete inequality.
- Number of hospital beds: this is calculated per 1000 people.

- The frequency at which information on hazard and risk are updated: this is measured in months to years and indicates the ability of institutions to communicate the conditions of danger and risk to the population.
- Involvement of citizens: This is based on the number of students, associations such as farmers and professionals, and citizens that can be reached across large areas through social networks (WP7 WSI Team, 2013) to disseminate information. The values in Figure 5d show the maximum achievable value in the three categories of citizen

involvement.

The normalized functions used in the calculation of these indices are shown in Figure 5.





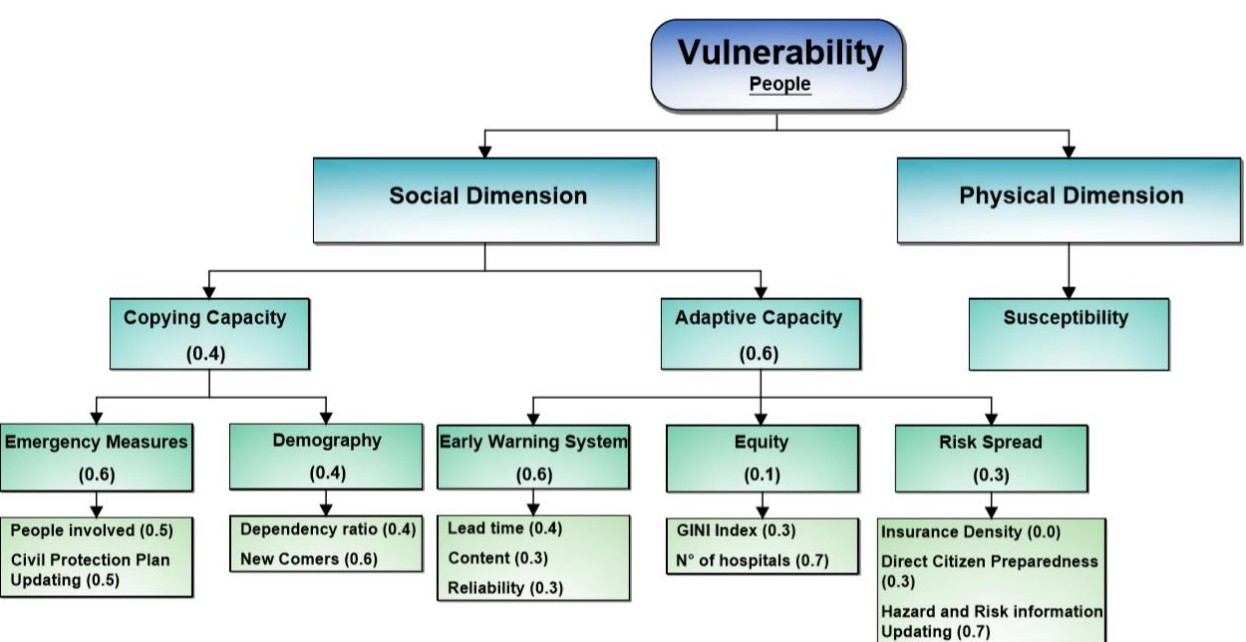

**Figure 3: Hierarchical combination of indicators and relative weights (in brackets) to calculate the vulnerability of the population.**


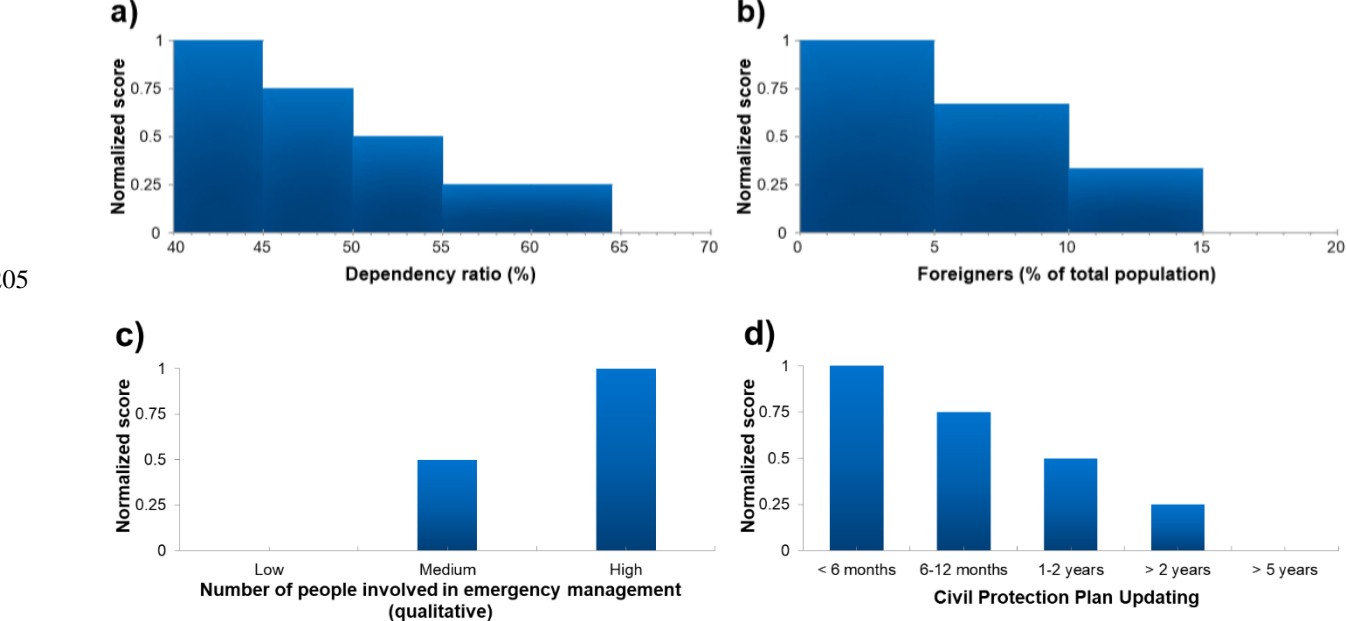

**Figure 4: Variables as normalized index functions for evaluating the Coping Capacity (from De Luca, 2013).**



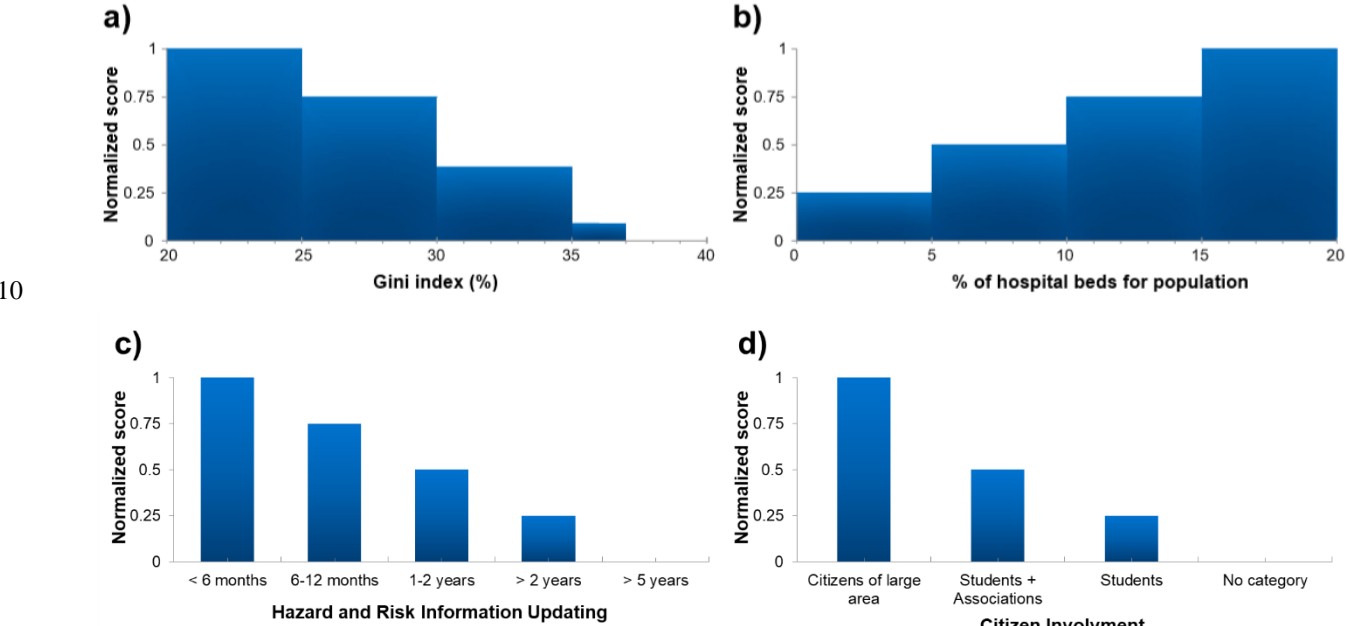

**Figure 5: Variables as normalized index functions for evaluating the Adaptive Capacity (from De Luca, 2013).**

Finally, forecasting systems are evaluated according to the three criteria, where the value functions are shown in Figure 6:

- Reliability: this is linked to the uncertainty of the results from the meteorological forecasts and the hydrological models (Schroter et al., 2008). False alarms can inconvenience people and hinder economic activities and should, therefore, be minimized.
- Lead time (or warning time): the number of hours before an event occurs that was predicted by the early warning system.
- Information Content: the amount of information provided by the forecasting systems, such as the time and the peak of the flooding at several points across the catchment.

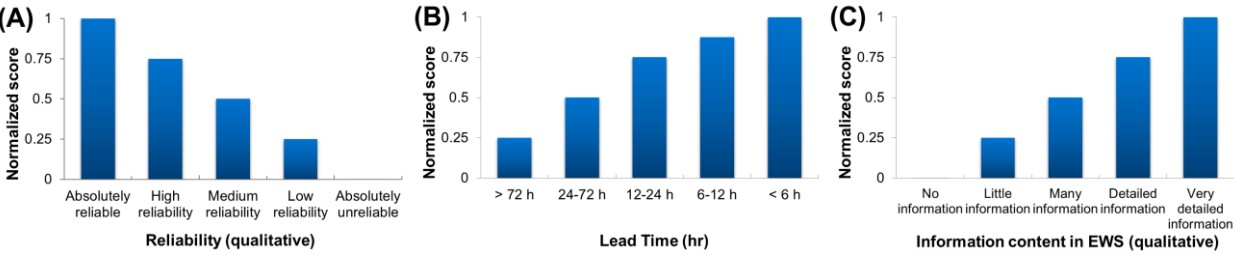

**Figure 6: Normalized function of the indices linked to the forecasting systems: A) reliability, B) lead time, and C) information content (from De Luca, 2013).**





### (ii) Economic activities affected

The vulnerability associated with economic activities, $V_E$, is evaluated using the land use categories in Table 1. Three main aspects are considered: buildings, network infrastructure and agricultural areas. For buildings, which are found in land use types 1 to 5, 14 to 15, 17 to and 23 in Table 1, effects from flooding include collapse due to water pressure and/or

undermining of the foundations. Moreover, solid materials, such as debris and wood, can be carried by a flood and can cause additional damage to structures. A damage function for brick and masonry buildings has been formulated by Clausen and Clark (1990). Regarding losses to indoor goods, laboratory results have shown that at a water height of 0.5m, the loss to indoor goods is around 50%, which is based on an evaluation made by Risk Frontiers, an independent research center sponsored by the insurance industry. The structural vulnerability of the buildings and losses of the associated indoor goods is

shown in Figure 7 as a function of water depth and flow velocity. For the camping land use type 11 (Table 1), the values have been modified based on results from Majala (2001).

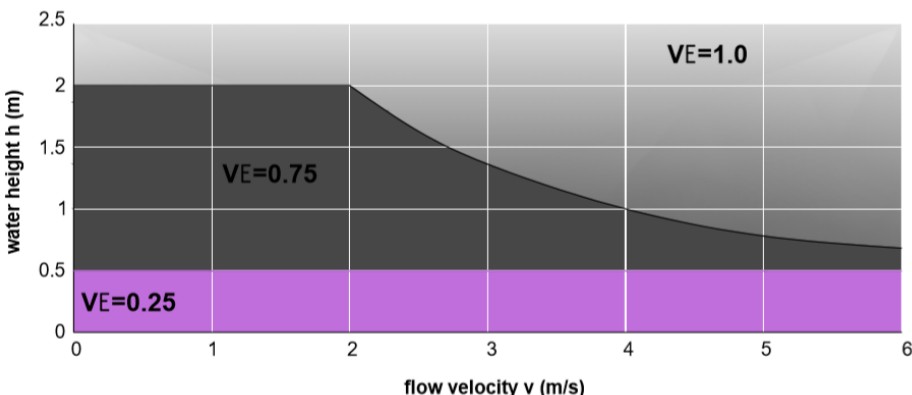

**Figure 7: Vulnerability values of buildings as a function of water depth ($h$) and flow velocity ($v$).**

Vulnerability to the road network is evaluated for land use types 12 and 13 in Table 1. Vulnerability occurs when it is not possible to use the road due to flooding. This can occur with or without structural damage to the road (i.e., this could be a simple inundation or a destruction of the road infrastructure). Based on the estimation of the water height and the critical velocity at which vehicles become unstable during a flood, which are derived from direct observation in laboratory experiments (Reiter, 2000), the vulnerability function for the road network is presented in Figure 8.

Regarding technological and service networks (land use type 16, Table 1), we assume a vulnerability value equal to 1 if the water height and flow velocity is greater than 2 m and 2 m/s, respectively; otherwise it is 0.

    To assess the vulnerability in agricultural areas (land use types 6 and 7 in Table 1), we assume that the damage is related to harvest loss, and when considering higher flow velocity and water depth values, to agricultural buildings and internal goods. However, the highest tolerable height at which agricultural land can be submerged depends on the crop type and

vegetation height. Citeau (2003) provides some examples that take water depth and flow velocity into account, e.g.,





maximum height is 1 m for orchards and 0.5 m for vineyards, and the maximum velocity varies from 0.25 m/s for vegetables and 0.5 m/s for orchards. Concerning cultivation in greenhouses, the maximum damage occurs at a height of 1 m. Finally, high velocities can cause direct damage to cultivated areas but can also lead to soil degradation due to erosion. The vulnerability values for four different types of land as a function of water depth and flow velocity are shown in Figure 9. In

the case of unproductive land (land use type 9 in Table 1), the vulnerability is assumed to be 0.25, regardless of the $h$ and $v$ values.

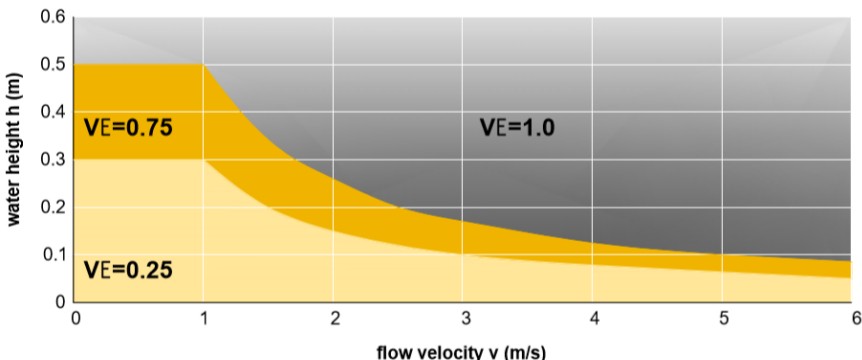

**Figure 8: Vulnerability values of the network infrastructure as a function of water depth ($h$) and flow velocity ($v$).**

**(iii) Environmental and cultural heritage assets affected**

Evers (2006) describes environmental flood susceptibility using three indicators: contamination/pollution, erosion and open space. Contamination is caused by industry, animal/human waste and the stagnation of flooded water. Erosion can produce disturbance to the land surface and to vegetation but can also damage infrastructure. Open spaces are natural areas used for recreational activities, such as tourist attractions and natural protected areas. The approach proposed here is to identify protected areas that could potentially be damaged by a flood. For areas that are susceptible to nutrients, including those

identified as vulnerable in Directive 91/676/CEE (Nitrate), and for those defined as susceptible in Directive 91/271/CEE (Urban Waste), we assume a value of 1 for vulnerability (land use type 20 in Table 1).

Similarly, in the areas identified for habitat and species protection, i.e., sites belonging to the Natura 2000 network established in accordance with the Habitat Directive 92/43/CEE and Birds Directive 79/409/CEE (land use types 8 and 22 in Table 1), the presence of Integrated Pollution Prevention and Control (IPPC) installations and/or other relevant pollution

sources are identified, and the vulnerability is 1. When pollution sources are not identified, the vulnerability is calculated as follows. If the flood velocity is less than or equal to 0.5 m/s and the water depth is less than or equal to 1 m, the vulnerability is 0.25; otherwise the value is 0.5.

Elements classified as "cultural heritage" are considered by the EC to be one of the potential adverse consequences of future flood events. As it is not currently possible to determine the vulnerability associated with different elements of

cultural heritage (land use type 21 in Table 1), we assign a vulnerability of 1 to such elements in a conservative approach.





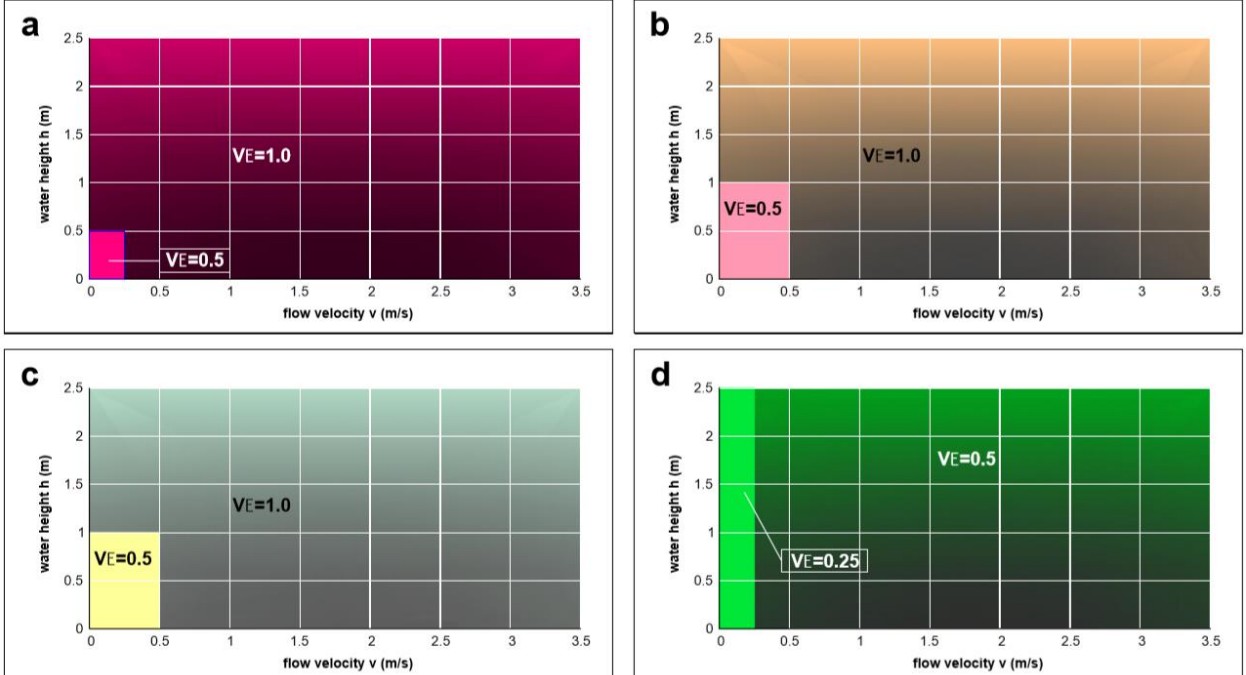

**Figure 9: Vulnerability values as a function of water depth (*h*) and flow velocity (*v*) for: (a) vineyards, (b) orchard and olive trees, (c) vegetables, and (d) natural and semi-natural environments.**

### 2.2.4 Calculation of total risk

The total risk, *R*, can be calculated as a single value based on the following formula:

$$R = \frac{p_P \cdot R_P + p_E \cdot R_E + p_{ECH} \cdot R_{ECH}}{p_P + p_E + p_{ECH}},$$   (4)

where $R_P$, $R_E$ and $R_{ECH}$ represent the risk for the three macro-categories and $p_P$, $p_E$ and $p_{ECH}$ are weights applied to each

macro-category, with values of 10, 1 and 1, respectively, which were defined based on stakeholder interviews. However, these weights can be adjusted based on the priorities of the community. To establish the level of risk (i.e., moderate, medium, high, very high), risk classes are introduced, as provided in Table 5.

The method described above produces total risk for every grid cell in the catchment that is analyzed, taking into account the three scenarios (section 2.2.1) defined in art. 6 of the EU Flood Directive.




**Table 5: Definition of risk classes.**

| Range of R | Description | Risk Category |
|---|---|---|
| 0.1 < R ≤ 0.2 | Moderate risk where social, economic and environmental damage are negligible or zero | R1 |
| 0.2 < R ≤ 0.5 | Medium risk for which minor damage to buildings, infrastructure and environmental heritage is possible, which does not affect the safety of people, the usability of buildings and economic activities | R2 |
| 0.5 < R ≤ 9 | High risk in terms of safety of people, damage to buildings and infrastructure (and/or unavailability of infrastructure), interruption of socio-economic activities and damage related to the environmental heritage | R3 |
| 0.9 < R ≤ 1 | Very high risk including loss of human life and serious injuries to people, serious damage to buildings, infrastructure and environmental heritage, and total disruption of socio-economic activities | R4 |

## 2.3 Cost-benefit analysis

According to EU directive 2007/60/EC, the flood risk plan must contain an analysis of the costs and benefits (hereafter referred to as CBA) that would be generated from each planned intervention. The purpose of the CBA is to compare the efficiency and effectiveness of different alternatives in technological, economic, social and environmental terms. These interventions can be public policies, projects or regulations that can be used to solve a specific problem.

The economic and social aspects related to exposure are considered through the Value Factor, which refers to the economic value of life, the willingness to pay or accept a reward, and the number of direct and indirect users. These factors are designed to support decision makers in assigning monetary value to damages and classifying them, as proposed by Merz et al. (2010). In this analysis, only the direct tangible costs due to damage resulting from a flood event are considered.

### 2.3.1 Determining the effectiveness of an action plan

To determine the effectiveness of the action plan, the modification to the risk class as a result of the intervention must be determined. The Synthetic Index of Risk Reduction (ISRR), which represents the effectiveness of an intervention relative to the current situation, can then be calculated as follows:

$$ISRR = \frac{\sum_{ij} k_{ij} \cdot A_j}{\sum_j A_j},\qquad(5)$$

where $A_j$ is the flooded area after an intervention and $k_{ij}$ are the weights listed in Table 6 for the risk class $i$ before the intervention and $j$ after the intervention. We then use the *ISRR* from equation (5) to calculate the CBA value:





$$CBA = \frac{Costoopera}{ISRR \cdot 10^6},$$ (6)


where *Costopera* is the cost of the intervention.

**Table 6: Weights (*k*) for the Synthetic Index of Risk Reduction (ISRR) for changes in risk before and after the intervention**

| Weights (*k*) | | Risk class before the intervention | | | |
|---|---|---|---|---|---|
| | | R1 | R2 | R3 | R4 |
| Risk class after the intervention | R1 | 0.0 | 10.0 | 20.0 | 30.0 |
| | R2 | -10.0 | 0.0 | 10.0 | 20.0 |
| | R3 | -20.0 | -10.0 | 0.0 | 10.0 |
| | R4 | -30.0 | -20.0 | -10.0 | 0.0 |


### 2.3.2 Financial quantification of the direct damage

To estimate the direct economic impact of the floods, the vulnerability and exposure functions presented in sections 2.2.2 and 2.2.3 are used to calculate the cost of the expected damage for each square meter of different land uses. Maximum damage functions related to the 44 land use classes in CORINE were developed by Huizinga (2007) for the 27 EU member

states based on replacement and productivity costs and their gross national products. The replacement costs for damage to buildings, soil and infrastructure assume complete rebuilding or restoration. Productivity costs are calculated based on the costs associated with an interruption in production activities inside the flooded area. The maximum flood damage values for the EU-27 and various EU countries are provided in Table 7.

**Table 7: Maximum flood damage values (€ / m2) per damage category (Huizinga, 2007).**

| Region/country | Residential building | Commerce | Industry | Road | Agriculture |
|---|---|---|---|---|---|
| EU27 | 575 | 476 | 409 | 18 | 0.59 |
| Italy | 618 | 511 | 440 | 20 | 0.63 |
| Luxembourg | 1443 | 1195 | 1028 | 46 | 1.28 |
| Germany | 666 | 551 | 474 | 21 | 0.68 |
| Netherlands | 747 | 619 | 532 | 24 | 0.77 |
| France | 646 | 535 | 460 | 21 | 0.66 |
| Bulgaria | 191 | 158 | 136 | 6 | 0.20 |



The direct economic impact of the flood is calculated by multiplying the maximum damage values per square meter (in each land use category) by the corresponding areas affected by the floods, weighted by the vulnerability value attributed to each area calculated using the value functions described above. Since the land use map used in this study does not distinguish between industrial and commercial areas, the average of the respective costs per square meter (475.5 € / m²) has been applied. Moreover, in discontinuous urban areas, 50% of the value of the damage related to continuous urban areas (i.e., 309 € / m²) was applied, due to the lower density of buildings in these areas.

The benefits are monetized as the "avoided" damage (to people, real estate, economic activities, protected areas, etc.) following the intervention. The average annual expected damage (*EAD*) can be calculated as follows, where *D* is the damage as a function of the probability of exceeding *P* for a return time *i* (Meyer et al., 2007):

$$EAD = \sum_{i=1}^{k} \frac{D(P_{i-1}) + D(P_i)}{2} \cdot |P_i - P_{i-1}| \tag{7}$$

$$D(P_i) = \sum_i \frac{\sum_j A_{Dj}^i * w_{Dj}}{\sum_j w_{Dj}} \cdot D^i, \tag{8}$$

where $w_{DJ}$ is the weight of the damage class, *j* is the damage category (Table 7) and *D* is the damage value shown in Table 7. In the CBA, this value allows the net benefit related to an intervention to be evaluated, which is expressed as the difference between the *EAD* value for the current situation compared to the *EAD* value after the intervention.

## 2.4 The Brenta-Bacchiglione catchment and the Citizen Observatory on Water

The EU Flood Directive requires that flood risk management plans are produced for each unit of management. In this case, the Brenta-Bacchiglione River catchment coincides with one unit of management (Figure 10). It also includes the Retrone and Astichiello Rivers, and falls within the Veneto Region in Northern Italy, which includes the cities of Padua and Vicenza. The catchment is surrounded by the Beric hills in the south and the Prealpi in the northwest. In this mountainous area, rapid or flash floods occur regularly and are difficult to predict.

Rapid floods generally affect the towns of Torri di Quartesolo, Longare and Montegaldella, although there is also widespread flooding in the cities of Vicenza and Padua, which includes industrial areas and areas with cultural heritage. For example, in 2010, a major flood hit 130 communities and 20,000 individuals in the Veneto region, with Vicenza being one of the most affected municipalities with 20% of the metropolitan area flooded.

Because rapid floods are difficult to predict, early warning systems and prevention measures are of less use in this region. However, reducing damage, and therefore risk, is critical as flood events are frequent and affect several urban areas. In the past, the Upper Adriatic Basin Authority gained practical experience with a Citizen Observatory on Water through the WeSenseIt research project (www.wesenseit.eu), funded under the 7th framework program (FP7-ENV-2012 n° 308429). The objective of this CO, which covered a smaller part of the catchment, was to collect citizen observations from the field, and to obtain a broader and more rapid picture of developments before and during a flood event. As this is sensitive information



that must be trustworthy enough to be acted upon directly, the Civil Protection Agency developed a separate e-collaboration

platform for trained volunteers. The use of the platform resulted in new tasks for organized volunteering groups like Alpinists and the Red Cross.

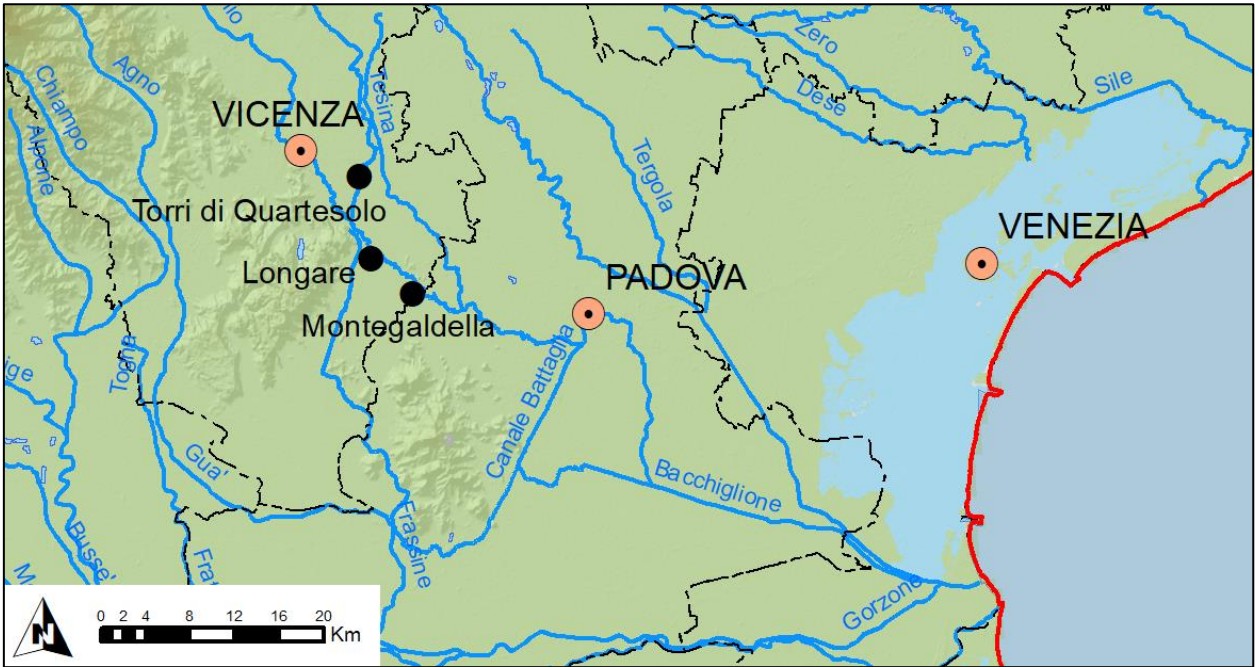

**Figure 10: Location of the Brenta-Bacchiglione catchment and its urban communities.**

This pilot was later adopted by the European Community as a "good practice" example of the application of Directive 2007/60/EC. After the experience in WeSenseIt, the development of a Citizen Observatory on Water at the district scale was included in the prevention measures of the Flood Risk Management Plan (PGRA) for the Brenta-Bacchiglione catchment to strengthen communication channels before and during flood events in accordance with 2007/60/EC. The Citizen Observatory on Water is currently being implemented in the Brenta-Bacchiglione catchment under green infrastructure, with

the objective of increasing resilience and addressing residual risk.

## 3 Results

### 3.1 The situation before the intervention

### 3.1.1 Hazard and risk

The results of the numerical simulations, carried out based on the methodology described in section 2, have shown that in

some sections of the Bacchiglione River, the flow capacity will exceed that of the river channel. This will result in flooding, which will affect the towns of Torri di Quartesolo, Longare and Montegaldella. There will also be widespread flooding in the





cities of Vicenza and Padua, including some industrial areas and others rich in cultural heritage. For a 30-year flood event, the potential flooding could extend to around 40,000 ha, where 25% of the area contains important urban areas with significant architectural assets. In the case of a 100-year flood event, the areas affected by the flood waters increase further, with more than 50,000 ha flooded, additionally affecting agricultural areas.


The results of the simulations are summarized in Tables 8 and 9 in terms of the areas affected in the catchment for different degrees of hazard and risk for 30-, 100- and 300-year flood events. In Figure 11, we provide a map showing the areas at risk in the territory of Padua for a 100-year flood event. Risk classes R1 (low risk) and R2 (medium risk) have the highest areas for all flood event frequencies. Although areas in R3 (high risk) and R4 (very high risk) may comprise a relatively smaller area when compared to the total area at risk, these also coincide with areas of high concentrations of


inhabitants in Vicenza and Padua.

**Table 8: The hazard classes for each return period of flooding in terms of area extension.**

| Hazard class | 30 year return period | 100 year return period | 300 year return period |
|---|---|---|---|
| | Area (km$^2$) | | |
| P1 (Low) | 185.12 | 294.77 | 370.07 |
| P2 (Medium) | 118.87 | 161.82 | 225.67 |
| P3 (High) | 54.18 | 74.55 | 104.61 |
| Total | 358.17 | 531.14 | 700.35 |


**Table 9: The risk classes for each return period of flooding in terms of area extension.**

| Risk class | 30 year return period | 100 year return period | 300 year return period |
|---|---|---|---|
| | Area (km$^2$) | | |
| R1 (Low) | 160.29 | 254.29 | 318.80 |
| R2 (Medium) | 137.26 | 191.89 | 262.03 |
| R3 (High) | 56.70 | 79.23 | 110.29 |
| R4 (Very High) | 3.92 | 5.73 | 9.23 |
| Total | 358.17 | 531.14 | 700.35 |

### 3.1.2 Expected damage

Based on the methodology in section 2.3, the direct damage was calculated for the three flood scenarios: high chance of occurrence (every 30 years), medium (every 100 years) or low (every 300 years). The results are summarized in Table 10.


**Table 10: Valuation of the direct damage due to flood events with difference chances of occurrence.**

| **Scenarios** (chance of flood occurrence) | **Return period** | **Damage (€)** |
|---|---|---|
| High | 30 years | 7,053,068,187 |
| Medium | 100 years | 8,670,252,625 |
| Low | 300 years | 10,853,328,570 |



It can be observed that in the event of very frequent flood events, urban areas will be damaged. Furthermore, it can be observed that passing from an event with a high probability of occurrence to one with medium probability results in a significant increase in the area flooded (i.e., a 48% increase as shown in Table 8) but with a smaller increase in damage (i.e., around 20%). This is explained by the fact that the flooded areas in a 100-year flood event (but not present in a 30-year flood event) are under agricultural use. Similar observations can be made when comparing floods with a low and high probability of occurrence. Substituting the values in Table 10 into equation (7), we obtain an expected average annual damage (EAD) of € 248,517,347.

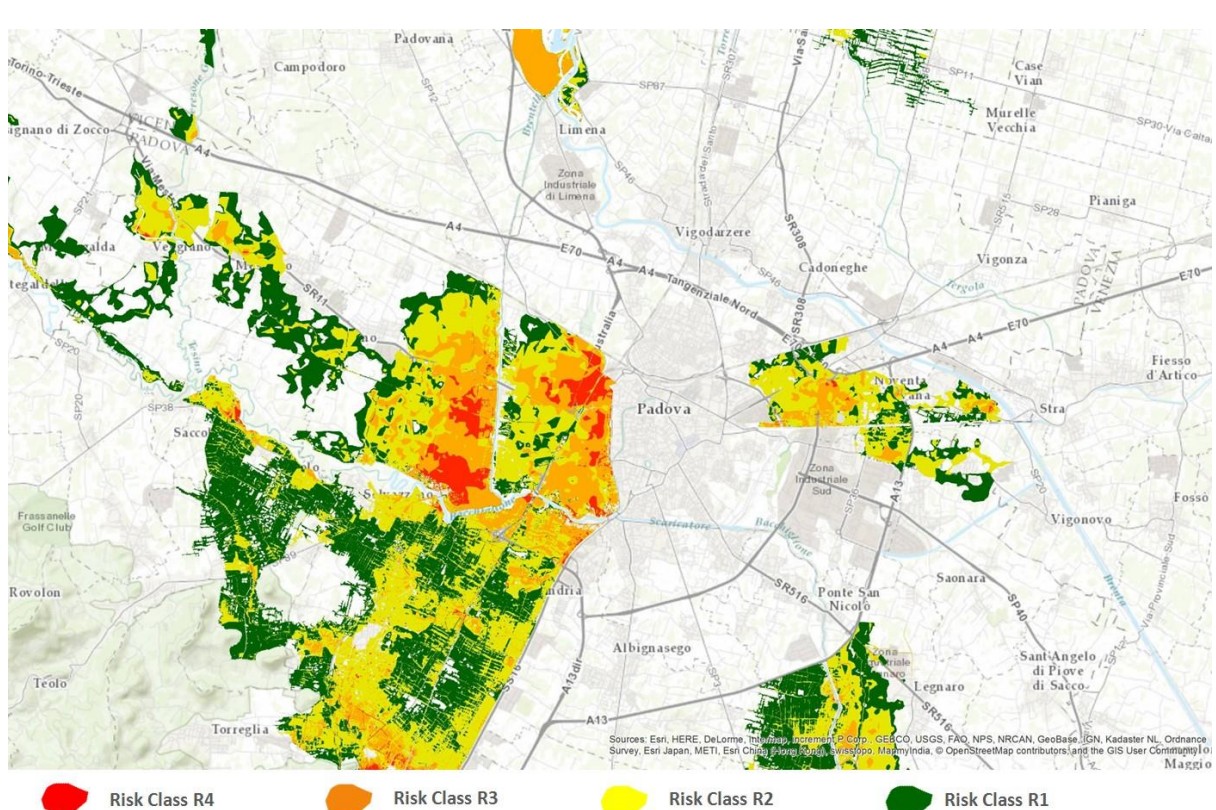

**Figure 11: Risk map for the metropolitan area of Padua for a 100-year flood event. © OpenStreetMap contributors 2019. Distributed under a Creatice Commons BY-SA License.**

**3.2 The result after the intervention of a Citizen Observatory in the Brenta-Bacchiglione catchment**

The results of the numerical simulations, carried out according to the methodology described in section 2, show that the implementation of the CO is able to significantly reduce the damage and consequently the risk for the inhabited areas of Vicenza, Padua, Torri di Quartesolo, Longare and Montegaldella. The hazard remains unchanged (results reported in Table



8). The results of the simulations are provided in Table 11, which summarize the areas affected in the catchment for different degrees of risk for 30-, 100- and 300-year flood events.

**Table 11: The risk classes for each return period of flooding in terms of area affected.**

| Risk class | 30 year return period | 100 year return period | 300 year return period |
|---|---|---|---|
| | Area (km$^2$) | | |
| R1 (Low) | 170.96 | 268.68 | 337.78 |
| R2 (Medium) | 168.99 | 235.18 | 322.41 |
| R3 (High) | 18.19 | 27.19 | 40.04 |
| R4 (Very High) | 0.03 | 0.09 | 0.12 |
| Total | 358.17 | 531.14 | 700.35 |


Comparing results in Tables 9 and 11, although the same areas are at risk, the areas affected in the high (R3) and very high classes (R4) are significantly reduced (R4 to almost zero) but at the detriment of an increase in areas affected in the lower risk classes. The results of the simulations showing areas at risk for a 100-year flood event for the territory of Padua are shown in Figure 12. The reduction in areas at high and very high risk are clearly visible compared to the situation before

the intervention shown in Figure 11.

The results of the numerical simulations carried out according to the methodology described in section 2.2 have shown that the CO is able to reduce the damage from the flood in the three different scenarios due to a reduction in the vulnerability, in particular in terms of improving the coping capacity of the population (Figure 3) through increasing the number of people involved in the emergency, improving the response time and the reliability of the early warning system.

The direct residual damage was calculated for the three flood scenarios with the CO intervention. The results are shown in Table 12. Substituting the values in Table 12 into equation (7), we obtain an EAD of € 111,344,596, which is a 45% reduction in the damage compared to results before the CO intervention.

**Table 12: Valuation of the residual damage due to flood events with difference chances of occurrence.**

| Scenarios (chance of flood occurrence) | Return period | Damage (€) |
|---|---|---|
| High | 30 years | 1,572,774,084 |
| Medium | 100 years | 5,439,785,419 |
| Low | 300 years | 3,419,635,261 |


The CO for Water in the Flood Risk Management Plan of the Eastern Alps has an estimated cost of around € 5,000,000. Based on the assessments made previously, the annual benefit in terms of avoided damage is approximately € 137,172,000, with an ISRR value of 2.5. The CBA index, calculated using equation 6, is equal to 2. This value shows a higher benefit to cost ratio compared with other hydraulic works planned in the Eastern Alps District and demonstrates the feasibility of the

CO approach. For example, using the same methodology, a CBA was undertaken for the construction of a retention basin in the municipalities of Sandrigo and Breganze to improve the hydraulic safety of the Bacchiglione River. Against an expected cost of € 70,700,000, which is much higher than the estimated cost for implementing the CO, a significant reduction in



flooded areas would be obtained although high risk would still be evident in the territory of the city of Padua. In terms of damage reduction with the construction of the retention basin, we would obtain an EAD of € 140,685,400, which is still

higher than that incurred by the implementation of the CO.

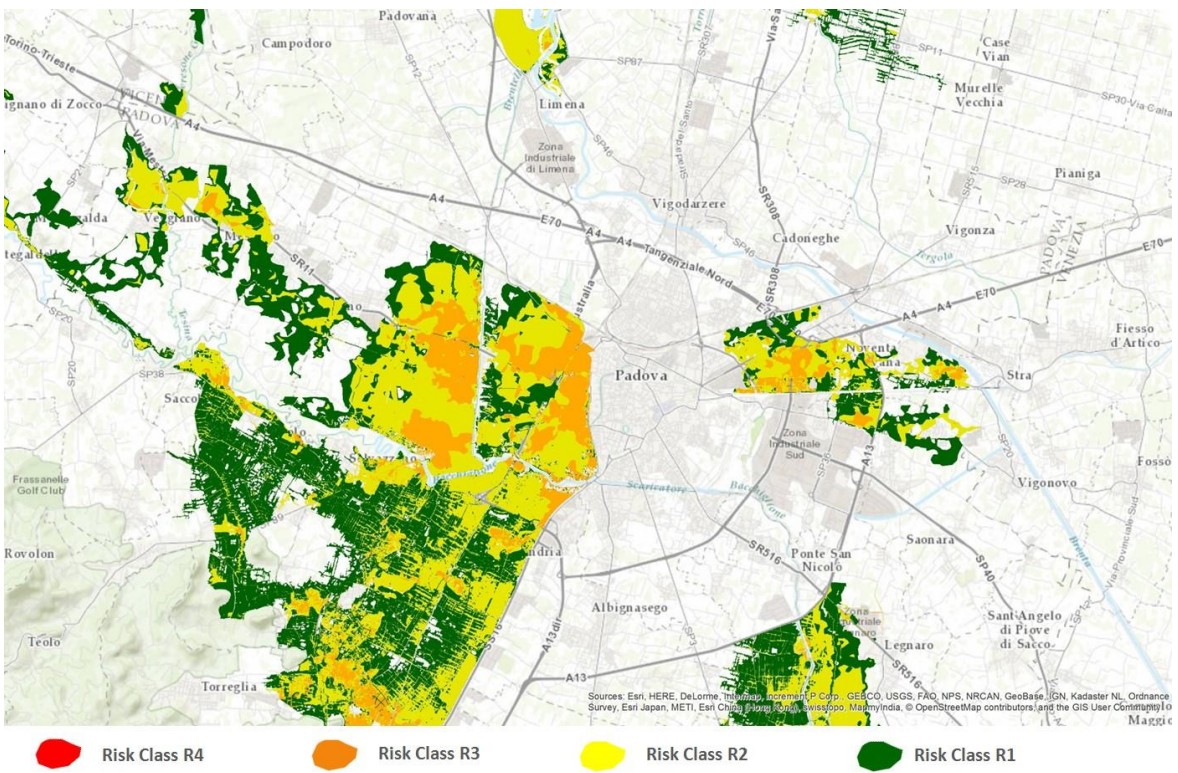

| Risk Class R4 | Risk Class R3 | Risk Class R2 | Risk Class R1 |

**Figure 12: Risk map for the metropolitan area of Padua for a 100-year flood event after the intervention of a Citizen Observatory. © OpenStreetMap contributors 2019. Distributed under a Creatice Commons BY-SA License.**

**4 Discussion and Conclusions**

The lack of available, appropriate and peer-reviewed evaluation methods and evidence on the added value of COs is holding

back the uptake and adoption of COs by policy makers and practitioners. This paper has aimed to fill this gap by presenting and applying a generic methodology for capturing the value of COs by means of a tailored, detailed cost-benefit analysis. The proposed methodology was applied using primary empirical evidence from a CO pilot that was undertaken by the WeSenseIt project in the Brenta-Bacchiglione catchment. As such, the contribution of this paper has been two-fold.

         First, it has outlined and demonstrated the application of a generic methodology for capturing the costs and benefits of

implementing a CO in the FRM domain. The generic nature of the methodology means that it can be applied to other catchments in any part of Italy or other parts of the world that are considering the implementation of a CO for FRM purposes. Secondly, the paper has produced case-specific insights. The CO cost-benefit analysis has shown that the





implementation of a CO in the Brenta-Bacchiglione is able to reduce the damage, and consequently the risk, for the inhabited areas from an expected average annual damage (EAD) of €248,517,347 to €111,344,596, i.e., a reduction of 45%.

The evidence on the costs and benefits of COs for FRM generated by the case study elaborated in this paper provides insights that policy makers, authorities and emergency managers can use to make informed choices about the adoption of COs for improving their respective FRM practices. In Italy, in general, citizen participation in FRM has been relatively limited. The previous strategy in the Brenta-Bacchiglione catchment focused on structural flood mitigation measures, dealing with emergencies, optimizing resources and effective and rapid response mechanisms. The inclusion of a CO on water has

been a true innovation in the FRM strategies of this region. Future research can focus on the application of the methodology in other catchments as well as to other fields of disaster management beyond floods. Such applications will serve to generate a broader evidence base for the validation of the proposed COCBA cost-benefit methodology.

   However, there are also limitations associated with this approach. For example, the cost-benefit analysis presented here did not consider indirect costs, such as those incurred after the event takes place, or in places other than those where the

flooding occurred (Merz et al., 2010). In accordance with other authors (e.g., van der Veen et al., 2003), all expenses related to disaster response (e.g., costs for sandbagging, evacuation) were classified as indirect damage. The presence of the CO in the territory does, however, reduce the costs related to emergency services, securing infrastructure, sandbagging and evacuation, all of which can be substantial during a flood event. Therefore, an analysis that takes indirect costs into account could help to further convince policy makers of the feasibility of a CO solution. Similarly, intangible costs were not

considered, i.e., the values lost due to an adverse natural event where monetary valuation is difficult because the impacts do not have a corresponding market value (e.g., health effects). Furthermore, the vulnerability assessment of economic activities considers only water depth and flow velocity but not additional factors such as contamination, building materials or the duration of the flood event, all of which could be taken into account in estimating the structural damage and monetary losses in the residential, commercial and agricultural sectors.

Despite these limitations, this analysis has highlighted the feasibility of a non-structural flood mitigation choice such as a CO for water compared to the implementation of much more expensive structural measures (e.g., retention areas) in terms of the construction costs and the cost of maintenance over time.

**Acknowledgements**

The research has been partly funded by the FP7 WeSenseIt project (No. 308429) and the Horizon2020 WeObserve project (No. 776740).

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
