# Peer review of "The Value of Citizen Science for Flood Risk Reduction: Cost-benefit Analysis of a Citizen Observatory in the Brenta-Bacchiglione Catchment"

_Hydrology and Earth System Sciences, 2019_

## Referee Comment (RC1) · Anonymous Referee #1 · 30 Jan 2020

General comments

This paper discusses a cost-benefit analysis for citizen observatories based on results in a specific catchment. The content is relevant and will be a valuable addition to citizen science research. Unfortunately, in its current form the paper is difficult to follow and lacks information to fully understand the content. Specifically the citizen observatory need to be explained in much greater detail in the methodology chapter. What exactly does the citizen observatory measure? Are sensors being used and if so for what variables? Are observations only made during floods? How many volunteers were

used? Do the volunteers get paid or is this part of their job? And in what way do these observatories reduce the cost of floods? These questions are very central to this paper and are currently not communicated. Furthermore, for some values (e.g. the weights) no rational or reference is given. A full assessment of the paper can only be made once this information is provided. I therefore encourage the authors to resubmit the manuscript after including this pertinent information and some other revisions.

Specific comments

- In general, the readability of the paper is poor. One option would be to reduce the number of abbreviations. The authors should also avoid long sentences when possible. Phrases such as "detriment of an increase" (L 422) are confusing – does that mean a decrease? In addition the paper could focus more on the main story (i.e., what is written in the abstract).

- There are many assumptions presented in the methodology. The discussion should include a paragraph that discusses the potential effects of these assumptions and the likely uncertainty due to these assumptions.

- Regarding costs (e.g. in table 12): If you are talking about costs in the millions and billions I would not write the value with a precision of one Euro. That is simply not realistic and gives a false impression of certainty. Ideally you would write ranges, or a value +/- another value.

- The literature review and the introduction could have been more extensive. Are there any other citizen science projects that discuss costs vs. benefits? Or at least that discuss costs and financial benefits? How did they try to assess this and why is your method different, or why has this not been done yet?

- L 9: In what way are citizen observatories a recent form of citizen science? (perhaps this will be resolved through a more thorough introduction to citizen observatories)

- L 10: "over a period of time" is a vague statement and could also just be a day, in

which case there is no difference to Blitzes. Is there a minimum required time?

- L 20-21: Worldwide? In Italy?

- L 29: Dominican Republic – not Dominica

- L 42-48: The first part of this paragraph describes collaborative citizen science and not necessarily a citizen observatory. What exactly is the main difference between a CO and a collaborative citizen science project? Is it the inclusion of a public authority? In that case start with that sentence to describe the difference. Not all long-term collaborative citizen science projects are COs.

- L 48-50: Not all of these references are related to a CO. Also, a more thorough explanation of what these individual papers have found would be nicer, rather than just a list of literature.

- 2.1 Input data: A table would provide a clearer picture of the input data. Also when is the pollutant data used? The conclusion specifies that contamination was not considered (L 477). Why is the data used for the hydraulic model (L 109) not included? Add a paragraph describing which data is used for what.

- L 86-87: Is the risk assessed differently at a different scale than mesoscale? Why does this not hold for other scales? This sentence is confusing.

- Figure 1: Is there an arrow missing from the box "Depth, Velocity, Persistence"? Why do the "Land Use Map" and "People distribution map" have a different design? Also capitalization should be uniform.

- L 93-98 and Table 1: would fit better into 2.2.2

- L 100: "must be addressed" –> by whom or for what?

- L 109: Which hydraulic model?

- Table 2: What does the sign "%" mean? Is this meant to be a "-"? (same in table 3)

- L 140-142: "Greater awareness tends to correspond to greater preparation if an event takes place." Please add a source for this statement.

- Figure 3: Where do the weights come from? Are they assumed by the author or based on literature? What is the reasoning behind the weights? Why is "Insurance density" stated, when the weight is 0? What is the uncertainty for these weights and how do they affect the results?

- Figure 4: What is the normalized index function and how was this value computed? What does it mean?

- Figures 4-6 could also be included in a supplementary material. In general section 2.2 is rather long and according to the abstract not central to this paper. Parts of this section could be included in a supplementary material.

- L 217: You could add that people are also less likely to take warnings serious in future.

- L 242: Why are road inundations lumped together, regardless of whether or not the infrastructure is damaged? Surely the cost differs significantly in those two scenarios?

- L 256: Does this mean that the vulnerability depends on the seasons, i.e., on when the crops are growing?

- L 260-266: I do not understand why "contamination/ pollution, erosion" and "open space" are lumped together here? Also apart from describing what an open space is, the connection to flood susceptibility unclear. Please make the connections clearer to the reader.

- Why is the vulnerability high (i.e., 1) when "Integrated Pollution Prevention and Control (IPPC) installations" are present? Shouldn't these installations reduce the vulnerability?

- L 273-274: Please rephrase. I suppose "cultural heritage" is not actually considered

an "adverse consequences of future flood events".

- Figure 9: Why does water height not play a factor in (d)?

- L 284: Repeat what the macro-categories are.

- L 287: I would rephrase "moderate" as "low", as moderate is actually a synonym of medium. (same in Table 5) You actually use "low" already in Table 9 – please homogenize.

- L 301: This is the first time you mention a reward. What reward do you mean? "Direct and indirect users" -> of what?

- L 308: One variable should not have multiple letters in an equation. This could be considered to be I*S*R*R. (Also applies to some other formulas in this paper.)

- Table 6: Where do the weights come from? Are they calculated or taken from another source?

- L 330: superscript for 2

- L 349: Please define what you mean with a "unit of management".

- L 358-359: "Because rapid floods are difficult to predict, early warning systems and prevention measures are of less use in this region." I have some doubts regarding this statement. Surely early warning systems (sirens) and prevention measures (reforestation) still reduce the risks associated with rapid floods? Please offer a source or rephrase.

- L 361: superscript for th

- L 365: Was data that was not collected by trained volunteers discarded? How many volunteers were trained and how many were not trained? This section needs to be expanded considerably as described in the general comments.

- L 377: "The situation before the intervention" The word situation is rather vague, but
more importantly you do not describe what exactly that intervention is (see general comments).

- L 279: Specify what you mean with numerical simulations. Do you mean the hydraulic model? Very little information is provided about the modelling that was done. The reference to section 2 could be more specific, do you mean section 2.2.1?

- Table 8: What is P1, P2 and P3?

- L 415: "implementation of the CO" –> not fully explained what that is, see general comments.

- Table 11: Please explain the difference to table 9 (not just the different values, but the different meaning).

- L 426-432: This paragraph is rather unclear, but this will likely improve once the general comments are addressed.

- Table 12: caption "different" not "difference"; also I would not just include the residual damage, but also the original and then the difference. Or you could make a plot with both the original and residual costs.

- L 436: How did you calculate the cost for CO? What are the cost components? Is it sensor costs, maintenance costs or personnel costs? Over what time do these costs accumulate?

- I have found at least one reference in the text that is not in the list of references. Please double check.

---

## Referee Comment (RC2) · Anonymous Referee #2 · 26 Mar 2020

Review This paper presents a framework that combines modelling with data collected through a citizen observatory running in northern Italy. The idea that citizen science could successfully be used to reduce the risk of catchments to the effects of flooding is interesting and timely. Having said that, I find several shortcomings in this paper that need to be fixed before it can be considered for publication in HESS. These shortcomings are: 1. The core message of the paper for me should be that citizen science and modelling are effective in reducing risk. However, the paper describes extensively the modelling approach whereas the citizen science part is very vaguely described.

[Figure]

Creative Commons BY license logo

The two together don't make a convincing story. 2. The concept of risk used in this paper is a known one. The paper uses quite some space in the methods section to go through the component of risk but it does it in a very confusing way. For example, one would expect that Fig 1 is used through the methods to arrive to the risk estimates, but it isn't, and therefore the presentation of the method is muddled. I would suggest re-writing the risk section, shortening and focusing it on the application to flooding risk, using a comprehensive figure to guide the reader. 3. The modelling approach uses many coefficients that lead to the estimation of risk. These coefficients presented in several figures, were taken apparently from a number of sources (not always disclosed) and are not subject to a thorough sensitivity analysis. The results of the modelling are heavily determined by the coefficients adopted so it is critically to explain these very well. I don't recommend this additional explanation is included in the methods section, but it has to be properly documented in Supplementary material. Without this, it will be extremely hard to test and to apply this method elsewhere. 4. The description of the Citizen Observatory is the most disappointing. It doesn't inform the reader in terms of data collected, how did the work, how it was implemented, etc. This should be lot more prominent in this paper. I do not recommend the publication of this paper because it requires significant re-writing.

Specific comments The abstract is not informative, and it should reflect the key innovation of this study. Presumably, the successful linking of risk modelling and citizen science should be the key message in the abstract. For the case study, the findings seem misleading because the study does not cost a infra-structure adaptive intervention, only the roughly estimated costs of the potential damage.

Introduction

L20-30 This section is irrelevant for the story of the paper, and not well written. I suggest to delete it, and add instead a clear definition of risk specific to flooding that introduce the paper.

L37-39 References required for the statement 'exponential growth' in citizen science. L40 unclear why references were added after ..."Among the various form of citizen science". Instead of references I would expect a list of the different forms. This whole sentence needs re-writing.

L61-63 and L65-68

L70 section 2.1 needs to be more carefully described. Details needed to interpret results.

L74 section 2.2 See general comment. This section up to 2.4 is so poorly written that it is hard to keep track of the method used, sources of information and assumptions made. In addition, the calculation of risk must be done from the beginning with a focus on flooding risk, the aim of this study.

Fig. 1 is not self-explanatory and it is not connected properly to the text that follows in 2.2 and 2.3.

Table 1 could be sent to supplementary material. It is not critical to the results.

L100-110. A description and testing of the hydrological model are needed because the reference included is not a peer-reviewed source and can't be accessed by the reader. This could be added to supplementary material

Table 2, Table 3 and Table 4. It should be clear what the sources for these coefficients are and why these are accepted to be reasonable without performing a sensitivity analysis.

L174 why is the use of 'value functions' the preferred approach, and what is the uncertainty associated with them? I don't see an uncertainty analysis conducted here.

Fig. 3 I wonder why this figure is presented in addition to Fig 1, and using slightly different terms and approach?

Fig. 4, 5, 6, 7 and 8. What is the uncertainty associated with this coefficients?

L214 Are forecasting systems the same as 'early warning systems' of Fig. 3? This is confusing.

Fig. 9 It is very hard to understand this figure. The caption is not self-explanatory.

L284. The component of the equation should be explained. This equation should be introduced after Fig 1 when the concepts are explained.

L295. This section on C/B analysis is not clear at all. I would have expected that the costs would be the cost of remedial and/or preventive actions, which are not clearly explained here. What are the units of ISRR? I would guess hectares of km2. And of CBA?

Table 7. I would expect large variability in these values. No uncertainty analysis performed.

L348 section 2.4. This should be one of the key section of the paper, but it is unfortunately very vague and doesn't provide the reader much information on how the citizen observatory worked, data collected, for how long etc.

Results. In view of all the methodological questions, it seems pointless to go through the results. From the paragraph included in L426-432, it seems that the paper should have explained the simulations of risk and damage, and what the citizen observatory programme did and achieved, which here remains as a black box.

End of review

---

## Author Comment (AC1) · 13 May 2020

We would like to thank the reviewer for the very detailed and insightful comments to the first draft of the paper, which have helped to greatly improve it. We agree that many details needed further clarification. We hope that this improved version is easier to understand and makes the key points in a clearer fashion. We have also attached these comments and responses as a table in PDF format.

GENERAL COMMENTS COMMENT: This paper discusses a cost-benefit analysis for

citizen observatories based on results in a specific catchment. The content is relevant and will be a valuable addition to citizen science research. RESPONSE: We thank the reviewer for their positive comment.

COMMENT: Unfortunately, in its current form the paper is difficult to follow and lacks information to fully understand the content. Specifically the citizen observatory need to be explained in much greater detail in the methodology chapter. What exactly does the citizen observatory measure? Are sensors being used and if so for what variables? Are observations only made during floods? How many volunteers were used? Do the volunteers get paid or is this part of their job? And in what way do these observatories reduce the cost of floods? These questions are very central to this paper and are currently not communicated. RESPONSE: We have added a separate section before the methodology to fully describe the citizen observatory including what is measured and information about the volunteers. The cost reduction is related to decreasing the vulnerability component of risk, which is described in more detail in the methodology.

COMMENT: Furthermore, for some values (e.g. the weights) no rational or reference is given. RESPONSE: See the responses to the specific comments below regarding the different values used in the analysis.

COMMENT: A full assessment of the paper can only be made once this information is provided. I therefore encourage the authors to resubmit the manuscript after including this pertinent information and some other revisions. RESPONSE: We understand and have taken careful note of the reviewer's comments and addressed them as detailed below in response to the more specific comments.

SPECIFIC COMMENTS COMMENT: In general, the readability of the paper is poor. One option would be to reduce the number of abbreviations. The authors should also avoid long sentences when possible. Phrases such as "detriment of an increase" (L 422) are confusing – does that mean a decrease? In addition the paper could focus more on the main story (i.e., what is written in the abstract). RESPONSE: We have

thoroughly edited the manuscript to improve the readability.

COMMENT: There are many assumptions presented in the methodology. The discussion should include a paragraph that discusses the potential effects of these assumptions and the likely uncertainty due to these assumptions. RESPONSE: A paragraph was added to the Discussion and Conclusions section within the limitations regarding the assumptions made in the methodology, the potential effects, etc.

COMMENT: Regarding costs (e.g. in table 12): If you are talking about costs in the millions and billions I would not write the value with a precision of one Euro. That is simply not realistic and gives a false impression of certainty. Ideally you would write ranges, or a value +/- another value. RESPONSE: These changes have been made.

COMMENT: The literature review and the introduction could have been more extensive. Are there any other citizen science projects that discuss costs vs. benefits? Or at least that discuss costs and financial benefits? How did they try to assess this and why is your method different, or why has this not been done yet? RESPONSE: The literature review in the introduction has been expanded regarding studies that have considered the costs and benefits associated with citizen science projects, e.g., the time invested by researchers in engaging and training citizens (Thornhill et al., 2016); to relate cost and participant performance for hydrometric observations in order to estimate the cost per observation (Davids et al., 2019); to estimate the costs as data-related costs, staff costs and other costs; and the benefits in terms of scientific benefits, public engagement benefits and the benefits of strengthened capacity of participants (Blaney et al., 2016); and to compare citizen science data and in-situ data (Goldstein et al., 2014; Hadj-Hammou et al., 2017). This approach is different because it calculates the damage costs that would be avoided if a citizen observatory was fully implemented in this basin.

COMMENT: L 9: In what way are citizen observatories a recent form of citizen science? (perhaps this will be resolved through a more thorough introduction to citizen observatories) RESPONSE: We have now specified the characteristics of citizen observatories further.

COMMENT: L 10: "over a period of time" is a vague statement and could also just be a day, in which case there is no difference to Blitzes. Is there a minimum required time? RESPONSE: Although we think the current expression was already clearly distinguishing between one off Blitzes and an extended time period in which a CO operates, we have further refined this.

COMMENT: L 20-21: Worldwide? In Italy? RESPONSE: The introduction has been rewritten so this is no longer relevant.

COMMENT: L 29: Dominican Republic – not Dominica RESPONSE: The introduction has been rewritten so this is no longer relevant.

COMMENT: L 42-48: The first part of this paragraph describes collaborative citizen science and not necessarily a citizen observatory. What exactly is the main difference between a CO and a collaborative citizen science project? Is it the inclusion of a public authority? In that case start with that sentence to describe the difference. Not all long-term collaborative citizen science projects are COs. RESPONSE: We have specified the characteristics of citizen observatories further and explained the required links to policy and the inclusion of public authorities.

COMMENT: L 48-50: Not all of these references are related to a CO. Also, a more thorough explanation of what these individual papers have found would be nicer, rather than just a list of literature. RESPONSE: We have removed references to citizen science rather than citizen observatories from the list. Moreover, we have summarised the findings from these papers as suggested.

COMMENT: 2.1 Input data: A table would provide a clearer picture of the input data. Also when is the pollutant data used? The conclusion specifies that contamination was not considered (L 477). Why is the data used for the hydraulic model (L 109) not

included? Add a paragraph describing which data is used for what. RESPONSE: We have summarized the input data in a table. We considered the presence of punctual and widespread sources of pollutants in the vulnerability assessment, but we did not use a model to evaluate the actual propagation of contamination. We have included the data coming from the hydraulic model and included more information about the model in the Supplementary Materials. A column was added to the input data table indicating where the data sets are used in the methodology.

COMMENT: L 86-87: Is the risk assessed differently at a different scale than mesoscale? Why does this not hold for other scales? This sentence is confusing. RESPONSE: To avoid confusion, we have removed this sentence. However, the assessment of risk does differ in terms of the data used at different scales. For example, at the microscale, assessments are characterized by a high level of detail regarding the exposure so they would take individual assets at risk into account, such as buildings, vehicles or infrastructure. At the mesoscale, exposure information is usually based on aggregation, using a land use map to assess exposure, as is carried out in this paper.

COMMENT: Figure 1: Is there an arrow missing from the box "Depth, Velocity, Persistence"? Why do the "Land Use Map" and "People distribution map" have a different design? Also capitalization should be uniform. RESPONSE: This figure has been modified to better align with the text and to address the comments of the reviewer.

COMMENT: L 93-98 and Table 1: would fit better into 2.2.2 RESPONSE: The original Table 1 has now been moved to the Supplementary Material (Table S1). We have also shifted text around in the methods section to improve clarity.

COMMENT: L 100: "must be addressed" –> by whom or for what? RESPONSE: We have modified this to read: According to Article 6 of the 2007/60/CE Flood Directive (EU, 2007), when local authorities implement a Flood Risk Management Plan, three hazard scenarios must be considered: etc.

COMMENT: L 109: Which hydraulic model? RESPONSE: Details of the hydraulic and

hydrological model have been added to the Supplementary Material.

COMMENT: Table 2: What does the sign "%" mean? Is this meant to be a "-"? (same in table 3) RESPONSE: This was meant to be a "-" sign and has been replaced in both tables.

COMMENT: L 140-142: "Greater awareness tends to correspond to greater preparation if an event takes place." Please add a source for this statement. RESPONSE: This sentence has been modified as follows: Some studies have found that if citizens have directly experienced a flood, their perception of flood risk is higher (e.g., Thistlethwaite et al., 2018) although the factors that determine flood risk perception are varied. Moreover, the results from different studies can be ambiguous and/or contradictory (Lechowska, 2018).

COMMENT: Figure 3: Where do the weights come from? Are they assumed by the author or based on literature? What is the reasoning behind the weights? Why is "Insurance density" stated, when the weight is 0? What is the uncertainty for these weights and how do they affect the results? RESPONSE: We have added the following text: These values have been developed by the Provincia Autonoma di Trento (2006) from decades of experience with understanding exposure related to flood risk. Moreover, they have been tested and shown to be valid within AAWA. The insurance density is one of the components of risk spread in the guidelines from ISPRA (2012) and appears in Figure 3 for completeness. However, in the case of the Brenta-Bacchiglione catchment, the insurance density is 0.0 because insurance companies do not currently offer premiums to protect goods against flood damage.

COMMENT: Figure 4: What is the normalized index function and how was this value computed? What does it mean? RESPONSE: This is now Figure S1 in the Supplementary Material. This has been referred to as a value function to be consistent throughout the text. We also explain what value functions are.

COMMENT: Figures 4-6 could also be included in a supplementary material. In general
section 2.2 is rather long and according to the abstract not central to this paper. Parts of this section could be included in a supplementary material. RESPONSE: Figures 4 to 6 have been added to the Supplementary Material.

COMMENT: L 217: You could add that people are also less likely to take warnings serious in future. RESPONSE: This statement has been added to the relevant section of the paper.

COMMENT: L 242: Why are road inundations lumped together, regardless of whether or not the infrastructure is damaged? Surely the cost differs significantly in those two scenarios? RESPONSE: Here the vulnerability is related to whether the road network can be used or not, i.e., at what stage cars become unstable during a flood. Hence, the vulnerability values are based on the estimation of the critical water height and velocity for the stability of vehicles during a flood (and have nothing to do with damage to the network infrastructure). The data have been derived from direct observation in laboratory experiments and from a report on the literature in this area (Reiter, 2000; Shand et al., 2011). In the estimation of the direct costs of the flood, the costs for damage to buildings, soil and infrastructure (including roads) assumes complete rebuilding or restoration and therefore is dealt with separately.

COMMENT: L 256: Does this mean that the vulnerability depends on the seasons, i.e., on when the crops are growing? RESPONSE: No, this refers to Corine Land Class 9: Unproductive land, which is defined as unproductive for the whole year.

COMMENT: L 260-266: I do not understand why "contamination/ pollution, erosion" and "open space" are lumped together here? Also apart from describing what an open space is, the connection to flood susceptibility unclear. Please make the connections clearer to the reader. RESPONSE: We have modified this section and focussed on contamination/pollution and erosion.

COMMENT: Why is the vulnerability high (i.e., 1) when "Integrated Pollution Prevention and Control (IPPC) installations" are present? Shouldn't these installations reduce

the vulnerability? RESPONSE: This has been modified as follows: …the presence of relevant pollution sources was identified (Tables 1 and S1) and assigned a vulnerability of 1.

COMMENT: L 273-274: Please rephrase. I suppose "cultural heritage" is not actually considered an "adverse consequences of future flood events". RESPONSE: We have removed this phrase.

COMMENT: Figure 9: Why does water height not play a factor in (d)? RESPONSE: From experimentation (Citeau, 2003), the height of the water does not affect the vulnerability of natural and semi-natural environments up to a water height of 2.5m.

COMMENT: L 284: Repeat what the macro-categories are. RESPONSE: These macro-categories have been repeated although we have also modified these sections.

COMMENT: L 287: I would rephrase "moderate" as "low", as moderate is actually a synonym of medium. (same in Table 5) You actually use "low" already in Table 9 – please homogenize. RESPONSE: This has been modified in the text and in Table 5.

COMMENT: L 301: This is the first time you mention a reward. What reward do you mean? "Direct and indirect users" -> of what? RESPONSE: We have modified this section so this is no longer relevant.

COMMENT: L 308: One variable should not have multiple letters in an equation. This could be considered to be $I*S*R*R$. (Also applies to some other formulas in this paper.) RESPONSE: We do not completely agree with this comment. For example, evapotranspiration is commonly referred to as ET, potential evapotranspiration as PET and we can find many examples in the literature (including in HESS) of multiple letter variables, e.g. RMSE, NSE, etc., which are used very regularly in hydrology.

COMMENT: Table 6: Where do the weights come from? Are they calculated or taken from another source? RESPONSE: The weights in this table have been determined through expert consultation within the Alto-Adriatico Water Authority (AAWA), supported by the guidelines from ISPRA (2012).

COMMENT: L 330: superscript for 2 RESPONSE: This has been corrected.

COMMENT: L 349: Please define what you mean with a "unit of management". RE-SPONSE: A unit of management is part of terminology from the INSPIRE Directive but we have removed this to avoid any confusion.

COMMENT: L 358-359: "Because rapid floods are difficult to predict, early warning systems and prevention measures are of less use in this region." I have some doubts regarding this statement. Surely early warning systems (sirens) and prevention measures (reforestation) still reduce the risks associated with rapid floods? Please offer a source or rephrase. RESPONSE: We have modified this paragraph and removed this statement.

COMMENT: L 361: superscript for th RESPONSE: This has been corrected.

COMMENT: L 365: Was data that was not collected by trained volunteers discarded? How many volunteers were trained and how many were not trained? This section needs to be expanded considerably as described in the general comments. RESPONSE: A new section was added to the paper about the citizen observatory, which addresses these questions. Citizen data were not discarded but used to investigate their value in complementing hydrological modelling (Mazzoleni et al., 2017; 2018).

COMMENT: L 377: "The situation before the intervention" The word situation is rather vague, but more importantly you do not describe what exactly that intervention is (see general comments). RESPONSE: We have modified many headings to replace intervention with implementation of the CO on flood risk management.

COMMENT: L 379: Specify what you mean with numerical simulations. Do you mean the hydraulic model? Very little information is provided about the modelling that was done. The reference to section 2 could be more specific, do you mean section 2.2.1? RESPONSE: This has been modified to indicate numerical simulations from the hy-

draulic model, and the reader is then referred to the Supplementary Materials.

COMMENT: Table 8: What is P1, P2 and P3? RESPONSE: These have been replaced with the hazard classes of low, medium and high.

COMMENT: L 415: "implementation of the CO" –> not fully explained what that is, see general comments. RESPONSE: A new section was added to the paper about the citizen observatory.

COMMENT: Table 11: Please explain the difference to table 9 (not just the different values, but the different meaning). RESPONSE: The difference between the two tables is the amount of area flooded by risk class and flood hazard scenario before (Table 9) and after (Table 11) implementation of the CO. The captions have been modified to clarify this point and additional text has been added.

COMMENT: L 426-432: This paragraph is rather unclear, but this will likely improve once the general comments are addressed. RESPONSE: We removed this paragraph but provided more details on how the risk changes with and without the implementation of the CO in the methodology.

COMMENT: Table 12: caption "different" not "difference"; also I would not just include the residual damage, but also the original and then the difference. Or you could make a plot with both the original and residual costs. RESPONSE: This caption has been corrected. The original costs and the differences have been added.

COMMENT: L 436: How did you calculate the cost for CO? What are the cost components? Is it sensor costs, maintenance costs or personnel costs? Over what time do these costs accumulate? I have found at least one reference in the text that is not in the list of references. Please double check. RESPONSE: Table S2 has been added to the Supplementary Material, which is a breakdown of the costs of the CO.

COMMENT: I have found at least one reference in the text that is not in the list of references. Please double check. RESPONSE: All references were checked to ensure

that they are in the reference list.

Please also note the supplement to this comment:
https://www.hydrol-earth-syst-sci-discuss.net/hess-2019-627/hess-2019-627-AC1-supplement.pdf

─────────────────────────

**Supplement:**

**Response to Reviewer 1**

We would like to thank the reviewer for the very detailed and insightful comments to the first draft of the paper, which have helped to greatly improve it. We agree that many details needed further clarification. We hope that this improved version is easier to understand and makes the key points in a clearer fashion.

| Comments from Reviewer 1 | Response to Comments |
|---|---|
| This paper discusses a cost-benefit analysis for citizen observatories based on results in a specific catchment. The content is relevant and will be a valuable addition to citizen science research. | We thank the reviewer for their positive comment. |
| Unfortunately, in its current form the paper is difficult to follow and lacks information to fully understand the content. Specifically the citizen observatory need to be explained in much greater detail in the methodology chapter. What exactly does the citizen observatory measure? Are sensors being used and if so for what variables? Are observations only made during floods? How many volunteers were used? Do the volunteers get paid or is this part of their job? And in what way do these observatories reduce the cost of floods? These questions are very central to this paper and are currently not communicated. | We have added a separate section before the methodology to fully describe the citizen observatory including what is measured and information about the volunteers. The cost reduction is related to decreasing the vulnerability component of risk, which is described in more detail in the methodology. |
| Furthermore, for some values (e.g. the weights) no rational or reference is given. | See the responses to the specific comments below regarding the different values used in the analysis. |
| A full assessment of the paper can only be made once this information is provided. I therefore encourage the authors to resubmit the manuscript after including this pertinent information and some other revisions. | We understand and have taken careful note of the reviewer's comments and addressed them as detailed below in response to the more specific comments. |
| **Specific comments** | |
| In general, the readability of the paper is poor. One option would be to reduce the number of abbreviations. The authors should also avoid long sentences when possible. Phrases such as "detriment of an increase" (L 422) are confusing – does that mean a decrease? In addition the paper could focus more on the main story (i.e., what is written in the abstract). | We have thoroughly edited the manuscript to improve the readability. |
| There are many assumptions presented in the methodology. The discussion should include a paragraph that discusses the potential effects of these assumptions and the likely uncertainty due to these assumptions. | A paragraph was added to the Discussion and Conclusions section within the limitations regarding the assumptions made in the methodology, the potential effects, etc. |

| Comments from Reviewer 1 | Response to Comments |
|---|---|
| Regarding costs (e.g. in table 12): If you are talking about costs in the millions and billions I would not write the value with a precision of one Euro. That is simply not realistic and gives a false impression of certainty. Ideally you would write ranges, or a value +/- another value. | These changes have been made. |
| The literature review and the introduction could have been more extensive. Are there any other citizen science projects that discuss costs vs. benefits? Or at least that discuss costs and financial benefits? How did they try to assess this and why is your method different, or why has this not been done yet? | The literature review in the introduction has been expanded regarding studies that have considered the costs and benefits associated with citizen science projects, e.g., the time invested by researchers in engaging and training citizens (Thornhill et al., 2016); to relate cost and participant performance for hydrometric observations in order to estimate the cost per observation (Davids et al., 2019); to estimate the costs as data-related costs, staff costs and other costs; and the benefits in terms of scientific benefits, public engagement benefits and the benefits of strengthened capacity of participants (Blaney et al., 2016); and to compare citizen science data and in-situ data (Goldstein et al., 2014; Hadj-Hammou et al., 2017). This approach is different because it calculates the damage costs that would be avoided if a citizen observatory was fully implemented in this basin. |
| L 9: In what way are citizen observatories a recent form of citizen science? (perhaps this will be resolved through a more thorough introduction to citizen observatories) | We have now specified the characteristics of citizen observatories further. |
| L 10: "over a period of time" is a vague statement and could also just be a day, in which case there is no difference to Blitzes. Is there a minimum required time? | Although we think the current expression was already clearly distinguishing between one off Blitzes and an extended time period in which a CO operates, we have further refined this. |
| L 20-21: Worldwide? In Italy? | The introduction has been rewritten so this is no longer relevant. |
| L 29: Dominican Republic – not Dominica | The introduction has been rewritten so this is no longer relevant. |
| L 42-48: The first part of this paragraph describes collaborative citizen science and not necessarily a citizen observatory. What exactly is the main difference between a CO and a collaborative citizen science project? Is it the inclusion of a public authority? In that case start with that sentence to describe the difference. Not all long-term collaborative citizen science projects are COs. | We have specified the characteristics of citizen observatories further and explained the required links to policy and the inclusion of public authorities. |

| Comments from Reviewer 1 | Response to Comments |
|---|---|
| L 48-50: Not all of these references are related to a CO. Also, a more thorough explanation of what these individual papers have found would be nicer, rather than just a list of literature. | We have removed references to citizen science rather than citizen observatories from the list. Moreover, we have summarised the findings from these papers as suggested. |
| 2.1 Input data: A table would provide a clearer picture of the input data. Also when is the pollutant data used? The conclusion specifies that contamination was not considered (L 477). Why is the data used for the hydraulic model (L 109) not included? Add a paragraph describing which data is used for what. | We have summarized the input data in a table. We considered the presence of punctual and widespread sources of pollutants in the vulnerability assessment, but we did not use a model to evaluate the actual propagation of contamination. We have included the data coming from the hydraulic model and included more information about the model in the Supplementary Materials. A column was added to the input data table indicating where the data sets are used in the methodology. |
| L 86-87: Is the risk assessed differently at a different scale than mesoscale? Why does this not hold for other scales? This sentence is confusing. | To avoid confusion, we have removed this sentence. However, the assessment of risk does differ in terms of the data used at different scales. For example, at the microscale, assessments are characterized by a high level of detail regarding the exposure so they would take individual assets at risk into account, such as buildings, vehicles or infrastructure. At the mesoscale, exposure information is usually based on aggregation, using a land use map to assess exposure, as is carried out in this paper. |
| Figure 1: Is there an arrow missing from the box "Depth, Velocity, Persistence"? Why do the "Land Use Map" and "People distribution map" have a different design? Also capitalization should be uniform. | This figure has been modified to better align with the text and to address the comments of the reviewer. |
| L 93-98 and Table 1: would fit better into 2.2.2 | The original Table 1 has now been moved to the Supplementary Material (Table S1). We have also shifted text around in the methods section to improve clarity. |
| L 100: "must be addressed" –> by whom or for what? | We have modified this to read: According to Article 6 of the 2007/60/CE Flood Directive (EU, 2007), when local authorities implement a Flood Risk Management Plan, three hazard scenarios must be considered: etc. |
| L 109: Which hydraulic model? | Details of the hydraulic and hydrological model have been added to the Supplementary Material. |
| Table 2: What does the sign "%" mean? Is this meant to be a "-"? (same in table 3) | This was meant to be a "-" sign and has been replaced in both tables. |
| L 140-142: "Greater awareness tends to correspond to greater preparation if an event takes place." Please add a source for this statement. | This sentence has been modified as follows: Some studies have found that if citizens have directly experienced a flood, their perception of flood |

| Comments from Reviewer 1 | Response to Comments |
|---|---|
|  | risk is higher (e.g., Thistlethwaite et al., 2018) although the factors that determine flood risk perception are varied. Moreover, the results from different studies can be ambiguous and/or contradictory (Lechowska, 2018). |
| Figure 3: Where do the weights come from? Are they assumed by the author or based on literature? What is the reasoning behind the weights? Why is "Insurance density" stated, when the weight is 0? What is the uncertainty for these weights and how do they affect the results? | We have added the following text: These values have been developed by the Provincia Autonoma di Trento (2006) from decades of experience with understanding exposure related to flood risk. Moreover, they have been tested and shown to be valid within AAWA. The insurance density is one of the components of risk spread in the guidelines from ISPRA (2012) and appears in Figure 3 for completeness. However, in the case of the Brenta-Bacchiglione catchment, the insurance density is 0.0 because insurance companies do not currently offer premiums to protect goods against flood damage. |
| Figure 4: What is the normalized index function and how was this value computed? What does it mean? | This is now Figure S1 in the Supplementary Material. This has been referred to as a value function to be consistent throughout the text. We also explain what value functions are. |
| Figures 4-6 could also be included in a supplementary material. In general section 2.2 is rather long and according to the abstract not central to this paper. Parts of this section could be included in a supplementary material. | Figures 4 to 6 have been added to the Supplementary Material. |
| L 217: You could add that people are also less likely to take warnings serious in future. | This statement has been added to the relevant section of the paper. |
| L 242: Why are road inundations lumped together, regardless of whether or not the infrastructure is damaged? Surely the cost differs significantly in those two scenarios? | Here the vulnerability is related to whether the road network can be used or not, i.e., at what stage cars become unstable during a flood. Hence, the vulnerability values are based on the estimation of the critical water height and velocity for the stability of vehicles during a flood (and have nothing to do with damage to the network infrastructure). The data have been derived from direct observation in laboratory experiments and from a report on the literature in this area (Reiter, 2000; Shand et al., 2011). In the estimation of the direct costs of the flood, the costs for damage to buildings, soil and infrastructure (including roads) assumes complete rebuilding or restoration and therefore is dealt with separately. |

| Comments from Reviewer 1 | Response to Comments |
|---|---|
| L 256: Does this mean that the vulnerability depends on the seasons, i.e., on when the crops are growing? | No, this refers to Corine Land Class 9: Unproductive land, which is defined as unproductive for the whole year. |
| L 260-266: I do not understand why "contamination/ pollution, erosion" and "open space" are lumped together here? Also apart from describing what an open space is, the connection to flood susceptibility unclear. Please make the connections clearer to the reader. | We have modified this section and focussed on contamination/pollution and erosion. |
| Why is the vulnerability high (i.e., 1) when "Integrated Pollution Prevention and Control (IPPC) installations" are present? Shouldn't these installations reduce the vulnerability? | This has been modified as follows: …the presence of relevant pollution sources was identified (Tables 1 and S1) and assigned a vulnerability of 1. |
| L 273-274: Please rephrase. I suppose "cultural heritage" is not actually considered an "adverse consequences of future flood events". | We have removed this phrase. |
| Figure 9: Why does water height not play a factor in (d)? | From experimentation (Citeau, 2003), the height of the water does not affect the vulnerability of natural and semi-natural environments up to a water height of 2.5m. |
| L 284: Repeat what the macro-categories are. | These macro-categories have been repeated although we have also modified these sections. |
| L 287: I would rephrase "moderate" as "low", as moderate is actually a synonym of medium. (same in Table 5) You actually use "low" already in Table 9 – please homogenize. | This has been modified in the text and in Table 5. |
| L 301: This is the first time you mention a reward. What reward do you mean? "Direct and indirect users" -> of what? | We have modified this section so this is no longer relevant. |
| L 308: One variable should not have multiple letters in an equation. This could be considered to be I*S*R*R. (Also applies to some other formulas in this paper.) | We do not completely agree with this comment. For example, evapotranspiration is commonly referred to as ET, potential evapotranspiration as PET and we can find many examples in the literature (including in HESS) of multiple letter variables, e.g. RMSE, NSE, etc., which are used very regularly in hydrology. |
| Table 6: Where do the weights come from? Are they calculated or taken from another source? | The weights in this table have been determined through expert consultation within the Alto-Adriatico Water Authority (AAWA), supported by the guidelines from ISPRA (2012). |
| L 330: superscript for 2 | This has been corrected. |
| L 349: Please define what you mean with a "unit of management". | A unit of management is part of terminology from the INSPIRE Directive but we have removed this to avoid any confusion. |

| Comments from Reviewer 1 | Response to Comments |
|---|---|
| L 358-359: "Because rapid floods are difficult to predict, early warning systems and prevention measures are of less use in this region." I have some doubts regarding this statement. Surely early warning systems (sirens) and prevention measures (reforestation) still reduce the risks associated with rapid floods? Please offer a source or rephrase. | We have modified this paragraph and removed this statement. |
| L 361: superscript for th | This has been corrected. |
| L 365: Was data that was not collected by trained volunteers discarded? How many volunteers were trained and how many were not trained? This section needs to be expanded considerably as described in the general comments. | A new section was added to the paper about the citizen observatory, which addresses these questions. Citizen data were not discarded but used to investigate their value in complementing hydrological modelling (Mazzoleni et al., 2017; 2018). |
| L 377: "The situation before the intervention" The word situation is rather vague, but more importantly you do not describe what exactly that intervention is (see general comments). | We have modified many headings to replace intervention with implementation of the CO on flood risk management. |
| L 379: Specify what you mean with numerical simulations. Do you mean the hydraulic model? Very little information is provided about the modelling that was done. The reference to section 2 could be more specific, do you mean section 2.2.1? | This has been modified to indicate numerical simulations from the hydraulic model, and the reader is then referred to the Supplementary Materials. |
| Table 8: What is P1, P2 and P3? | These have been replaced with the hazard classes of low, medium and high. |
| L 415: "implementation of the CO" –> not fully explained what that is, see general comments. | A new section was added to the paper about the citizen observatory. |
| Table 11: Please explain the difference to table 9 (not just the different values, but the different meaning). | The difference between the two tables is the amount of area flooded by risk class and flood hazard scenario before (Table 9) and after (Table 11) implementation of the CO. The captions have been modified to clarify this point and additional text has been added. |
| L 426-432: This paragraph is rather unclear, but this will likely improve once the general comments are addressed. | We removed this paragraph but provided more details on how the risk changes with and without the implementation of the CO in the methodology. |
| Table 12: caption "different" not "difference"; also I would not just include the residual damage, but also the original and then the difference. Or you could make a plot with both the original and residual costs. | This caption has been corrected. The original costs and the differences have been added. |

| Comments from Reviewer 1 | Response to Comments |
|---|---|
| L 436: How did you calculate the cost for CO? What are the cost components? Is it sensor costs, maintenance costs or personnel costs? Over what time do these costs accumulate? | Table S2 has been added to the Supplementary Material, which is a breakdown of the costs of the CO. |
| I have found at least one reference in the text that is not in the list of references. Please double check. | All references were checked to ensure that they are in the reference list. |

---

## Author Comment (AC2) · 13 May 2020

We would like to thank the reviewer for the very detailed and insightful comments to the first draft of the paper, which have helped to greatly improve it. We agree that many details needed further clarification. We hope that this improved version is easier to understand and makes the key points in a clearer fashion. We have add the comments and responses in table form as an attached PDF, which may be easier to read.

GENERAL COMMENTS COMMENT: This paper presents a framework that combines

modelling with data collected through a citizen observatory running in northern Italy. The idea that citizen science could successfully be used to reduce the risk of catchments to the effects of flooding is interesting and timely. Having said that, I find several shortcomings in this paper that need to be fixed before it can be considered for publication in HESS. These shortcomings are: 1. The core message of the paper for me should be that citizen science and modelling are effective in reducing risk. However, the paper describes extensively the modelling approach whereas the citizen science part is very vaguely described. The two together don't make a convincing story. RESPONSE: We have strengthened the core message of the paper, i.e., that citizen science and modelling are effective in reducing risk. In particular, we have added a section describing the citizen observatory in more detail. This should hopefully provide the necessary context for making this story more convincing.

COMMENT: 2. The concept of risk used in this paper is a known one. The paper uses quite some space in the methods section to go through the component of risk but it does it in a very confusing way. For example, one would expect that Fig 1 is used through the methods to arrive to the risk estimates, but it isn't, and therefore the presentation of the method is muddled. I would suggest re-writing the risk section, shortening and focusing it on the application to flooding risk, using a comprehensive figure to guide the reader. RESPONSE: The introductory section on risk has been removed and flood risk has been discussed in the introduction as per your more specific comment below. A new Figure 1 has been added and better aligned to the description of the methodology.

COMMENT: 3. The modelling approach uses many coefficients that lead to the estimation of risk. These coefficients presented in several figures, were taken apparently from a number of sources (not always disclosed) and are not subject to a thorough sensitivity analysis. The results of the modelling are heavily determined by the coefficients adopted so it is critically to explain these very well. I don't recommend this additional explanation is included in the methods section, but it has to be properly documented

in Supplementary material. Without this, it will be extremely hard to test and to apply this method elsewhere. RESPONSE: We have now specified the source of all of the coefficients, weights and value functions used in the methodology. They are based on existing literature, expert consultation and the guidelines on flood risk estimation published by ISPRA (2012). Some of these values are based on years of experience (e.g., exposure by land use type) and have been internally validated.

COMMENT: 4. The description of the Citizen Observatory is the most disappointing. It doesn't inform the reader in terms of data collected, how did the work, how it was implemented, etc. This should be lot more prominent in this paper. I do not recommend the publication of this paper because it requires significant re-writing. RESPONSE: We have thoroughly revised the description of the citizen observatory, moving this to a separate section 2.

SPECIFIC COMMENTS COMMENT: The abstract is not informative, and it should reflect the key innovation of this study. Presumably, the successful linking of risk modelling and citizen science should be the key message in the abstract. RESPONSE: The abstract has been rewritten to reflect the main message, i.e., potential reduction in risk possible when linking citizen science with modelling.

COMMENT: For the case study, the findings seem misleading because the study does not cost a infra-structure adaptive intervention, only the roughly estimated costs of the potential damage. RESPONSE: In the results section, the cost of constructing a retention basin in the municipalities of Sandrigo and Breganze is provided.

COMMENT: Introduction L20-30 This section is irrelevant for the story of the paper, and not well written. I suggest to delete it, and add instead a clear definition of risk specific to flooding that introduce the paper. RESPONSE: These lines have been removed and the introduction rewritten to include a clear definition of risk specific to flooding.

COMMENT: L37-39 References required for the statement 'exponential growth' in citizen science. RESPONSE: This statement was modified from exponential growth to

the rise in citizen science and crowdsourcing, and references were added.

COMMENT: L40 unclear why references were added after . . ."Among the various form of citizen science". Instead of references I would expect a list of the different forms. This whole sentence needs re-writing. RESPONSE: We agree with the reviewer that this was confusing. We have removed this and replaced it with a simpler statement and a reference.

COMMENT: L61-63 and L65-68 RESPONSE: There were no comments provided with these line numbers. Please clarify if there are specific comments to address.

COMMENT: L70 section 2.1 needs to be more carefully described. Details needed to interpret results. RESPONSE: More details have been added on the input data including a table.

COMMENT: L74 section 2.2 See general comment. This section up to 2.4 is so poorly written that it is hard to keep track of the method used, sources of information and assumptions made. In addition, the calculation of risk must be done from the beginning with a focus on flooding risk, the aim of this study. RESPONSE: We have modified these sections and also shifted some material to the Supplementary Materials.

COMMENT: Fig. 1 is not self-explanatory and it is not connected properly to the text that follows in 2.2 and 2.3. RESPONSE: Figure 1 has been modified and better connected to the text.

COMMENT: Table 1 could be sent to supplementary material. It is not critical to the results. RESPONSE: We have moved this to the Supplementary Material.

COMMENT: L100-110. A description and testing of the hydrological model are needed because the reference included is not a peer-reviewed source and can't be accessed by the reader. This could be added to supplementary material RESPONSE: We have added a description of the hydrological hydraulic model to the Supplementary Material.

COMMENT: Table 2, Table 3 and Table 4. It should be clear what the sources for these

coefficients are and why these are accepted to be reasonable without performing a sensitivity analysis. RESPONSE: The weights in Table 2 have been developed through stakeholder consultation and guided by flood risk assessment in ISPRA (2012). The weights in Table 3 have been developed over decades of experiences with exposure by the province of Trento (with the reference added). The weights in Table 4 are from a UK DEFRA study and are cited in the guidance on flood risk provided by ISPRA (2012). These sources have been added to the paper.

COMMENT: L174 why is the use of 'value functions' the preferred approach, and what is the uncertainty associated with them? I don't see an uncertainty analysis conducted here. RESPONSE: We have added more information about what and why this approach is used and removed the text regarding the method being the preferred approach. The value functions have been derived through extensive expert and stakeholder consultation. An uncertainty analysis has not been conducted but we mention this in the discussion section of the paper.

COMMENT: Fig. 3 I wonder why this figure is presented in addition to Fig 1, and using slightly different terms and approach? RESPONSE: Figure 1 has now been updated to better align with the text in the paper and to be more consistent with Figure 3.

COMMENT: Fig. 4, 5, 6, 7 and 8. What is the uncertainty associated with this coefficients? RESPONSE: The coefficients in Figures 4 to 6 (now Figures S1 to S3 in the Supplementary Material) have been determined through expert consultation (at the provincial level) and stakeholders at AAWA. Therefore, they represent a consensus view. In fact, the reason for using expert consultation is because of uncertainty. We have added a paragraph to the Discussion and Conclusions section to discuss this aspect of the paper. The coefficients for Figures 7 and 8 (now placed in the Supplementary Materials) are based on laboratory experiments and sources are provided in the text. Although we agree that there will be uncertainties around these figures, the final vulnerability coefficients have been further agreed upon through expert consultation at AAWA and represent conservative estimates.

COMMENT: L214 Are forecasting systems the same as 'early warning systems' of Fig. 3? This is confusing. RESPONSE: We now use early warning system to be consistent.

COMMENT: Fig. 9 It is very hard to understand this figure. The caption is not self-explanatory. RESPONSE: We have corrected water depth to water height to be consistent with the graphs. For each land use type, there are two vulnerability values. Figure 9a for vineyards indicates that at a water height of less than 0.5m and a flow velocity of less than 0.25 m/s, the vulnerability is 0.5. Values greater than 0.5 m and 0.25 m/s have a vulnerability of 1.0.

COMMENT: L284. The component of the equation should be explained. This equation should be introduced after Fig 1 when the concepts are explained. RESPONSE: The individual components of the equation are now explained but these rely on explanations that are contained in the hazard, exposure and vulnerability sections so the equation has been kept in the same location.

COMMENT: L295. This section on C/B analysis is not clear at all. I would have expected that the costs would be the cost of remedial and/or preventive actions, which are not clearly explained here. What are the units of ISRR? I would guess hectares of km2. And of CBA? RESPONSE: The ISRR is a unitless index. If positive, it means that there has been an overall reduction in the risk due to the implementation of the CO. If negative, then the risk has increased. In this example, the ISRR is 2.5 so the overall risk has been reduced. The CBA equation has been removed as the damage compared to the avoided damage provides a monetary assessment of the benefits.

COMMENT: Table 7. I would expect large variability in these values. No uncertainty analysis performed. RESPONSE: These figures come from a study by Huizinga (2007) from the Joint Research Center (JRC) in Italy. In 2017, Huizinga et al. published a report on global flood depth damage functions, comparing the results in 2017 with those in 2007. The overall patterns matched the 2017 values but showed overestimates in Europe, which were corrected by assuming a 40% inalterable portion for European

buildings. The numbers then matched well. Hence some uncertainty analysis has been performed by the original authors of the figures. We would also assume they are conservative, having been published in 2007.

COMMENT: L348 section 2.4. This should be one of the key section of the paper, but it is unfortunately very vague and doesn't provide the reader much information on how the citizen observatory worked, data collected, for how long etc. RESPONSE: We agree with the reviewer. As the paper has now been revised substantially, we hope there is more clarity.

COMMENT: Results. In view of all the methodological questions, it seems pointless to go through the results. From the paragraph included in L426-432, it seems that the paper should have explained the simulations of risk and damage, and what the citizen observatory programme did and achieved, which here remains as a black box. RESPONSE: We have added a section explaining the citizen observatory and changed the headings to more clearly show that the risk calculations have been undertaken with and without the implementation of the citizen observatory.

Please also note the supplement to this comment:
https://www.hydrol-earth-syst-sci-discuss.net/hess-2019-627/hess-2019-627-AC2-supplement.pdf

**Supplement:**

**Response to Reviewer 2**

We would like to thank the reviewer for the very detailed and insightful comments to the first draft of the paper, which have helped to greatly improve it. We agree that many details needed further clarification. We hope that this improved version is easier to understand and makes the key points in a clearer fashion.

| Comments from Reviewer 2 | Response |
|---|---|
| This paper presents a framework that combines modelling with data collected through a citizen observatory running in northern Italy. The idea that citizen science could successfully be used to reduce the risk of catchments to the effects of flooding is interesting and timely. Having said that, I find several shortcomings in this paper that need to be fixed before it can be considered for publication in HESS. These shortcomings are: 1. The core message of the paper for me should be that citizen science and modelling are effective in reducing risk. However, the paper describes extensively the modelling approach whereas the citizen science part is very vaguely described. The two together don't make a convincing story. | We have strengthened the core message of the paper, i.e., that citizen science and modelling are effective in reducing risk. In particular, we have added a section describing the citizen observatory in more detail. This should hopefully provide the necessary context for making this story more convincing. |
| 2. The concept of risk used in this paper is a known one. The paper uses quite some space in the methods section to go through the component of risk but it does it in a very confusing way. For example, one would expect that Fig 1 is used through the methods to arrive to the risk estimates, but it isn't, and therefore the presentation of the method is muddled. I would suggest re-writing the risk section, shortening and focusing it on the application to flooding risk, using a comprehensive figure to guide the reader. | The introductory section on risk has been removed and flood risk has been discussed in the introduction as per your more specific comment below. A new Figure 1 has been added and better aligned to the description of the methodology. |
| 3. The modelling approach uses many coefficients that lead to the estimation of risk. These coefficients presented in several figures, were taken apparently from a number of sources (not always disclosed) and are not subject to a thorough sensitivity analysis. The results of the modelling are heavily determined by the coefficients adopted so it is critically to explain these very well. I don't recommend this additional explanation is included in the methods section, but it has to be properly documented in Supplementary material. Without this, it will be extremely hard to test and to apply this method elsewhere. | We have now specified the source of all of the coefficients, weights and value functions used in the methodology. They are based on existing literature, expert consultation and the guidelines on flood risk estimation published by ISPRA (2012). Some of these values are based on years of experience (e.g., exposure by land use type) and have been internally validated. |

| Comments from Reviewer 2 | Response |
|---|---|
| 4. The description of the Citizen Observatory is the most disappointing. It doesn't inform the reader in terms of data collected, how did the work, how it was implemented, etc. This should be lot more prominent in this paper. I do not recommend the publication of this paper because it requires significant re-writing. | We have thoroughly revised the description of the citizen observatory, moving this to a separate section 2. |
| Specific comments. The **abstract** is not informative, and it should reflect the key innovation of this study. Presumably, the successful linking of risk modelling and citizen science should be the key message in the abstract. | The abstract has been rewritten to reflect the main message, i.e., potential reduction in risk possible when linking citizen science with modelling. |
| For the case study, the findings seem misleading because the study does not cost a infra-structure adaptive intervention, only the roughly estimated costs of the potential damage. | In the results section, the cost of constructing a retention basin in the municipalities of Sandrigo and Breganze is provided. |
| Introduction L20-30 This section is irrelevant for the story of the paper, and not well written. I suggest to delete it, and add instead a clear definition of risk specific to flooding that introduce the paper. | These lines have been removed and the introduction rewritten to include a clear definition of risk specific to flooding. |
| L37-39 References required for the statement 'exponential growth' in citizen science. | This statement was modified from exponential growth to the rise in citizen science and crowdsourcing, and references were added. |
| L40 unclear why references were added after . . ."Among the various form of citizen science". Instead of references I would expect a list of the different forms. This whole sentence needs re-writing. | We agree with the reviewer that this was confusing. We have removed this and replaced it with a simpler statement and a reference. |
| L61-63 and L65-68 | There were no comments provided with these line numbers. Please clarify if there are specific comments to address. |
| L70 section 2.1 needs to be more carefully described. Details needed to interpret results. | More details have been added on the input data including a table. |
| L74 section 2.2 See general comment. This section up to 2.4 is so poorly written that it is hard to keep track of the method used, sources of information and assumptions made. In addition, the calculation of risk must be done from the beginning with a focus on flooding risk, the aim of this study. | We have modified these sections and also shifted some material to the Supplementary Materials. |
| Fig. 1 is not self-explanatory and it is not connected properly to the text that follows in 2.2 and 2.3. | Figure 1 has been modified and better connected to the text. |

| Comments from Reviewer 2 | Response |
|---|---|
| Table 1 could be sent to supplementary material. It is not critical to the results. | We have moved this to the Supplementary Material. |
| L100-110. A description and testing of the hydrological model are needed because the reference included is not a peer-reviewed source and can't be accessed by the reader. This could be added to supplementary material | We have added a description of the hydrological hydraulic model to the Supplementary Material. |
| Table 2, Table 3 and Table 4. It should be clear what the sources for these coefficients are and why these are accepted to be reasonable without performing a sensitivity analysis. | The weights in Table 2 have been developed through stakeholder consultation and guided by flood risk assessment in ISPRA (2012). The weights in Table 3 have been developed over decades of experiences with exposure by the province of Trento (with the reference added). The weights in Table 4 are from a UK DEFRA study and are cited in the guidance on flood risk provided by ISPRA (2012). These sources have been added to the paper. |
| L174 why is the use of 'value functions' the preferred approach, and what is the uncertainty associated with them? I don't see an uncertainty analysis conducted here. | We have added more information about what and why this approach is used and removed the text regarding the method being the preferred approach. The value functions have been derived through extensive expert and stakeholder consultation. An uncertainty analysis has not been conducted but we mention this in the discussion section of the paper. |
| Fig. 3 I wonder why this figure is presented in addition to Fig 1, and using slightly different terms and approach? | Figure 1 has now been updated to better align with the text in the paper and to be more consistent with Figure 3. |
| Fig. 4, 5, 6, 7 and 8. What is the uncertainty associated with this coefficients? | The coefficients in Figures 4 to 6 (now Figures S1 to S3 in the Supplementary Material) have been determined through expert consultation (at the provincial level) and stakeholders at AAWA. Therefore, they represent a consensus view. In fact, the reason for using expert consultation is because of uncertainty. We have added a paragraph to the Discussion and Conclusions section to discuss this aspect of the paper. The coefficients for Figures 7 and 8 (now placed in the Supplementary Materials) are based on laboratory experiments and sources are provided in the text. Although we agree that there will be uncertainties around these figures, the final vulnerability coefficients have been further agreed upon through expert consultation at AAWA and represent conservative estimates. |

| Comments from Reviewer 2 | Response |
|---|---|
| L214 Are forecasting systems the same as 'early warning systems' of Fig. 3? This is confusing. | We now use early warning system to be consistent. |
| Fig. 9 It is very hard to understand this figure. The caption is not self-explanatory. | We have corrected water depth to water height to be consistent with the graphs. For each land use type, there are two vulnerability values. Figure 9a for vineyards indicates that at a water height of less than 0.5m and a flow velocity of less than 0.25 m/s, the vulnerability is 0.5. Values greater than 0.5 m and 0.25 m/s have a vulnerability of 1.0. |
| L284. The component of the equation should be explained. This equation should be introduced after Fig 1 when the concepts are explained. | The individual components of the equation are now explained but these rely on explanations that are contained in the hazard, exposure and vulnerability sections so the equation has been kept in the same location. |
| L295. This section on C/B analysis is not clear at all. I would have expected that the costs would be the cost of remedial and/or preventive actions, which are not clearly explained here. What are the units of ISRR? I would guess hectares of km2. And of CBA? | The ISRR is a unitless index. If positive, it means that there has been an overall reduction in the risk due to the implementation of the CO. If negative, then the risk has increased. In this example, the ISRR is 2.5 so the overall risk has been reduced. The CBA equation has been removed as the damage compared to the avoided damage provides a monetary assessment of the benefits. |
| Table 7. I would expect large variability in these values. No uncertainty analysis performed. | These figures come from a study by Huizinga (2007) from the Joint Research Center (JRC) in Italy. In 2017, Huizinga et al. published a report on global flood depth damage functions, comparing the results in 2017 with those in 2007. The overall patterns matched the 2017 values but showed overestimates in Europe, which were corrected by assuming a 40% inalterable portion for European buildings. The numbers then matched well. Hence some uncertainty analysis has been performed by the original authors of the figures. We would also assume they are conservative, having been published in 2007. |
| L348 section 2.4. This should be one of the key section of the paper, but it is unfortunately very vague and doesn't provide the reader much information on how the citizen observatory worked, data collected, for how long etc. | We agree with the reviewer. As the paper has now been revised substantially, we hope there is more clarity. |
| Results. In view of all the methodological questions, it seems pointless to go through the results. From the paragraph included in L426-432, it seems that the paper should have explained the simulations of risk and damage, and | We have added a section explaining the citizen observatory and changed the headings to more clearly show that the risk calculations have been undertaken with and without the implementation of the citizen observatory. |

| Comments from Reviewer 2 | Response |
|---|---|
| what the citizen observatory programme did and achieved, which here remains as a black box. | |

---

## Author Comment (AC3) · 13 May 2020

**The Value of Citizen Science for Flood Risk Reduction: Cost-benefit Analysis of a Citizen Observatory in the Brenta-Bacchiglione Catchment**

Michele Ferri[1], Uta Wehn[2], Linda See[3], Martina Monego[1], Steffen Fritz[3]

[1]Alto-Adriatico Water Authority (AAWA), Cannaregio 4314, 30121 Venice, Italy
[2]IHE Delft Institute for Water Education, Westvest 7, 2611 AX Delft, The Netherlands
[3]International Institute for Applied Systems Analysis (IIASA), Schlossplatz 1, 2361 Laxenburg, Austria

*Correspondence to*: Michele Ferri (michele.ferri@distrettoalpiorientali.it)

**Abstract.** ~~Citizen observatories are a relatively recent form of citizen science, which involve citizens in making environmental observations over a period of time. These observations can help to inform the decision making of local authorities and other stakeholders, creating a platform for two way interaction between citizens and public agencies. Although citizen observatories can clearly generate many different benefits, they also have an associated cost. There are currently no examples of quantifying the costs and benefits of citizen observatories in the literature, yet this type of analysis is critical if there is to be real uptake of citizen observatories by public agencies more generally. This paper presents and applies a generic methodology for capturing the value of a citizen observatory for flood risk reduction in the Brenta-Bacchiglione catchment using a cost-benefit analysis. The results show that the benefits of implementing a citizen observatory approach outweigh the costs by approximately 2 to 1 and can reduce the annual expected damage to a greater degree than a much more costly structural approach~~Citizen observatories are a relatively recent form of citizen science. As part of the flood risk management strategy of the Brenta-Bacchiglione catchment, a citizen observatory for flood risk management has been proposed and is currently being implemented. Citizens are involved through monitoring water levels and obstructions and providing other relevant information through mobile apps, where the data are assimilated with other sensor data in a hydrological-hydraulic model used in early warning. A cost benefit analysis of the citizen observatory was undertaken to demonstrate the value of this approach in monetary terms. Although not yet fully operational, the citizen observatory is assumed to decrease the social vulnerability of the flood risk. By calculating the hazard, exposure and vulnerability of three flood scenarios (required for flood risk management planning by the EU Directive on Flood Risk Management) with and without the proposed citizen observatory, it is possible to evaluate the benefits in terms of the average annual avoided damage costs. Although currently a hypothetical exercise, the results showed a reduction in avoided damage of 45% compared to a business as usual scenario. Thus, linking citizen science with hydrological modelling, and to raise awareness of flood hazards, has great potential in reducing flood risk in the Brenta-Bacchiglione catchment in the future. Moreover, such approaches are easily transferable to other catchments.

**1 Introduction**

In 2018, flooding affected the highest number of people of any natural disaster globally and caused major damage worldwide (CRED, 2019). With climate change, the frequency and magnitude of extreme events will increase, leading to a higher risk of flooding (Schiermeier, 2011). This risk will be further exacerbated by future economic and population growth (Tanoue et al., 2016). Thus, managing flood risk is critical for reducing future negative impacts. Flood risk assessments are undertaken by the insurance industry for determining properties at high risk (Hsu et al., 2011), but they are also a national requirement in the European Union as set out in the EU Flood Risk Management Directive, which requires that flood risk management plans are produced for each river basin (EU, 2007; Müller, 2013). The assessment of flood risk involves quantifying three main drivers (National Research Council, 2015): (a) flood hazard, which is the probability that a flood of a certain magnitude

40  will occur in a certain period of time in a given area; (b) exposure, which is the economic value of the human lives and assets affected by the flood hazard; and (c) vulnerability, which is the degree to which different elements (i.e., people, buildings, infrastructure, economic activities, etc.) will suffer damage associated with the flood hazard. In addition, flood risk can be mitigated through hard engineering strategies such as implementation of structural flood protection schemes, soft engineering approaches comprising more natural methods of flood management (Levy and Hall, 2005), and community-

45  based flood risk management (Smith et al., 2017). As part of requirements in the EU Flood Risk Management Directive, any mitigation actions must be accompanied by a cost-benefit analysis.

Flood hazard is generally determined through hydrological and hydraulic modelling. Hence accurate predictions are critical for effective flood risk management, particularly in densely populated urban areas (Mazzoleni et al., 2017). The input data required for modelling are often incomplete in terms of resolution and density (Lanfranchi et al., 2014), which translates

50  into variable accuracy in flood predictions (Werner et al., 2005). New sources of data are becoming available to support flood risk management. For example, the rise of citizen science and crowdsourcing (Howe, 2006; Sheldon and Ashcroft, 2016), accelerated by the rapid diffusion of information and communication technologies, is providing additional, complementary sources of data for hydrological monitoring (Njue et al., 2019). Citizen science refers to the involvement of the public in any step of the scientific method (Shirk et al., 2012). However, one of the most common forms of participation

55  is in data collection (Njue et al., 2019). Citizen observatories (CO) are a particular form of citizen science in so far as they constitute the means not just for new knowledge creation but also for its application, which is why they are typically set up with linkages to specific policy domains (Wehn et al., 2019). COs must therefore include a public authority (e.g., a local, regional or national body) to enable two-way communication between citizens and the authorities to create a new source of high quality, authoritative data for decision making and for the benefit of society. Moreover, COs involve citizens in

60  environmental observations over an extended period of time of typically months and years (rather than one-off exercises such as data collection 'Blitzes'), and hence contribute to improved temporal resolution of the data, using dedicated apps, easy-to-use physical sensors and other monitoring technologies linked to a dedicated platform (Liu et al., 2014; Mazumdar et al., 2016). COs are increasingly being used in hydrology/water sciences and management and in various stages of the flood risk management cycle, as reviewed and reported by Assumpção (2018), Etter et al. (2018), Mazzoleni et al. (2017), Buytaert

65  et al. (2014), Wehn and Evers (2015) and Wehn et al. (2015). These studies found that the characteristic links of COs to authorities and policy do not automatically translate into higher levels of participation in flood risk management, nor that communication between stakeholders improves; rather, changes towards fundamentally more involved citizen roles with higher impact in flood risk management take years to evolve (Wehn et al., 2015).

The promising potential of the contribution of COs to improved flood risk management is paralleled by limited evidence

70  of their actual impacts and added value. Efforts are ongoing such as the consolidation of evaluation methods and empirical evidence by the H2020 project WeObserve[1] Community of Practice on the value and impact of citizen science and COs, and the development and application of methods for measuring the impacts of citizen science by the H2020 project MICS[2]. To date, the societal and science-related impacts have received most attention, while the focus on economic impacts, costs and benefits has been both more limited and more recent (Wehn et al., 2020a). The studies that do focus on economic impacts

75  related to citizen science (rather than citizen observatories) propose to consider the time invested by researchers in engaging and training citizens (Thornhill et al., 2016); to relate cost and participant performance for hydrometric observations in order to estimate the cost per observation (Davids et al., 2019); to estimate the costs as data-related costs, staff costs and other costs; and the benefits in terms of scientific benefits, public engagement benefits and the benefits of strengthened capacity of participants (Blaney et al., 2016); and to compare citizen science data and in-situ data (Goldstein et al., 2014; Hadj-Hammou

80  et al., 2017). Wehn et al. (2020b) assessed the value of COs from a data perspective and a cost perspective, respectively, to
* * *
[1] https://www.weobserve.eu/
[2] https://mics.tools/

qualify the degree of complementarity that the data collected by citizens offers to in-situ networks and to quantify the relation between the investments required to set up a CO and the actual amount of data collected. Based on a comparison of four COs, they suggest that setting up a CO for the sole purpose of data collection appears to be an expensive undertaking (for the public sector organization(s) benefitting from the respective CO) since, depending on the process of (co)designing the CO, it may not necessarily complement the existing in-situ monitoring network (with the likely exception of infrastructure-weak areas in developing countries).

Overall, there is a lack of available, appropriate and peer-reviewed evaluation methods and of evidence of the added value of COs, which is holding back the uptake and adoption of COs by policy makers and practitioners. In this paper, we take a different approach to previous studies by using a more conventional cost-benefit analysis framework to assess the implementation of a citizen observatory on flood risk management in the Brenta-Bacchiglione catchment in northern Italy. The purpose of a cost-benefit analysis is to compare the effectiveness of different alternative actions, where these actions can be public policies, projects or regulations that can be used to solve a specific problem. We treat the citizen observatory in the same way as any other flood mitigation action for which a cost-benefit analysis would be undertaken in this catchment. Although the citizen observatory is still being implemented, the assumptions for the cost-benefit analysis are based on primary empirical evidence from a CO pilot that was undertaken by the WeSenseIt project in the town of Vicenza, Italy, described in more detail in section 2.1 and now extended to the wider catchment (sections 2.2 and 2.3). In section 3 we present the flood risk and cost benefit methodology followed by the results in section 4. Conclusions, limitations of the methodology and case-specific insights are provided in section 5.

**2 The Development of a Citizen Observatory for Flood Risk Management**

**2.1 The WeSenseIt Project**

Through the WeSenseIt research project (www.wesenseit.eu), funded under the 7th framework program (FP7-ENV-2012 n° 308429), a CO for flood risk was developed with the Upper Adriatic Basin Authority in northern Italy. The objective of this CO was to collect citizen observations from the field, and to obtain a broader and more rapid picture of developments before and during a flood event. The CO involved many stakeholders concerned with the management and use of the water resources, and with water-related hazards in the Bacchiglione River basin. The main actors included the local municipalities, the regional and local civil protection agencies, environment agencies and the irrigation authorities. The Alto Adriatico Water Authority (AAWA) facilitated access to a highly trained group of citizen observers, namely civil protection volunteers, who undertook the observations (i.e., using staff gauges with a QR code to measure the water level and reporting water way obstructions) as part of their volunteer activities. Additional volunteers were also recruited during the project from the Italian Red Cross, the National Alpine Trooper Association, the Italian Army Police and other civil protection groups, with more than 200 volunteers taking part in the CO pilot. Training courses for the volunteers were organized to disseminate and explain the use of a smartphone application and an e-collaboration platform, which were developed as part of the WeSenseIt project. In addition to the low cost sensing equipment, the CO also used data from physical sensors: 3 sonar sensors (river water level), 4 weather stations (wind velocity and direction, precipitation, air temperature and humidity) and 5 soil moisture sensors. The combined visualization of the sensors (including existing sensors from the Venice Environment Agency) was available in the online e-collaboration platform. During the WeSenseIt project, research into the value of crowdsourced data for hydrological modelling was investigated (Mazzoleni et al., 2017, 2018) and found to complement traditional sensor networks.

This pilot was later adopted by the European Community as a "good practice" example of the application of Directive 2007/60/EC. After the positive experience in WeSenseIt, funds were made available to develop a CO for flood risk management at the district scale, covering the larger Brenta-Bacchiglione catchment. At this stage, a cost-benefit analysis

was undertaken, which is reported in this paper. The next section provides details of the Brenta-Bacchiglione catchment followed by ongoing developments in the CO for flood risk management.

**2.2 The Brenta-Bacchiglione Catchment**

125 The Brenta-Bacchiglione River catchment includes the Retrone and Astichiello Rivers, and falls within the Veneto Region in Northern Italy, which includes the cities of Padua and Vicenza (Figure 1). The catchment is surrounded by the Beric hills in the south and the Prealpi in the northwest. In this mountainous area, rapid or flash floods occur regularly and are difficult to predict. For the past three years, extreme weather events (including flooding) are the top risk in terms of likelihood and among the top three risks in terms of impact, where this combination makes it the top risk in 2019 (WEF, 2017, 2018, 2019).

130 Between 1995-2015, flooding alone accounted for 47% of all weather related disasters, affecting 2.3 billion people globally (CRED and UNISDR, 2015). Continuing an upward trend, financial losses in 2017 due to global weather related disasters exceeded US$300 billion (Swiss Re, 2017). Hurricane Harvey, in particular, caused US$125 billion damage in 2017, led to the death of 88 people, destroyed more than 12,700 homes and resulted in a rise in gas prices due to the impacts on oil production (Amadeo, 2019). However, economic losses can go well beyond damage to infrastructure and assets, e.g.,

135 disruption to businesses and supply chains can equal or exceed the costs of infrastructure damage (Hallegatte, 2008; Jongman, 2018). Moreover, developing countries and small island states affected by tropical storms are likely to suffer greater losses. In 2017, Hurricane Maria caused an estimated total damage and loss of US$1.3 billion to Dominica while US$5.4 billion in damage and loss was estimated for other islands due to the combined effects of Hurricanes Maria and Irma (Asariotis, 2018).

140 Accurate predictions are crucial for flood risk management (FRM), e.g., to control river structures and water levels, in order to reduce risks and damages from flooding, particularly in densely populated urban areas (Mazzoleni et al., 2017b). However, weather patterns are local in nature, not easily captured or predicted by existing in-situ and remote sensing based modelling approaches, and are likely to be intensified by climate change (Pachauri et al., 2014; Tol, 2014). The data acquired using these methods are often incomplete in terms of resolution and density (Lanfranchi et al., 2014). This translates into

145 variable accuracy in flood predictions (Werner et al., 2005).

The recent exponential growth in citizen science and crowdsourcing approaches, accelerated by the rapid diffusion of information and communication technologies, is providing additional, complementary sources of data for hydrological and hydraulic models. Citizen science refers to the involvement of the public in any step of the scientific method (Shirk et al., 2012). Among the various forms of citizen science (Cooper et al., 2007; Bonney et al., 2009;

150 Shirk et al., 2012), contributory forms are of particular interest here, focusing on the observations that citizens can contribute (as opposed to their collaboration in the entire research process or the co-design of the research). Citizen observatories (CO) are a particular form of citizen science in so far as they involve citizens in environmental observations over an extended period of time (rather than one-off exercises such as data collection 'Blitzes'), and hence contribute to improved temporal resolution of the data, using dedicated apps, easy-to-use physical sensors and

155 other monitoring technologies linked to a dedicated platform (Liu et al., 2014; Mazumdar et al., 2016). COs must also Rapid floods generally affect the towns of Torri di Quartesolo, Longare and Montegaldella, although there is also widespread flooding in the cities of Vicenza and Padua, which includes industrial areas and areas of cultural heritage. For example, in 2010, a major flood affected 130 communities and 20,000 individuals in the Veneto region. The city of Vicenza was one of the most affected municipalities, with 20% of the metropolitan area flooded.

160

[Figure]

**Figure 1: Location of the Brenta-Bacchiglione catchment and its urban communities.**

**2.3 The Citizen Observatory for Flood Risk Management for the Brenta-Bacchiglione Catchment**

The CO for flood risk management, which is currently being implemented, was included in the prevention measures of the Flood Risk Management Plan (PGRA) for the Brenta-Bacchiglione catchment. The purpose of the CO is to strengthen communication channels before and during flood events in accordance with the EU Flood Directive on Flood Risk Management, to increase the resilience of the local communities and to address residual risk. Building on the WeSenseIt experience, an IT platform to aid decision support during the emergency phases of a flood event is being implemented. This platform will integrate information from the hydrological model, which is equipped with a data assimilation module that integrates the crowdsourced data collected by citizens and trained experts with official sensor data. A mobile app for data collection based on the WeSenseIt project is under development. The platform and mobile technology will guarantee user traceability and facilitate two-way communication between the authorities, the citizens and the operators in the field, thereby significantly increasing the effectiveness of civil protection operations during all phases of an emergency. The fully operational CO will include 64 additional staff gauges equipped with a QR code (58 to measure water level and 6 for snow height), 12 sonar sensors and 8 weather stations.

To engage and maintain the involvement of "expert" CO participants (i.e., civil protection volunteers, technicians belonging to professional associations, members of environmental associations), a set of training courses will be run. The involvement of technicians (formalized in November 2018 with an agreement between the respective associations and AAWA) offers an important opportunity to use the specific knowledge and expertise of these technicians to better understand the dynamics of flood events and to acquire high quality data to feed the models and databases. When an extreme event (i.e., heavy rain) is forecast, AAWA will call upon any available technicians in providing data (with a reimbursement of 75 €/day (including insurance costs) and a minimum activity per day of 3 hours). There are currently 41 technicians involved in the CO, which includes civil/hydraulic/geotechnical engineers, agronomists and forestry graduates. Participants must attend two training sessions followed by a final examination. To give an example of the valuable information that the expert CO participants can provide, AAWA called upon technicians during two heavy rainfall events (November 2019; 5 days). These technicians collected relevant data on the status of the rivers including the vegetation, the water levels, the status of bridges and levees, collecting 1660 images and completing 700 status reports.

To engage citizens, a different approach is being taken. Within the 120 municipalities currently in high flood risk zones, engagement of schools is currently ongoing, including the development of educational programs for teachers. The aim is to raise student awareness of existing flood risks in their own area, and to help students recognize the value of the CO (and the

mobile technology) in protecting their families, e.g., using the app to send important information about flooding, which then contributes to everyone's safety. This component of the CO involves 348 primary schools and 340 middle and secondary schools. The three universities in the area will also be involved through conferences and webinars. Communication through the CO website, via social media campaigns, radio broadcasts and regional newspapers will be used to engage and maintain citizen involvement in the CO. This communication plan, which will continue over the next five years, has the ambitious goal of involving 75,000 people in the CO to download the app and contribute observations.

**3 Methodology**

The methodology consists of three steps: (i) mapping of the flood risk (section 3.1); (ii) quantification of the flood risk reduction (section 3.2); and (iii) calculation of the damage from flooding under three flood scenarios (section 3.3), all of which consider the flood risk with and without the implementation of the CO on flood risk management.

**3.1 Flood risk mapping**

[revised manuscript text omitted]

The final component is vulnerability, which has a physical and social dimension. Physical vulnerability is defined as the susceptibility of an exposed element such as people or buildings to flooding (Balbi et al., 2012) and is calculated using the

same three macro-categories as that of exposure, i.e., the population affected, the economic activities affected, and the environmental and cultural-archaeological assets affected. Within the people affected category, we also consider social vulnerability. This refers to the perception or awareness that an adverse event may occur. Some studies have found that if citizens have directly experienced a flood, their perception of flood risk is higher (e.g., Thistlethwaite et al., 2018) although the factors that determine flood risk perception are varied. Moreover, the results from different studies can be ambiguous and/or contradictory (Lechowska, 2018). Social vulnerability can be divided into: (i) adaptive capacity, which is the capacity of an individual, community, society or organization to prepare for and respond to the consequences of a flood event (IPCC, 2012; Torresan et al., 2012); and (ii) coping capacity, which is the ability of an individual, community, society or organization to cope with adverse conditions resulting from a flood event using existing resources (IPCC, 2012; Torresan et al., 2012). The calculation of vulnerability is described in section 3.1.4. Risk is then calculated as the product of hazard, exposure and vulnerability as described in more detail in section 3.1.5, from which the direct tangible costs associated with the flood risk can be calculated (outlined in section 3.2).

**3.1.1 Input data**

There are several data sets used as inputs to the assessment of flood risk as outlined in Table 1. For the evaluation of flood hazard, the water height, flow velocity and flooded areas are provided by AAWA using the methodology described in the Supplementary Materials. Several data sets are used to evaluate flood exposure and vulnerability, but a key data set is Corine Land Cover (CLC) 2006 produced by the European Environment Agency (Steemans, 2008). Other data sets used to determine exposure include layers on population, infrastructure and buildings, areas of cultural heritage, protected areas and sources of pollution, where these data sets were obtained from different Italian ministries to complement the CLC. Data from OpenStreetMap on infrastructure and buildings were also used.

**Table 2.2.1 Hazard**

**1: Input data used to calculate risk.**

| Component of risk | Data | Source |
|---|---|---|
| Flood Hazard (low, medium, high hazard scenarios) | Water height (m) | AAWA; see Supplementary Materials for model details |
| | Water speed (m/s) | |
| | Flooded area (km$^2$) | |
| Flood Exposure | Population in residential areas | ISTAT, census data, 2001 |
| | Infrastructure and buildings | Corine Land Cover 2006, OpenStreetMap |
| | Types of agriculture | Corine Land Cover 2006 |
| | Natural and semi-natural systems | Corine Land Cover 2006 |
| | Areas of cultural heritage | Corine Land Cover 2006, MiBACT-Italian Ministry for cultural heritage |
| | Protected areas | Corine Land Cover 2006, MATTM-Italian Ministry for Environment, Veneto Region |
| | Point and widespread sources of pollution (Directives 82/501/EC, 2008/1/EC) | ISTAT, https://prtr.eea.europa.eu |
| Flood Vulnerability (Susceptibility) | Vegetation cover | Corine Land Cover 2006 |
| | Soil type | Corine Land Cover 2006 |

**3.1.2 Flood Hazard Mapping**

According to Article 6 of the 2007/60/CE Flood Directive (EU, 2007), when local authorities implement a Flood Risk Management Plan, three hazard scenarios must be considered:

1. A flood with a low probability, which is 300-year return period in the study area;
2. A flood with a medium probability, which is a 100-year return period in the study area; and
3. A flood with a high probability, which is a 30-year return period in the study area.

These have been calculated using a two-dimensional hydrological and hydraulic model to generate the water levels and the water speeds at a spatial resolution of 10 m (Ferri et al., 2010). Details of the model can be found in the Supplementary Materials. The hazard associated with these scenarios was calculated in relative terms as a value between 0 and 1.

**3.1.3 Flood Exposure Mapping**

The 2006 CLC map provides the underlying spatial information to calculate exposure; the land use classes used here are shown

| ID | Description |
|----|-------------|
| 1 | Residential |
| 2 | Hospital facilities, health care, social assistance |
| 3 | Buildings for public services |
| 4 | Commercial and artisan |
| 5 | Industrial |
| 6 | Specialized agricultural |
| 7 | Woods, meadows, pastures, cemeteries, urban parks, hobby agriculture |
| 8 | Tourist-Recreation |
| 9 | Unproductive |
| 10 | Ski areas, Golf course, Horse riding |
| 11 | Campsites |
| 12 | Communication and transportation networks: roads of primary importance |
| 13 | Communication and transportation networks: roads of secondary importance |
| 14 | Railway area |
| 15 | Area for tourist facilities, Zone for collective equipment (supra-municipal, subsoil) |
| 16 | Technological and service networks |
| 17 | Facilities supporting communication/transportation networks (airports, ports, service areas, parking lots) |
| 18 | Area for energy production |
| 19 | Landfills, Waste treatment plants, Mining areas, Purifiers |
| 20 | Areas on which plants are installed as per Annex I of Legislative Decree 18 February 2005, n. 59 |
| 21 | Areas of historical, cultural and archaeological importance; cultural heritage |
| 22 | Environmental goods |
| 23 | Military zone |
* * *
 in Table S1 in the Supplementary Materials. As mentioned above, the  first macro-category is the people affected
* * *
 by the flooding, or the exposure of the population ($E_P$), which is calculated as follows:

$$E_p = F_d * F_t \qquad\qquad (2)$$

where $F_d$ is a factor characterizing the density of the population in relation to the number of people present (Table 2), which uses gridded population from the census (Table 1), and $F_t$, which is the proportion of time spent in different locations (e.g., houses, schools, etc. ., using the land use types listed in Table S1) over a 24 hour period (Provincia Autonoma di Trento, 2006). The four classes in Table 2 reflect a very slight decrease in exposure as population density decreases, and were defined by stakeholders in the AAWA based on guidance from ISPRA (2012).

[revised manuscript text omitted]

360 **3.1.4 Flood Vulnerability Mapping**

Vulnerability is also quantified for each of the three macro-categories (i.e., people, economic activities and environmental/cultural-archaeological assets affected) as outlined below. but we additionally differentiate between physical and social vulnerability as described in Section 3.1 and shown in Figure 3.

[Figure]

365 **Figure 3: Hierarchical combination of indicators and relative weights (in brackets) to calculate the vulnerability of the population.**

**(i)  PeoplePhysical vulnerability of people affected by flooding**

To characterize theThe physical vulnerability associated with human presence, we refer topeople considers the values of
370 flow velocity ($v$) and water depthheight ($h$) values that produce "instability" with respect to remaining in an upright position. Many authors have dealt with the instability of people in flowing water (see e.g., Chanson and Brown, 2018)(see e.g., Chanson and Brown, 2018), and critical values have been derived from the product of $h$ and $v$ have been proposed. For example, Ramsbottom et al. (2004) and Penning-Rowsell et al. (2005)(2005) have proposed a semi-quantitative equation that links a flood hazard index, referred to as the Flood Hazard Rating (FHR), to $h$, $v$ and a factor related to the amount of
375 transported debris, i.e., the Debris Factor ($DF$), as follows:

$$FHR = h * (v + 0.5) + DF \qquad (3)$$

The values of $DF$ related to different ranges of $h$, $v$ and land use are reported in Table 4.

380

The values of the $DF$ related to different ranges of $h$, $v$ and land use are reported in Table 4, which were taken from a study by the UK Department for Environment, Food and Rural Affairs (DEFRA) and the UK Environment Agency (2006) as reported in ISPRA (2012).

385 **Table 4:4: The Debris Factor (*DF*) for different water depths (*h*), flow velocities (*v*) and land uses.**

| Values of $h$ and $v$ | Grazing/Agricultural land | Forest | Urban |
|---|---|---|---|
| 0 m $< h \leq$ 0.25 m | 0 | 0 | 0 |
| 0.25 m $< h \leq$ 0.75 m | 0 | 0.5 | 1 |

| $h > 0.75$ OR $v > 2$ m/s | 0.5 | 1 | 1 |

Based onUsing the FHR, the physical vulnerability of the population, $V_P$, can be calculated. One assumption, which is that people are vulnerable at water heights greater than 0.25m. People located in "hospital and social assistance structures", whosesummarized in Figure 4.

[Figure]

Figure 4: Physical vulnerability is considered as 1 for an FHR > 0.75, represent an exception because the physical condition of people living in such structures makes them more vulnerable. These relationships are summarized in Figure 2.

The method to evaluate the adaptive and coping capacities is based on the hierarchical combination of indicators as shown in Figure 3, where the weights used in the calculation are reported in brackets. The data related to the social indicators have different units of measurement. Therefore, it is necessary to adopt a normalization procedure using value functions (Mojtahed et al., 2013).

[Figure]

**Figure 2: Vulnerability values for the population (V_p) as a function of water depthheight ($h$) and flow velocity ($v$).**

To evaluate the Coping Capacity, four different variables are included (shown in Figure 4 along with their normalized functions):

**(ii) Social vulnerability of people affected by flooding**

Figure 3 shows the components of social vulnerability, i.e., the adaptive and coping capacity and their respective indicators, along with the weights associated with each of them. The weights and values assigned to each of these indicators have been determined through an expert consultation process carried out by AAWA. Because the different indicators have varying units of measurement, they were first normalized so that they could be combined. Several normalization techniques exist in the literature (Biausque, 2012) but the 'value function' was chosen because it represents a mathematical expression of a human judgement that can be compared in a systematic and explicit way (Beinat, 1997; Mojtahed, et al., 2013). The coping

410 capacity is comprised of the following demographic and emergency measure indicators, where the corresponding value functions are shown in Figure S1:

- Dependency ratio: the number of citizens aged under 14 and over 65 as a percentage of the total population. A high value of this index implies a reduced ability to adapt to hazardous events.
- Foreigners: the number of foreigners as a percentage of the total population. Due to language barriers and
415 other cultural reasons, areas with a high number of immigrants may not cope as well after a flood event and during emergency situations.
- Number of people involved in emergency management: the number of operators who have been trained to manage an emergency in the region, expressed qualitatively as low, medium and high.
- How frequently civil protection plans are updated: Updating is
420 measured in months to years and indicates how often new hydraulic, urban and technological information is incorporated into civil protection plans.

The adaptive capacity is comprised of three components: the early warning system, equity and risk spread. Early warning systems are evaluated according to three criteria, where the value functions are shown in Figure S2:

425 - Lead time (or warning time): the number of hours before an event occurs that was predicted by the early warning system.
- Content: the amount of information provided by the early warning system, such as the time and the peak of the flooding at several points across the catchment.
- Reliability: this is linked to the uncertainty of the results from the meteorological forecasts and the hydrological
430 models (Schroter et al., 2008). False alarms can cause inconvenience to people, hinder economic activities, and people may be less likely to take warnings seriously in the future; therefore, they should be minimized.

Finally, equity and spread (shown in Figure S3) are characterized by:

- Gini Index: a measure of the inequality of income distribution within the population. A value of 0 means perfect
435 equality while 1 is complete inequality.
- Number of hospital beds: this is calculated per 1000 people.
- Insurance density: this is the ratio of total insurance premiums (in €) to the total population (Lenzi and Millo, 2005). Values with higher insurance density lead to increased adaptive capacity. However, the insurance density is set to zero because insurance companies in this part of Italy do not currently offer premiums to protect goods against
440 flood damage.
- The frequency at which information on hazard and risk are updated: this is measured in months to years and indicates the ability of institutions to communicate the conditions of danger and risk to the population.
- Involvement of citizens: This is based on the number of students, associations such as farmers and professionals, and citizens that can be reached across large areas through social networks (WP7 WSI Team, 2013) to disseminate
445 information. The values in Figure S3d show the maximum achievable value in the three categories of citizen involvement.

[Figure]

Figure 3: Hierarchical combination of indicators and relative weights (in brackets) to calculate the vulnerability of the population.

450

[Figure]

Figure 4: Variables as normalized index functions for evaluating the Coping Capacity (from De Luca, 2013).

455

460    Finally, forecasting systems are evaluated according to the three criteria, where the value functions are shown in Figure 6:

- Reliability: this is linked to the uncertainty of the results from the meteorological forecasts and the hydrological models (Schroter et al., 2008). False alarms can inconvenience people and hinder economic activities and should, therefore, be minimized.

465    - Lead time (or warning time): the number of hours before an event occurs that was predicted by the early warning system.

- Information Content: the amount of information provided by the forecasting systems, such as the time and the peak of the flooding at several points across the catchment.

[Figure]

470    Figure 6: Normalized function of the indices linked to the forecasting systems: A) reliability, B) lead time, and C) information content (from De Luca, 2013).

**(ii) Economic activities affected**

The value for social vulnerability is the sum of the coping and adaptive capacities while the final value for the vulnerability of people is calculated by multiplying the physical and the social vulnerability together.

475    **(iii) Physical vulnerability of economic activities affected by flooding**

The vulnerability associated with economic activities, $V_E$, is evaluated using the land use categories in Table 1. Three main aspects are considered: considers buildings, network infrastructure and agricultural areas. For buildings, which are found in land use types 1 to 5, 14 to 15, 17 to and 23 in Table 1,the effects from flooding include collapse due to water pressure and/or undermining of the foundations. Moreover, solid materials, such as debris and wood, can be carried by a flood and

480    can cause additional damage to structures. A damage function for brick and masonry buildings has been formulated by Clausen and Clark (1990)(1990). Regarding losses to indoor goods, laboratoryLaboratory results have shown that at a water height of 0.5m, the loss to indoor goods is around 50%, which is based on an evaluation made by Risk Frontiers, an independent research center sponsored by the insurance industry. The structural vulnerability of the buildings and losses of the associated indoor goods is shown in Figure 7S4 as a function of the height of the water depth and flow velocity., which

485    are applied to land use types containing buildings (Table S1). For the camping land use type 11 (Table 1S1), the values have been modified based on results from Majala (2001).

[Figure]

**Figure 7:** Vulnerability

of the road network is evaluated for land use types 12 and 13 in Table  S1, which occurs when it is not possible to use the road due to flooding. This is based on an estimation of the water height and the critical velocity at which vehicles become unstable during a flood, which are derived from direct observation in laboratory experiments and from a report on the literature in this area (Reiter, 2000; Shand et al., 2011); the vulnerability function for the road network is presented in Figure

S5. Regarding technological and service networks (land use type 16, Table S1), we assume a vulnerability value equal to 1 if the water height and flow velocity are greater than 2 m and 2 m/s, respectively, otherwise 0.

To assess the vulnerability in agricultural areas (land use types 6 and 7 in Table S1), we assume that the damage is related to harvest loss, and when considering higher flow velocities and water heights, to agricultural buildings and internal goods.  Citeau (2003) provides relationships that take water height and flow velocity into account, e.g., the maximum height is 1 m for orchards and 0.5 m for vineyards, and the maximum velocity varies from 0.25 m/s for vegetables and 0.5 m/s for orchards. Concerning cultivation in greenhouses, the maximum damage occurs at a height of 1 m. Finally, high velocities can cause direct damage to cultivated areas but can also lead to soil degradation due to erosion. The vulnerability values for four different types of land as a function of water height and flow velocity are shown in Figure S6. In the case of unproductive land (land use type 9 in Table 1), the vulnerability is assumed to be 0.25, regardless of the *h* and *v* values.

[Figure]

**Figure 8: Vulnerability values**Physical vulnerability of

**(iii)(iv)   environmental and cultural heritage assets affected by flooding**

 Environmental flood susceptibility is described using contamination/pollution and erosion as indicators. Contamination is caused by industry, animal/human waste and stagnant flooded waters. Erosion can produce disturbance to the land surface and to vegetation but can also damage infrastructure.  The approach taken here was to identify protected areas that could potentially be damaged by a flood. For areas  susceptible to nutrients, including those identified as vulnerable in Directive 91/676/CEE (Nitrate), and for those defined as susceptible in Directive 91/271/CEE (Urban Waste), we assume a value of 1 for vulnerability (land use type 20 in Table
S1). Similarly, in  areas identified for habitat and species protection, i.e., sites belonging to the Natura 2000 network established in accordance with the Habitat Directive 92/43/CEE and Birds Directive 79/409/CEE (land use types 8 and 22 in Table S1), the presence of  relevant pollution sources was identified (Tables 1 and S1) and assigned a vulnerability of 1. In the absence of pollution sources , the vulnerability was calculated as 0.25 if the flood velocity was less than or equal to 0.5 m/s and the water depth was less than or equal to 1 m; otherwise it was 0.5. Regarding cultural heritage (land use type 21 in Table S1), we assigned a vulnerability of 1 to these areas, taking a conservative approach.

[Figure]
* * ** * *
$$R = \frac{p_P \cdot R_P + p_E \cdot R_E + p_{ECH} \cdot R_{ECH}}{p_P + p_E + p_{ECH}}, \tag{4}$$

540 **3.1.5 Mapping flood risk before and after implementation of a CO on flood risk management**

Once the hazard, exposure and vulnerability are mapped, the flood risk, $R$, for the three flood hazard scenarios, $i$, can be mapped as follows:

545
$$R = \sum_{i=1}^{3} R_i = \frac{w_P\,(H_i \cdot E_P \cdot V_P) + w_E\,(H_i \cdot E_E \cdot V_E) + w_{ECH}\,(H_i \cdot E_{ECH} \cdot V_{ECH})}{w_P + w_E + w_{ECH}} \tag{4}$$

where $H$, $E$ and $V$ are the hazard, exposure and vulnerability associated with the three macro-categories $P$, $E$ and $ECH$ which are the people, economic activities and environmental/cultural-archaeological assets affected, and $w_P$, $w_E$ and $w_{ECH}$ are weights applied to each macro-category, with values of 10, 1 and 1, respectively, which were defined based on stakeholder

550 interviews.  undertaken by AAWA. To establish the level of risk  four risk classes were defined (Table 5).

555

**Table 5: Definition of risk classes.**

| Range of R | Description | Risk Category |
|---|---|---|
| $0.1 < R \le 0.2$ | Low risk where social, economic and environmental damage are negligible or zero | R1 |
| $0.2 < R \le 0.5$ | Medium risk for which minor damage to buildings, infrastructure and environmental heritage is possible, which does not affect the safety of people, the usability of buildings and economic activities | R2 |
| $0.5 < R \le 9$ | High risk in terms of safety of people, damage to buildings and infrastructure (and/or unavailability of infrastructure), interruption of socio-economic activities and damage related to the environmental heritage | R3 |
| $0.9 < R \le 1$ | Very high risk including loss of human life and serious injuries to people, serious damage to buildings, infrastructure and environmental heritage, and total disruption of socio-economic activities | R4 |

560

~~According to EU directive 2007/60/EC, the flood risk plan must contain an analysis of the costs and benefits (hereafter referred to as CBA) that would be generated from each planned intervention. The purpose of the CBA is to compare the efficiency and effectiveness of different alternatives in technological, economic, social and environmental terms. These interventions can be public policies, projects or regulations that can be used to solve a specific problem.~~

565

are designed to support decision makers in assigning monetary value to damages and classifying them, as proposed by Merz et al. (2010). In this analysis, only the direct tangible costs due to damage resulting from a flood event are considered.

**2.3.1 Determining the effectiveness of an action plan**

These risk classes were then mapped with and without the implementation of the CO for flood risk management. The main change in the calculation of risk is in the social dimension of vulnerability. Before the CO is implemented, this component has a value close to 1. Based on the experience gained in the WeSenseIt project and the goals of the CO, the changes in social vulnerability with the implementation of the CO are shown in Table 6, which decreases the social vulnerability to a value of 0.63. For example, in the coping capacity, the number of people employed in emergency management does not change but as a result of the CO, they will work in a much more efficient manner due to the technology that allows for better emergency management. These tools will also lead to more frequent updating of civil protection plans as well as hazard and risk information updates. In addition, the early warning system will improve in terms of lead time, content and reliability through the greater involvement of trained volunteers and citizens.

Table 6: Changes in the indicators of social vulnerability with and without implementation of the CO on flood risk management.

| Social vulnerability | Indicator | Value without CO | Value with CO |
|---|---|---|---|
| Adaptive capacity | Number of people involved in emergency management | Medium | High |
| | Frequency of civil protection plan updating | > 5 years | > 2 years |
| Coping capacity | Lead time of EWS | < 6 hours | 24-72 hours |
| | Content of EWS | Little information | Very detailed information |
| | Reliability of EWS | None | High |
| | Citizen involvement | None | Citizens of large area |
| | Hazard and risk information updating | > 5 years | 1-2 years |

**3.2. Quantifying the risk reduction after implementation of the CO for flood risk management**

To determine the effectiveness (or benefit) of implementing the action plan,CO for flood risk management, we consider the modificationchanges to the risk class as a result of the intervention must be determined. The, which has been mapped across the study area before and after implementation of the CO. To aggregate this change in risk after implementation of the CO, the Synthetic Index of Risk Reduction (ISRR), which represents the effectiveness of an intervention relative to the current situation, can then be ) is calculated as follows:

$$ISRR = \frac{\sum_{ij} k_{ij} \cdot A_j}{\sum_j A_j},$$

(57)

where $A_j$ is the flooded area after an intervention and $k_{ij}$ are the weights listed in Table 6 for the risk class $i$ before the intervention and $j$ after the intervention. We then use the *ISRR* from equation (5) to calculate the CBA value:

$$CBA = \frac{Costo\,opera}{ISRR \cdot 10^6},$$

(6)

where *Costo opera* is the cost of the intervention.

Table 6:where $A_i$ is the flooded area after the CO is implemented and $k_{ij}$ are the weights from Table 7 for the risk class $i$ before the CO is implemented and $j$ after the implementation. The weights in Table 7 have been determined through expert consultation within the AAWA, supported by the guidelines from ISPRA (2012). If the ISRR is positive, then the overall risk is reduced.

**Table 7: Weights ($k$) for the Synthetic Index of Risk Reduction (ISRR) for changes in risk  after implementation of the CO for flood risk management**

[revised manuscript text omitted]

In Figure 11, we provide a map showing

| | | | |
|---|---|---|---|
| Low | 185.12 | 294.77 | 370.07 |
| Medium | 118.87 | 161.82 | 225.67 |
| High | 54.18 | 74.55 | 104.61 |
| Total | 358.17 | 531.14 | 700.35 |

685 **Table 9: The risk classes for each return period in terms of area flooded before implementation of the CO.**

| Risk Class | 30 year return period | 100 year return period | 300 year return period |
|---|---|---|---|
| | Area (km$^2$) | | |
| Low (R1) | 160.29 | 254.29 | 318.80 |
| Medium (R2) | 137.26 | 191.89 | 262.03 |
| High (R3) | 56.70 | 79.23 | 110.29 |
| Very High (R4) | 3.92 | 5.73 | 9.23 |
| Total | 358.17 | 531.14 | 700.35 |

Figure 5 shows the areas at risk in the territory of Padua for a 100-year flood event. Risk classes R1 (low risk) and R2 (medium risk) have the highest areas for all flood event frequencies. Although areas in R3 (high risk) and R4 (very high risk) may comprise a relatively smaller area when compared to the total area at risk, these also coincide with areas of high
690 concentrations of inhabitants in Vicenza and Padua.

695

[Figure]

**Figure 5: Risk map for the metropolitan area of Padua for a 100-year flood event before implementation of a CO on flood risk management.**

**4.1.2 Expected damage**

700 The direct damage was calculated for the three flood scenarios: high chance of occurrence (every 30 years), medium (every 100 years) or low (every 300 years) and is summarized in Table 10.

**Table 10: Direct damage (without the CO) for three flood scenarios.**

| **Scenarios** (chance of flood occurrence) | **Return period** | **Damage** (million €) |
|---|---|---|
| High | 30 years | 7,053 |
| Medium | 100 years | 8,670 |
| Low | 300 years | 10,853 |

In the event of very frequent flood events, urban areas will be damaged. Furthermore, moving from an event with a high probability of occurrence to one with a medium probability results in a significant increase in the area flooded (i.e., a 48% increase as shown in Table 8) but with a smaller increase in damage

710 (i.e., around 20%). This is explained by the fact that the flooded areas in a 100-year flood event (but not present in a 30-year flood event) are under agricultural use. Similar patterns can be observed when comparing floods with a low and high probability of occurrence. Substituting the values in Table 10 into equation (5), we obtain an expected average annual damage (EAD) of €248.5 million Euros.

[Figure]

| Risk Class R4 | Risk Class R3 | Risk Class R2 | Risk Class R1 |

715

3.2

Flood risk estimation with **the implementation of** a flood risk
720 management CO

4.2.1 Hazard **and**  risk

As mentioned previously, the hazard remains unchanged (i.e., the results reported in Table 8)), but the risk is reduced after implementation of a CO for flood risk management as shown in Table 11
725  due to the reductions in vulnerability outlined in section 3.1.5. The areas affected in the high (R3) and very high classes (R4) are significantly reduced (R4 to almost zero) compared to the results shown in Table 9 but the areas in the lower risk classes increase. The risk map for a 100-year flood event for the territory of Padua is shown in Figure 6, where the reduction in areas at high and very high risk are clearly visible compared to the situation before implementation of the CO, which is shown in Figure 5.

730

Table 11: **The risk classes for each return period of flooding in terms of area** flooded after implementation of the CO.

| Risk class | 30 year return period | 100 year return period | 300 year return period |
|---|---|---|---|
| | Area (km$^2$) | | |
| R1 (Low) | 170.96 | 268.68 | 337.78 |
| R2 (Medium) | 168.99 | 235.18 | 322.41 |
| R3 (High) | 18.19 | 27.19 | 40.04 |
| R4 (Very High) | 0.03 | 0.09 | 0.12 |
| Total | 358.17 | 531.14 | 700.35 |

735

[Figure]

🔴 Risk Class R4    🟠 Risk Class R3    🟡 Risk Class R2    🟢 Risk Class R1

740    **Figure 6: Risk map for the metropolitan area of Padua for a 100-year flood event after implementation of a CO on flood risk management.**

**4.2.2** **Expected** damage
745

The residual damage was calculated for the three flood scenarios after implementation of the CO on flood risk reduction, which is shown in Table 12. Substituting  these residual damage values  into equation (5), we obtain an EAD of €111,3 million Euros, which is a 45% reduction in the damage compared to results before the CO implementation.

750

Table 12: Comparison of the direct (without CO) and residual damage (with CO) for three flood  scenarios and the cost difference .

| Scenarios (chance of flood occurrence) | Return period | Direct damage (million €) | Residual damage (million €) | Difference in costs (million €) |
|---|---|---|---|---|
| High | 30 years | 7,053 | 1,573 | -5,480 |
| Medium | 100 years | 8,670 | 5,440 | -3,230 |
| Low | 300 years | 10,853 | 3,420 | -7,433 |

The CO for flood risk management has an estimated cost
755    of around 5 million Euros (as detailed in Table S2 in the Supplementary Materials). Taking the EAD with and without implementation of the CO, the annual benefit in terms of avoided damage is approximately 137.2 million Euros. Hence the benefits considerably outweigh the costs. The ISRR value is 2.5. , which also indicates a
760    positive reduction in risk. The same methodology was applied to the construction of a retention basin in the municipalities of Sandrigo and Breganze to improve the hydraulic safety of the Bacchiglione River. Against an expected cost of 7 million Euros, which is

much higher than the estimated cost for implementing the CO, a significant reduction in flooded areas would be obtained although high risk would still be evident in the  city of Padua. In terms of damage reduction with the construction of the retention basin, we would obtain an EAD of 140.7 million Euros so the cost to benefit ratio would be much lower.

[Figure]

**Figure 12: Risk map for the metropolitan area of Padua for a 100-year flood event after the intervention of a Citizen Observatory.**

**5 Discussion and Conclusions**

There is currently a lack of available, appropriate and peer-reviewed evaluation methods and evidence on the added value of citizen observatories, which is required before they will be more widely adopted by policy makers and practitioners. This paper has aimed to fill this gap by demonstrating how a traditional cost-benefit analysis.  can be used to capture the value of a CO for flood risk management. Although the CO is still being implemented, the proposed methodology was applied using primary empirical evidence from a CO pilot that was undertaken by the WeSenseIt project in the smaller Bacchiglione catchment.  to guide changes in the values associated with social vulnerability once the

CO is implemented. This allowed the  risk and flood damages to be calculated with and without implementation of the CO, which showed that implementation of a CO in the Brenta-Bacchiglione is able to reduce the damage, and consequently the risk, for the inhabited areas from an expected average annual damage (EAD) of €248.5 to €111.3 million euros, i.e., a reduction of 45%. Hence, the implementation of the CO could significantly reduce the damage and consequently the risk for the inhabited areas of Vicenza, Padua, Torri di Quartesolo, Longare and Montegaldella. The nature

of the methodology also means that it can be applied to other catchments in any part of Italy or other parts of the world that are considering the implementation of a CO for flood risk management purposes.

The evidence on the costs and benefits of COs for flood risk management generated by this case study can provide insights that policy makers, authorities and emergency managers can use to make informed choices about the adoption of COs for improving their respective flood risk management practices. In Italy, in general, citizen participation in flood risk management has been relatively limited. The previous strategy in the Brenta-Bacchiglione catchment has focused on structural flood mitigation measures, dealing with emergencies and optimizing resources for rapid response. The inclusion of a CO on flood risk management has been a true innovation in the flood risk management strategies of this region. Future research can focus on the application of the methodology in other catchments as well as to other fields of disaster management beyond floods. Such applications will serve to generate a broader evidence base for using these types of  cost-benefit methodologies.

However, there are also limitations associated with this approach. For example, the  analysis presented here did not consider indirect costs, such as those incurred after the event takes place, or in places other than those where the flooding occurred (Merz et al., 2010). In accordance with other authors (e.g., van der Veen et al., 2003), all expenses related to disaster response (e.g., costs for sandbagging, evacuation) are classified as indirect damage. However, the presence of the CO in this catchment does reduce the costs related to emergency services, securing infrastructure, sandbagging and evacuation, all of which can be substantial during a flood event. Therefore, an analysis that takes indirect costs into account could help to further convince policy makers of the feasibility of a CO solution. Similarly, intangible costs were not considered, i.e., the values lost due to an adverse natural event where monetary valuation is difficult because the impacts do not have a corresponding market value (e.g., health effects). Furthermore, the vulnerability assessment of economic activities considers only water depth and flow velocity but not additional factors such as the dynamics of contamination propagation in surface waters during the flood or the duration of the flood event, all of which could be taken into account in estimating the structural damage and monetary losses in the residential, commercial and agricultural sectors.

Another limitation is that this methodology is built on many assumptions, i.e., the numerous coefficients, value functions and weights used to estimate the exposure and vulnerability. Many of these values have been derived through expert consultation and experience and validated internally within AAWA or other Italian agencies. Value functions, in particular, are a way of capturing human judgement in way that can be quantified in situations of high uncertainty. We would argue that the expert consultations have not been undertaken lightly and have often resulted in conservative estimates in the values. Other values have been derived from the literature, all of which will have some uncertainties associated with their derivation. We have not undertaken an uncertainty analysis or a sensitivity analysis. Although we might be able to demonstrate a range of costs and benefits through such an approach, the current benefits heavily outweigh the costs so tweaking individual parameters will be unlikely to have large effects. That said, this cost-benefit analysis is hypothetical because the CO for flood risk management is still being implemented. Hence the real benefits will only be realized once the CO is fully operational. At that stage it will be interesting to validate the assumptions about reductions in social vulnerability and which indicators are the key to reducing flood risk.

Despite these various limitations, this analysis has highlighted the feasibility of a non-structural flood mitigation choice such as a CO for flood risk management compared to the implementation of much more expensive structural measures (e.g., retention areas) in terms of the construction costs and the cost of maintenance over time. By involving citizens in a two-way communication with local authorities through a CO, flood forecasting models can be improved, increased awareness of flood hazard and flood preparedness can be achieved, and community resilience to flood risk can be bolstered.

**Acknowledgements**

The research has been partly funded by the FP7 WeSenseIt project (No. 308429) and the Horizon2020 WeObserve project (No. 776740).

---

## Author Comment (AC4) · 13 May 2020

**The Value of Citizen Science for Flood Risk Reduction: Cost-benefit Analysis of a Citizen Observatory in the Brenta-Bacchiglione Catchment**

Michele Ferri[1], Uta Wehn[2], Linda See[3], Martina Monego[1], Steffen Fritz[3]

[1]Alto-Adriatico Water Authority (AAWA), Cannaregio 4314, 30121 Venice, Italy
[2]IHE Delft Institute for Water Education, Westvest 7, 2611 AX Delft, The Netherlands
[3]International Institute for Applied Systems Analysis (IIASA), Schlossplatz 1, 2361 Laxenburg, Austria

*Correspondence to*: Michele Ferri (michele.ferri@distrettoalpiorientali.it)

**Abstract.** Citizen observatories are a relatively recent form of citizen science. As part of the flood risk management strategy of the Brenta-Bacchiglione catchment, a citizen observatory for flood risk management has been proposed and is currently being implemented. Citizens are involved through monitoring water levels and obstructions and providing other relevant information through mobile apps, where the data are assimilated with other sensor data in a hydrological-hydraulic model used in early warning. A cost benefit analysis of the citizen observatory was undertaken to demonstrate the value of this approach in monetary terms. Although not yet fully operational, the citizen observatory is assumed to decrease the social vulnerability of the flood risk. By calculating the hazard, exposure and vulnerability of three flood scenarios (required for flood risk management planning by the EU Directive on Flood Risk Management) with and without the proposed citizen observatory, it is possible to evaluate the benefits in terms of the average annual avoided damage costs. Although currently a hypothetical exercise, the results showed a reduction in avoided damage of 45% compared to a business as usual scenario. Thus, linking citizen science with hydrological modelling, and to raise awareness of flood hazards, has great potential in reducing flood risk in the Brenta-Bacchiglione catchment in the future. Moreover, such approaches are easily transferable to other catchments.

**1 Introduction**

In 2018, flooding affected the highest number of people of any natural disaster globally and caused major damage worldwide (CRED, 2019). With climate change, the frequency and magnitude of extreme events will increase, leading to a higher risk of flooding (Schiermeier, 2011). This risk will be further exacerbated by future economic and population growth (Tanoue et al., 2016). Thus, managing flood risk is critical for reducing future negative impacts. Flood risk assessments are undertaken by the insurance industry for determining properties at high risk (Hsu et al., 2011), but they are also a national requirement in the European Union as set out in the EU Flood Risk Management Directive, which requires that flood risk management plans are produced for each river basin (EU, 2007; Müller, 2013). The assessment of flood risk involves quantifying three main drivers (National Research Council, 2015): (a) flood hazard, which is the probability that a flood of a certain magnitude will occur in a certain period of time in a given area; (b) exposure, which is the economic value of the human lives and assets affected by the flood hazard; and (c) vulnerability, which is the degree to which different elements (i.e., people, buildings, infrastructure, economic activities, etc.) will suffer damage associated with the flood hazard. In addition, flood risk can be mitigated through hard engineering strategies such as implementation of structural flood protection schemes, soft engineering approaches comprising more natural methods of flood management (Levy and Hall, 2005), and community-based flood risk management (Smith et al., 2017). As part of requirements in the EU Flood Risk Management Directive, any mitigation actions must be accompanied by a cost-benefit analysis.

Flood hazard is generally determined through hydrological and hydraulic modelling. Hence accurate predictions are critical for effective flood risk management, particularly in densely populated urban areas (Mazzoleni et al., 2017). The input

40  data required for modelling are often incomplete in terms of resolution and density (Lanfranchi et al., 2014), which translates into variable accuracy in flood predictions (Werner et al., 2005). New sources of data are becoming available to support flood risk management. For example, the rise of citizen science and crowdsourcing (Howe, 2006; Sheldon and Ashcroft, 2016), accelerated by the rapid diffusion of information and communication technologies, is providing additional, complementary sources of data for hydrological monitoring (Njue et al., 2019). Citizen science refers to the involvement of

45  the public in any step of the scientific method (Shirk et al., 2012). However, one of the most common forms of participation is in data collection (Njue et al., 2019). Citizen observatories (CO) are a particular form of citizen science in so far as they constitute the means not just for new knowledge creation but also for its application, which is why they are typically set up with linkages to specific policy domains (Wehn et al., 2019). COs must therefore include a public authority (e.g., a local, regional or national body) to enable two-way communication between citizens and the authorities to create a new source of

50  high quality, authoritative data for decision making and for the benefit of society. Moreover, COs involve citizens in environmental observations over an extended period of time of typically months and years (rather than one-off exercises such as data collection 'Blitzes'), and hence contribute to improved temporal resolution of the data, using dedicated apps, easy-to-use physical sensors and other monitoring technologies linked to a dedicated platform (Liu et al., 2014; Mazumdar et al., 2016). COs are increasingly being used in hydrology/water sciences and management and in various stages of the flood

55  risk management cycle, as reviewed and reported by Assumpção (2018), Etter et al. (2018), Mazzoleni et al. (2017), Buytaert et al. (2014), Wehn and Evers (2015) and Wehn et al. (2015). These studies found that the characteristic links of COs to authorities and policy do not automatically translate into higher levels of participation in flood risk management, nor that communication between stakeholders improves; rather, changes towards fundamentally more involved citizen roles with higher impact in flood risk management take years to evolve (Wehn et al., 2015).

60  The promising potential of the contribution of COs to improved flood risk management is paralleled by limited evidence of their actual impacts and added value. Efforts are ongoing such as the consolidation of evaluation methods and empirical evidence by the H2020 project WeObserve[1] Community of Practice on the value and impact of citizen science and COs, and the development and application of methods for measuring the impacts of citizen science by the H2020 project MICS[2]. To date, the societal and science-related impacts have received most attention, while the focus on economic impacts, costs and

65  benefits has been both more limited and more recent (Wehn et al., 2020a). The studies that do focus on economic impacts related to citizen science (rather than citizen observatories) propose to consider the time invested by researchers in engaging and training citizens (Thornhill et al., 2016); to relate cost and participant performance for hydrometric observations in order to estimate the cost per observation (Davids et al., 2019); to estimate the costs as data-related costs, staff costs and other costs; and the benefits in terms of scientific benefits, public engagement benefits and the benefits of strengthened capacity of

70  participants (Blaney et al., 2016); and to compare citizen science data and in-situ data (Goldstein et al., 2014; Hadj-Hammou et al., 2017). Wehn et al. (2020b) assessed the value of COs from a data perspective and a cost perspective, respectively, to qualify the degree of complementarity that the data collected by citizens offers to in-situ networks and to quantify the relation between the investments required to set up a CO and the actual amount of data collected. Based on a comparison of four COs, they suggest that setting up a CO for the sole purpose of data collection appears to be an expensive undertaking

75  (for the public sector organization(s) benefitting from the respective CO) since, depending on the process of (co)designing the CO, it may not necessarily complement the existing in-situ monitoring network (with the likely exception of infrastructure-weak areas in developing countries).

Overall, there is a lack of available, appropriate and peer-reviewed evaluation methods and of evidence of the added value of COs, which is holding back the uptake and adoption of COs by policy makers and practitioners. In this paper, we

80  take a different approach to previous studies by using a more conventional cost-benefit analysis framework to assess the
* * *
[1] https://www.weobserve.eu/
[2] https://mics.tools/

implementation of a citizen observatory on flood risk management in the Brenta-Bacchiglione catchment in northern Italy. The purpose of a cost-benefit analysis is to compare the effectiveness of different alternative actions, where these actions can be public policies, projects or regulations that can be used to solve a specific problem. We treat the citizen observatory in the same way as any other flood mitigation action for which a cost-benefit analysis would be undertaken in this catchment. Although the citizen observatory is still being implemented, the assumptions for the cost-benefit analysis are based on primary empirical evidence from a CO pilot that was undertaken by the WeSenseIt project in the town of Vicenza, Italy, described in more detail in section 2.1 and now extended to the wider catchment (sections 2.2 and 2.3). In section 3 we present the flood risk and cost benefit methodology followed by the results in section 4. Conclusions, limitations of the methodology and case-specific insights are provided in section 5.

**2 The Development of a Citizen Observatory for Flood Risk Management**

**2.1 The WeSenseIt Project**

Through the WeSenseIt research project (www.wesenseit.eu), funded under the 7th framework program (FP7-ENV-2012 n° 308429), a CO for flood risk was developed with the Upper Adriatic Basin Authority in northern Italy. The objective of this CO was to collect citizen observations from the field, and to obtain a broader and more rapid picture of developments before and during a flood event. The CO involved many stakeholders concerned with the management and use of the water resources, and with water-related hazards in the Bacchiglione River basin. The main actors included the local municipalities, the regional and local civil protection agencies, environment agencies and the irrigation authorities. The Alto Adriatico Water Authority (AAWA) facilitated access to a highly trained group of citizen observers, namely civil protection volunteers, who undertook the observations (i.e., using staff gauges with a QR code to measure the water level and reporting water way obstructions) as part of their volunteer activities. Additional volunteers were also recruited during the project from the Italian Red Cross, the National Alpine Trooper Association, the Italian Army Police and other civil protection groups, with more than 200 volunteers taking part in the CO pilot. Training courses for the volunteers were organized to disseminate and explain the use of a smartphone application and an e-collaboration platform, which were developed as part of the WeSenseIt project. In addition to the low cost sensing equipment, the CO also used data from physical sensors: 3 sonar sensors (river water level), 4 weather stations (wind velocity and direction, precipitation, air temperature and humidity) and 5 soil moisture sensors. The combined visualization of the sensors (including existing sensors from the Venice Environment Agency) was available in the online e-collaboration platform. During the WeSenseIt project, research into the value of crowdsourced data for hydrological modelling was investigated (Mazzoleni et al., 2017, 2018) and found to complement traditional sensor networks.

This pilot was later adopted by the European Community as a "good practice" example of the application of Directive 2007/60/EC. After the positive experience in WeSenseIt, funds were made available to develop a CO for flood risk management at the district scale, covering the larger Brenta-Bacchiglione catchment. At this stage, a cost-benefit analysis was undertaken, which is reported in this paper. The next section provides details of the Brenta-Bacchiglione catchment followed by ongoing developments in the CO for flood risk management.

**2.2 The Brenta-Bacchiglione Catchment**

The Brenta-Bacchiglione River catchment includes the Retrone and Astichiello Rivers, and falls within the Veneto Region in Northern Italy, which includes the cities of Padua and Vicenza (Figure 1). The catchment is surrounded by the Beric hills in the south and the Prealpi in the northwest. In this mountainous area, rapid or flash floods occur regularly and are difficult to predict. Rapid floods generally affect the towns of Torri di Quartesolo, Longare and Montegaldella, although there is also widespread flooding in the cities of Vicenza and Padua, which includes industrial areas and areas of cultural heritage. For

example, in 2010, a major flood affected 130 communities and 20,000 individuals in the Veneto region. The city of Vicenza was one of the most affected municipalities, with 20% of the metropolitan area flooded.

[Figure]

125 **Figure 1: Location of the Brenta-Bacchiglione catchment and its urban communities.**

**2.3 The Citizen Observatory for Flood Risk Management for the Brenta-Bacchiglione Catchment**

The CO for flood risk management, which is currently being implemented, was included in the prevention measures of the Flood Risk Management Plan (PGRA) for the Brenta-Bacchiglione catchment. The purpose of the CO is to strengthen communication channels before and during flood events in accordance with the EU Flood Directive on Flood Risk
130 Management, to increase the resilience of the local communities and to address residual risk. Building on the WeSenseIt experience, an IT platform to aid decision support during the emergency phases of a flood event is being implemented. This platform will integrate information from the hydrological model, which is equipped with a data assimilation module that integrates the crowdsourced data collected by citizens and trained experts with official sensor data. A mobile app for data collection based on the WeSenseIt project is under development. The platform and mobile technology will guarantee user
135 traceability and facilitate two-way communication between the authorities, the citizens and the operators in the field, thereby significantly increasing the effectiveness of civil protection operations during all phases of an emergency. The fully operational CO will include 64 additional staff gauges equipped with a QR code (58 to measure water level and 6 for snow height), 12 sonar sensors and 8 weather stations.

To engage and maintain the involvement of "expert" CO participants (i.e., civil protection volunteers, technicians
140 belonging to professional associations, members of environmental associations), a set of training courses will be run. The involvement of technicians (formalized in November 2018 with an agreement between the respective associations and AAWA) offers an important opportunity to use the specific knowledge and expertise of these technicians to better understand the dynamics of flood events and to acquire high quality data to feed the models and databases. When an extreme event (i.e., heavy rain) is forecast, AAWA will call upon any available technicians in providing data (with a reimbursement
145 of 75 €/day (including insurance costs) and a minimum activity per day of 3 hours). There are currently 41 technicians involved in the CO, which includes civil/hydraulic/geotechnical engineers, agronomists and forestry graduates. Participants must attend two training sessions followed by a final examination. To give an example of the valuable information that the expert CO participants can provide, AAWA called upon technicians during two heavy rainfall events (November 2019; 5 days). These technicians collected relevant data on the status of the rivers including the vegetation, the water levels, the
150 status of bridges and levees, collecting 1660 images and completing 700 status reports.

To engage citizens, a different approach is being taken. Within the 120 municipalities currently in high flood risk zones, engagement of schools is currently ongoing, including the development of educational programs for teachers. The aim is to raise student awareness of existing flood risks in their own area, and to help students recognize the value of the CO (and the mobile technology) in protecting their families, e.g., using the app to send important information about flooding, which then contributes to everyone's safety. This component of the CO involves 348 primary schools and 340 middle and secondary schools. The three universities in the area will also be involved through conferences and webinars. Communication through the CO website, via social media campaigns, radio broadcasts and regional newspapers will be used to engage and maintain citizen involvement in the CO. This communication plan, which will continue over the next five years, has the ambitious goal of involving 75,000 people in the CO to download the app and contribute observations.

**3 Methodology**

The methodology consists of three steps: (i) mapping of the flood risk (section 3.1); (ii) quantification of the flood risk reduction (section 3.2); and (iii) calculation of the damage from flooding under three flood scenarios (section 3.3), all of which consider the flood risk with and without the implementation of the CO on flood risk management.

**3.1 Flood risk mapping**

Figure 2 provides an overview of the flood risk methodology employed in the paper, which uses input data outlined in section 3.1.1. As mentioned in the introduction, risk is evaluated from three different components. The first is the flood hazard, which is calculated using a hydrological-hydraulic model to generate flood hazard maps and is described in section 3.1.2. The second is exposure, outlined in section 3.1.3, which is calculated for three macro-categories as set out in the EU 2007/60/CE Flood Directive (EU, 2007): the population affected (art.6-5.a); the types of economic activities affected (art.6-5.b); and the environmental and cultural-archaeological assets affected (art.6.5.c).

[Figure]

**Figure 2: Flowchart outlining the determination of risk in a flood risk assessment context.**

The final component is vulnerability, which has a physical and social dimension. Physical vulnerability is defined as the susceptibility of an exposed element such as people or buildings to flooding (Balbi et al., 2012) and is calculated using the same three macro-categories as that of exposure, i.e., the population affected, the economic activities affected, and the environmental and cultural-archaeological assets affected. Within the people affected category, we also consider social

vulnerability. This refers to the perception or awareness that an adverse event may occur. Some studies have found that if citizens have directly experienced a flood, their perception of flood risk is higher (e.g., Thistlethwaite et al., 2018) although the factors that determine flood risk perception are varied. Moreover, the results from different studies can be ambiguous and/or contradictory (Lechowska, 2018). Social vulnerability can be divided into: (i) adaptive capacity, which is the capacity of an individual, community, society or organization to prepare for and respond to the consequences of a flood event (IPCC, 2012; Torresan et al., 2012); and (ii) coping capacity, which is the ability of an individual, community, society or organization to cope with adverse conditions resulting from a flood event using existing resources (IPCC, 2012; Torresan et al., 2012). The calculation of vulnerability is described in section 3.1.4. Risk is then calculated as the product of hazard, exposure and vulnerability as described in more detail in section 3.1.5, from which the direct tangible costs associated with the flood risk can be calculated (outlined in section 3.2).

**3.1.1 Input data**

There are several data sets used as inputs to the assessment of flood risk as outlined in Table 1. For the evaluation of flood hazard, the water height, flow velocity and flooded areas are provided by AAWA using the methodology described in the Supplementary Materials. Several data sets are used to evaluate flood exposure and vulnerability, but a key data set is Corine Land Cover (CLC) 2006 produced by the European Environment Agency (Steemans, 2008). Other data sets used to determine exposure include layers on population, infrastructure and buildings, areas of cultural heritage, protected areas and sources of pollution, where these data sets were obtained from different Italian ministries to complement the CLC. Data from OpenStreetMap on infrastructure and buildings were also used.

**Table 1: Input data used to calculate risk.**

| Component of risk | Data | Source |
|---|---|---|
| Flood Hazard (low, medium, high hazard scenarios) | Water height (m) | AAWA; see Supplementary Materials for model details |
| | Water speed (m/s) | |
| | Flooded area (km$^2$) | |
| Flood Exposure | Population in residential areas | ISTAT, census data, 2001 |
| | Infrastructure and buildings | Corine Land Cover 2006, OpenStreetMap |
| | Types of agriculture | Corine Land Cover 2006 |
| | Natural and semi-natural systems | Corine Land Cover 2006 |
| | Areas of cultural heritage | Corine Land Cover 2006, MiBACT-Italian Ministry for cultural heritage |
| | Protected areas | Corine Land Cover 2006, MATTM-Italian Ministry for Environment, Veneto Region |
| | Point and widespread sources of pollution (Directives 82/501/EC, 2008/1/EC) | ISTAT, https://prtr.eea.europa.eu |
| Flood Vulnerability (Susceptibility) | Vegetation cover | Corine Land Cover 2006 |
| | Soil type | Corine Land Cover 2006 |

**3.1.2 Flood Hazard Mapping**

According to Article 6 of the 2007/60/CE Flood Directive (EU, 2007), when local authorities implement a Flood Risk Management Plan, three hazard scenarios must be considered:

1. A flood with a low probability, which is 300-year return period in the study area;
2. A flood with a medium probability, which is a 100-year return period in the study area; and
3. A flood with a high probability, which is a 30-year return period in the study area.

These have been calculated using a two-dimensional hydrological and hydraulic model to generate the water levels and the water speeds at a spatial resolution of 10 m (Ferri et al., 2010). Details of the model can be found in the Supplementary Materials. The hazard associated with these scenarios was calculated in relative terms as a value between 0 and 1.

**3.1.3 Flood Exposure Mapping**

The 2006 CLC map provides the underlying spatial information to calculate exposure; the land use classes used here are shown in Table S1 in the Supplementary Materials. As mentioned above, the first macro-category is the people affected by the flooding, or the exposure of the population ($E_P$), which is calculated as follows:

$$E_p = F_d * F_t \qquad\qquad (2)$$

where $F_d$ is a factor characterizing the density of the population in relation to the number of people present (Table 2), which uses gridded population from the census (Table 1), and $F_t$, which is the proportion of time spent in different locations (e.g., houses, schools, etc., using the land use types listed in Table S1) over a 24 hour period (Provincia Autonoma di Trento, 2006). The four classes in Table 2 reflect a very slight decrease in exposure as population density decreases, and were defined by stakeholders in the AAWA based on guidance from ISPRA (2012).

**Table 2: A factor characterizing the density of people ($F_d$) in relation to the number of people present.**

| Number of people | $F_d$ |
|---|---|
| 1 – 50 | 0.90 |
| 51 – 100 | 0.95 |
| 101 – 500 | 0.98 |
| > 500 | 1 |

The exposure or impact on economic activities ($E_E$), which is the second macro-category, is calculated from the restoration costs, and the costs resulting from losses in production and services. The final macro-category, i.e., the exposure of assets in the environmental and cultural heritage category ($E_{ECH}$) is calculated from estimates of potential damage caused by an adverse flood event. These various costs were obtained from the Provincia Autonoma di Treno (2006) and have been calculated for each of the land use classes in Table S1.

The relative values of exposure by land use type for each of the three macro-categories ($E_P$, $E_E$ and $E_{ECH}$) are provided in Table 3. These values have been derived by the Provincia Autonoma di Treno (2006) from decades of experience with understanding exposure related to flood risk. Moreover, they have been tested over time and shown to be valid within AAWA.

**Table 3: The relative values of exposure for people, economic activities, and environmental/cultural assets by land use type.**

| ID | Description | $E_P$ | $E_E$ | $E_{ECH}$ |
|---|---|---|---|---|
| 1 | Residential | 1 | 1 | 1 |
| 2 | Hospital facilities, health care, social assistance | 1 | 1 | 1 |
| 3 | Buildings for public services | 1 | 1 | 1 |
| 4 | Commercial and artisan | 0.5 - 1 | 1 | 0.8 |

| ID | Description | $E_P$ | $E_E$ | $E_{ECH}$ |
|----|-------------|-------|-------|-----------|
| 5 | Industrial | 0.5 - 1 | 1 | 0.3 - 1 |
| 6 | Specialized agricultural | 0.1 - 0.5 | 0.3 - 1 | 0.7 |
| 7 | Woods, meadows, pastures, cemeteries, urban parks | 0.1 - 0.5 | 0.3 | 0.7 |
| 8 | Tourist recreation | 0.4 - 0.5 | 0.5 | 0.1 |
| 9 | Unproductive | 0.1 | 0.1 | 0.3 |
| 10 | Ski areas, Golf course, Horse riding | 0.3 - 0.5 | 0.3 - 1 | 0.3 |
| 11 | Campsites | 1 | 0.5 | 0.1 |
| 12 | Roads of primary importance | 0.5 | 1 | 0.2 |
| 13 | Roads of secondary importance | 0.5 | 0.5 - 1 | 0.1 |
| 14 | Railway area | 0.7 - 1 | 1 | 0.7 |
| 15 | Area for tourist facilities, Zone for collective equipment (supra-municipal, subsoil) | 1 | 0.3 | 0.3 |
| 16 | Technological and service networks | 0.3 - 0.5 | 1 | 0.1 |
| 17 | Facilities supporting communication and transportation networks (airports, ports, service areas, parking lots) | 0.7 - 1 | 1 | 1 |
| 18 | Area for energy production | 0.4 | 1 | 1 |
| 19 | Landfill, Waste treatment plants, Mining areas, Purifiers | 0.3 | 0.5 | 1 |
| 20 | Areas on which plants are installed as per Annex I of Legislative Decree 18 February 2005, n. 59 | 0.9 | 1 | 1 |
| 21 | Areas of historical, cultural and archaeological importance | 0.5 - 1 | 1 | 1 |
| 22 | Environmental goods | 0.5 - 1 | 1 | 1 |
| 23 | Military zone | 0.1 - 1 | 0.1 - 1 | 0.1 - 1 |

**3.1.4 Flood Vulnerability Mapping**

235     Vulnerability is also quantified for each of the three macro-categories (i.e., people, economic activities and environmental/cultural-archaeological assets affected) as outlined below but we additionally differentiate between physical and social vulnerability as described in Section 3.1 and shown in Figure 3.

[Figure]

**Figure 3: Hierarchical combination of indicators and relative weights (in brackets) to calculate the vulnerability of the population.**

**(i) Physical vulnerability of people affected by flooding**

The physical vulnerability associated with people considers the values of flow velocity ($v$) and water height ($h$) that produce "instability" with respect to remaining in an upright position. Many authors have dealt with the instability of people in flowing water (see e.g., Chanson and Brown, 2018), and critical values have been derived from the product of $h$ and $v$. For

245 example, Ramsbottom et al. (2004) and Penning-Rowsell et al. (2005) have proposed a semi-quantitative equation that links a flood hazard index, referred to as the Flood Hazard Rating (FHR), to $h$, $v$ and a factor related to the amount of transported debris, i.e., the Debris Factor ($DF$), as follows:

$$FHR = h * (v + 0.5) + DF \qquad (3)$$

250

The values of the $DF$ related to different ranges of $h$, $v$ and land use are reported in Table 4, which were taken from a study by the UK Department for Environment, Food and Rural Affairs (DEFRA) and the UK Environment Agency (2006) as reported in ISPRA (2012).

255 **Table 4: The Debris Factor ($DF$) for different water depths ($h$), flow velocities ($v$) and land uses.**

| Values of $h$ and $v$ | Grazing/Agricultural land | Forest | Urban |
|---|---|---|---|
| 0 m < $h \leq$ 0.25 m | 0 | 0 | 0 |
| 0.25 m < $h \leq$ 0.75 m | 0 | 0.5 | 1 |
| $h$ > 0.75 OR $v$ > 2 m/s | 0.5 | 1 | 1 |

Using the FHR, the physical vulnerability of the population can be calculated, which is summarized in Figure 4.

[Figure]

**Figure 4: Physical vulnerability values for the population as a function of water height ($h$) and flow velocity ($v$).**

260

**(ii) Social vulnerability of people affected by flooding**

Figure 3 shows the components of social vulnerability, i.e., the adaptive and coping capacity and their respective indicators, along with the weights associated with each of them. The weights and values assigned to each of these indicators have been determined through an expert consultation process carried out by AAWA. Because the different indicators have varying

265 units of measurement, they were first normalized so that they could be combined. Several normalization techniques exist in the literature (Biausque, 2012) but the 'value function' was chosen because it represents a mathematical expression of a human judgement that can be compared in a systematic and explicit way (Beinat, 1997; Mojtahed, et al., 2013). The coping

capacity is comprised of the following demographic and emergency measure indicators, where the corresponding value functions are shown in Figure S1:

- Dependency ratio: the number of citizens aged under 14 and over 65 as a percentage of the total population. A high value of this index implies a reduced ability to adapt to hazardous events.
- Foreigners: the number of foreigners as a percentage of the total population. Due to language barriers and other cultural reasons, areas with a high number of immigrants may not cope as well after a flood event and during emergency situations.
- Number of people involved in emergency management: the number of operators who have been trained to manage an emergency in the region, expressed qualitatively as low, medium and high.
- How frequently civil protection plans are updated: Updating is measured in months to years and indicates how often new hydraulic, urban and technological information is incorporated into civil protection plans.

The adaptive capacity is comprised of three components: the early warning system, equity and risk spread. Early warning systems are evaluated according to three criteria, where the value functions are shown in Figure S2:

- Lead time (or warning time): the number of hours before an event occurs that was predicted by the early warning system.
- Content: the amount of information provided by the early warning system, such as the time and the peak of the flooding at several points across the catchment.
- Reliability: this is linked to the uncertainty of the results from the meteorological forecasts and the hydrological models (Schroter et al., 2008). False alarms can cause inconvenience to people, hinder economic activities, and people may be less likely to take warnings seriously in the future; therefore, they should be minimized.

Finally, equity and spread (shown in Figure S3) are characterized by:

- Gini Index: a measure of the inequality of income distribution within the population. A value of 0 means perfect equality while 1 is complete inequality.
- Number of hospital beds: this is calculated per 1000 people.
- Insurance density: this is the ratio of total insurance premiums (in €) to the total population (Lenzi and Millo, 2005). Values with higher insurance density lead to increased adaptive capacity. However, the insurance density is set to zero because insurance companies in this part of Italy do not currently offer premiums to protect goods against flood damage.
- The frequency at which information on hazard and risk are updated: this is measured in months to years and indicates the ability of institutions to communicate the conditions of danger and risk to the population.
- Involvement of citizens: This is based on the number of students, associations such as farmers and professionals, and citizens that can be reached across large areas through social networks (WP7 WSI Team, 2013) to disseminate information. The values in Figure S3d show the maximum achievable value in the three categories of citizen involvement.

The value for social vulnerability is the sum of the coping and adaptive capacities while the final value for the vulnerability of people is calculated by multiplying the physical and the social vulnerability together.

**(iii) Physical vulnerability of economic activities affected by flooding**

The vulnerability associated with economic activities considers buildings, network infrastructure and agricultural areas. For buildings, the effects from flooding include collapse due to water pressure and/or undermining of the foundations. Moreover, solid materials, such as debris and wood, can be carried by a flood and can cause additional damage to structures. A damage function for brick and masonry buildings has been formulated by Clausen and Clark (1990). Laboratory results have shown that at a water height of 0.5m, the loss to indoor goods is around 50%, which is based on an evaluation made by Risk

Frontiers, an independent research center sponsored by the insurance industry. The structural vulnerability of buildings and losses of associated indoor goods is shown in Figure S4 as a function of the height of the water and flow velocity, which are applied to land use types containing buildings (Table S1). For the camping land use type 11 (Table S1), the values have been modified based on results from Majala (2001).

Vulnerability of the road network is evaluated for land use types 12 and 13 in Table S1, which occurs when it is not possible to use the road due to flooding. This is based on an estimation of the water height and the critical velocity at which vehicles become unstable during a flood, which are derived from direct observation in laboratory experiments and from a report on the literature in this area (Reiter, 2000; Shand et al., 2011); the vulnerability function for the road network is presented in Figure S5. Regarding technological and service networks (land use type 16, Table S1), we assume a vulnerability value equal to 1 if the water height and flow velocity are greater than 2 m and 2 m/s, respectively, otherwise 0.

To assess the vulnerability in agricultural areas (land use types 6 and 7 in Table S1), we assume that the damage is related to harvest loss, and when considering higher flow velocities and water heights, to agricultural buildings and internal goods. Citeau (2003) provides relationships that take water height and flow velocity into account, e.g., the maximum height is 1 m for orchards and 0.5 m for vineyards, and the maximum velocity varies from 0.25 m/s for vegetables and 0.5 m/s for orchards. Concerning cultivation in greenhouses, the maximum damage occurs at a height of 1 m. Finally, high velocities can cause direct damage to cultivated areas but can also lead to soil degradation due to erosion. The vulnerability values for four different types of land as a function of water height and flow velocity are shown in Figure S6. In the case of unproductive land (land use type 9 in Table 1), the vulnerability is assumed to be 0.25, regardless of the $h$ and $v$ values.

**(iv) Physical vulnerability of environmental and cultural heritage assets affected by flooding**

Environmental flood susceptibility is described using contamination/pollution and erosion as indicators. Contamination is caused by industry, animal/human waste and stagnant flooded waters. Erosion can produce disturbance to the land surface and to vegetation but can also damage infrastructure. The approach taken here was to identify protected areas that could potentially be damaged by a flood. For areas susceptible to nutrients, including those identified as vulnerable in Directive 91/676/CEE (Nitrate), and for those defined as susceptible in Directive 91/271/CEE (Urban Waste), we assume a value of 1 for vulnerability (land use type 20 in Table S1). Similarly, in areas identified for habitat and species protection, i.e., sites belonging to the Natura 2000 network established in accordance with the Habitat Directive 92/43/CEE and Birds Directive 79/409/CEE (land use types 8 and 22 in Table S1), the presence of relevant pollution sources was identified (Tables 1 and S1) and assigned a vulnerability of 1. In the absence of pollution sources, the vulnerability was calculated as 0.25 if the flood velocity was less than or equal to 0.5 m/s and the water depth was less than or equal to 1 m; otherwise it was 0.5. Regarding cultural heritage (land use type 21 in Table S1), we assigned a vulnerability of 1 to these areas, taking a conservative approach.

**3.1.5 Mapping flood risk before and after implementation of a CO on flood risk management**

Once the hazard, exposure and vulnerability are mapped, the flood risk, $R$, for the three flood hazard scenarios, $i$, can be mapped as follows:

$$R = \sum_{i=1}^{3} R_i = \frac{w_P (H_i \cdot E_P \cdot V_P) + w_E (H_i \cdot E_E \cdot V_E) + w_{ECH} (H_i \cdot E_{ECH} \cdot V_{ECH})}{w_P + w_E + w_{ECH}} \tag{4}$$

where $H$, $E$ and $V$ are the hazard, exposure and vulnerability associated with the three macro-categories $P$, $E$ and $ECH$ which are the people, economic activities and environmental/cultural-archaeological assets affected, and $w_P$, $w_E$ and $w_{ECH}$ are weights applied to each macro-category, with values of 10, 1 and 1, respectively, which were defined based on stakeholder interviews undertaken by AAWA. To establish the level of risk, four risk classes were defined (Table 5).

**Table 5: Definition of risk classes.**

| Range of R | Description | Risk Category |
|---|---|---|
| 0.1 < R ≤ 0.2 | Low risk where social, economic and environmental damage are negligible or zero | R1 |
| 0.2 < R ≤ 0.5 | Medium risk for which minor damage to buildings, infrastructure and environmental heritage is possible, which does not affect the safety of people, the usability of buildings and economic activities | R2 |
| 0.5 < R ≤ 9 | High risk in terms of safety of people, damage to buildings and infrastructure (and/or unavailability of infrastructure), interruption of socio-economic activities and damage related to the environmental heritage | R3 |
| 0.9 < R ≤ 1 | Very high risk including loss of human life and serious injuries to people, serious damage to buildings, infrastructure and environmental heritage, and total disruption of socio-economic activities | R4 |

These risk classes were then mapped with and without the implementation of the CO for flood risk management. The main change in the calculation of risk is in the social dimension of vulnerability. Before the CO is implemented, this component has a value close to 1. Based on the experience gained in the WeSenseIt project and the goals of the CO, the changes in social vulnerability with the implementation of the CO are shown in Table 6, which decreases the social vulnerability to a value of 0.63. For example, in the coping capacity, the number of people employed in emergency management does not change but as a result of the CO, they will work in a much more efficient manner due to the technology that allows for better emergency management. These tools will also lead to more frequent updating of civil protection plans as well as hazard and risk information updates. In addition, the early warning system will improve in terms of lead time, content and reliability through the greater involvement of trained volunteers and citizens.

**Table 6: Changes in the indicators of social vulnerability with and without implementation of the CO on flood risk management.**

| Social vulnerability | Indicator | Value without CO | Value with CO |
|---|---|---|---|
| Adaptive capacity | Number of people involved in emergency management | Medium | High |
| | Frequency of civil protection plan updating | > 5 years | > 2 years |
| Coping capacity | Lead time of EWS | < 6 hours | 24-72 hours |
| | Content of EWS | Little information | Very detailed information |
| | Reliability of EWS | None | High |
| | Citizen involvement | None | Citizens of large area |
| | Hazard and risk information updating | > 5 years | 1-2 years |

**3.2. Quantifying the risk reduction after implementation of the CO for flood risk management**

To determine the effectiveness (or benefit) of implementing the CO for flood risk management, we consider the changes to the risk class, which has been mapped across the study area before and after implementation of the CO. To aggregate this change in risk after implementation of the CO, the Synthetic Index of Risk Reduction (ISRR) is calculated as:

$$ISRR = \frac{\sum_{ij} k_{ij} \cdot A_j}{\sum_j A_j}, \tag{7}$$

where $A_j$ is the flooded area after the CO is implemented and $k_{ij}$ are the weights from Table 7 for the risk class $i$ before the CO is implemented and $j$ after the implementation. The weights in Table 7 have been determined through expert consultation within the AAWA, supported by the guidelines from ISPRA (2012). If the ISRR is positive, then the overall risk is reduced.

**Table 7: Weights (*k*) for the Synthetic Index of Risk Reduction (ISRR) for changes in risk after implementation of the CO for flood risk management**

| Weights (*k*) | | Risk class before the intervention | | | |
|---|---|---|---|---|---|
| | | R1 | R2 | R3 | R4 |
| Risk class after the intervention | R1 | 0.0 | 10.0 | 20.0 | 30.0 |
| | R2 | -10.0 | 0.0 | 10.0 | 20.0 |
| | R3 | -20.0 | -10.0 | 0.0 | 10.0 |
| | R4 | -30.0 | -20.0 | -10.0 | 0.0 |

**3.3 Financial quantification of the direct damage due to flooding**

To estimate the direct tangible costs due to damage resulting from a flood event, we use the maximum damage functions related to the 44 land use classes in the CLC developed by Huizinga (2007) for the 27 EU member states, which are based on replacement and productivity costs and their gross national products. The replacement costs for damage to buildings, soil and infrastructure assume complete rebuilding or restoration. Productivity costs are calculated based on the costs associated with an interruption in production activities inside the flooded area. The maximum flood damage values for the EU-27 and various EU countries are provided in Table S3. The direct economic impact of the flood is calculated by multiplying the maximum damage values per square meter (in each land use category) by the corresponding areas affected by the floods, i.e., the flood hazard (Section 3.1.2), weighted by the vulnerability value associated with each grid cell. Since the land use map used in this study does not distinguish between industrial and commercial areas, the average of the respective costs per square meter ($475.5 \ €/m^2$) has been applied. Moreover, in discontinuous urban areas, 50% of the value of the damage related to continuous urban areas (i.e., $309 \ €/m^2$) was applied, due to the lower density of buildings in these areas.

The average annual expected damage (*EAD*) can be calculated as follows, where *D* is the damage as a function of the probability of exceeding *P* for a return time *i* (Meyer et al., 2007):

$$EAD = \sum_{i=1}^{k} \frac{D(P_{i-1}) + D(P_i)}{2} \cdot |P_i - P_{i-1}| \tag{5}$$

$$D(P_i) = \sum_i \frac{\sum_j A_{Dj}^i * w_{Dj}}{\sum_j w_{Dj}} \cdot D^i \tag{6}$$

where $w_{Dj}$ is the weight of the damage class, *j* is the damage category and *D* is the damage value shown in Table S3. The EAD is calculated before and after implementing the CO for flood risk management. The monetary benefits are the "avoided" damage costs (to people, buildings, economic activities, protected areas, etc.) if the CO for flood risk management is implemented.

**4 Results**

**4.1 Flood risk estimation without implementation of a flood risk management CO**

**4.1.1 Hazard and risk**

The results of the numerical simulations from the hydraulic model, which were carried out based on the methodology described in the Supplementary Materials, have shown that in some sections of the Bacchiglione River, the flow capacity will exceed that of the river channel. This will result in flooding, which will affect the towns of Torri di Quartesolo, Longare and Montegaldella. There will also be widespread flooding in the cities of Vicenza and Padua, including some industrial areas and others rich in cultural heritage. For a 30-year flood event, the potential flooding could extend to around 40,000 ha,

where 25% of the area contains important urban areas with significant architectural assets. In the case of a 100-year flood event, the areas affected by the flood waters increase further, with more than 50,000 ha flooded, additionally affecting agricultural areas. The results of the simulations are summarized in Tables 8 and 9 in terms of the areas affected in the catchment for different degrees of hazard and risk for 30-, 100- and 300-year flood events.

410

**Table 8: The hazard classes for each return period in terms of area flooded before implementation of the CO.**

| Hazard class | 30 year return period | 100 year return period | 300 year return period |
|---|---|---|---|
| | Area (km²) | | |
| Low | 185.12 | 294.77 | 370.07 |
| Medium | 118.87 | 161.82 | 225.67 |
| High | 54.18 | 74.55 | 104.61 |
| Total | 358.17 | 531.14 | 700.35 |

**Table 9: The risk classes for each return period in terms of area flooded before implementation of the CO.**

| Risk Class | 30 year return period | 100 year return period | 300 year return period |
|---|---|---|---|
| | Area (km²) | | |
| Low (R1) | 160.29 | 254.29 | 318.80 |
| Medium (R2) | 137.26 | 191.89 | 262.03 |
| High (R3) | 56.70 | 79.23 | 110.29 |
| Very High (R4) | 3.92 | 5.73 | 9.23 |
| Total | 358.17 | 531.14 | 700.35 |

415 Figure 5 shows the areas at risk in the territory of Padua for a 100-year flood event. Risk classes R1 (low risk) and R2 (medium risk) have the highest areas for all flood event frequencies. Although areas in R3 (high risk) and R4 (very high risk) may comprise a relatively smaller area when compared to the total area at risk, these also coincide with areas of high concentrations of inhabitants in Vicenza and Padua.

[Figure]

420

**Figure 5: Risk map for the metropolitan area of Padua for a 100-year flood event before implementation of a CO on flood risk management.**

**4.1.2 Expected damage**

The direct damage was calculated for the three flood scenarios: high chance of occurrence (every 30 years), medium (every 425 100 years) or low (every 300 years) and is summarized in Table 10.

**Table 10: Direct damage (without the CO) for three flood scenarios.**

| Scenarios (chance of flood occurrence) | Return period | Damage (million €) |
|---|---|---|
| High | 30 years | 7,053 |
| Medium | 100 years | 8,670 |
| Low | 300 years | 10,853 |

In the event of very frequent flood events, urban areas will be damaged. Furthermore, moving from an event with a high probability of occurrence to one with a medium probability results in a significant increase in the area flooded (i.e., a 48% increase as shown in Table 8) but with a smaller increase in damage (i.e., around 20%). This is explained by the fact that the flooded areas in a 100-year flood event (but not present in a 30-year flood event) are under agricultural use. Similar patterns can be observed when comparing floods with a low and high probability of occurrence. Substituting the values in Table 10 into equation (5), we obtain an expected average annual damage (EAD) of 248.5 million Euros.

**4.2 Flood risk estimation with the implementation of a flood risk management CO**

**4.2.1 Hazard and risk**

As mentioned previously, the hazard remains unchanged (i.e., the results reported in Table 8), but the risk is reduced after implementation of a CO for flood risk management as shown in Table 11 due to the reductions in vulnerability outlined in section 3.1.5. The areas affected in the high (R3) and very high classes (R4) are significantly reduced (R4 to almost zero) compared to the results shown in Table 9 but the areas in the lower risk classes increase. The risk map for a 100-year flood event for the territory of Padua is shown in Figure 6, where the reduction in areas at high and very high risk are clearly visible compared to the situation before implementation of the CO, which is shown in Figure 5.

**Table 11: The risk classes for each return period of flooding in terms of area flooded after implementation of the CO.**

| Risk class | 30 year return period | 100 year return period | 300 year return period |
|---|---|---|---|
| | Area (km$^2$) | | |
| R1 (Low) | 170.96 | 268.68 | 337.78 |
| R2 (Medium) | 168.99 | 235.18 | 322.41 |
| R3 (High) | 18.19 | 27.19 | 40.04 |
| R4 (Very High) | 0.03 | 0.09 | 0.12 |
| Total | 358.17 | 531.14 | 700.35 |

[Figure]

**Figure 6: Risk map for the metropolitan area of Padua for a 100-year flood event after implementation of a CO on flood risk management.**

**4.2.2 Expected damage**

The residual damage was calculated for the three flood scenarios after implementation of the CO on flood risk reduction,
450 which is shown in Table 12. Substituting these residual damage values into equation (5), we obtain an EAD of 111.3 million
Euros, which is a 45% reduction in the damage compared to results before the CO implementation.

**Table 12: Comparison of the direct (without CO) and residual damage (with CO) for three flood scenarios and the cost difference.**

| Scenarios (chance of flood occurrence) | Return period | Direct damage (million €) | Residual damage (million €) | Difference in costs (million €) |
|---|---|---|---|---|
| High | 30 years | 7,053 | 1,573 | -5,480 |
| Medium | 100 years | 8,670 | 5,440 | -3,230 |
| Low | 300 years | 10,853 | 3,420 | -7,433 |

455    The CO for flood risk management has an estimated cost of around 5 million Euros (as detailed in Table S2 in the
Supplementary Materials). Taking the EAD with and without implementation of the CO, the annual benefit in terms of
avoided damage is approximately 137.2 million Euros. Hence the benefits considerably outweigh the costs. The ISRR value
is 2.5, which also indicates a positive reduction in risk. The same methodology was applied to the construction of a retention
basin in the municipalities of Sandrigo and Breganze to improve the hydraulic safety of the Bacchiglione River. Against an
460 expected cost of 70.7 million Euros, which is much higher than the estimated cost for implementing the CO, a significant
reduction in flooded areas would be obtained although high risk would still be evident in the city of Padua. In terms of
damage reduction with the construction of the retention basin, we would obtain an EAD of 140.7 million Euros so the cost to
benefit ratio would be much lower.

**5 Discussion and Conclusions**

465 There is currently a lack of available, appropriate and peer-reviewed evaluation methods and evidence on the added value of
citizen observatories, which is required before they will be more widely adopted by policy makers and practitioners. This
paper has aimed to fill this gap by demonstrating how a traditional cost-benefit analysis can be used to capture the value of a
CO for flood risk management. Although the CO is still being implemented, the proposed methodology was applied using
primary empirical evidence from a CO pilot that was undertaken by the WeSenseIt project in the smaller Bacchiglione
470 catchment to guide changes in the values associated with social vulnerability once the CO is implemented. This allowed the
risk and flood damages to be calculated with and without implementation of the CO, which showed that implementation of a
CO in the Brenta-Bacchiglione is able to reduce the damage, and consequently the risk, for the inhabited areas from an
expected average annual damage (EAD) of €248.5 to €111.3 million euros, i.e., a reduction of 45%. Hence, the
implementation of the CO could significantly reduce the damage and consequently the risk for the inhabited areas of
475 Vicenza, Padua, Torri di Quartesolo, Longare and Montegaldella. The nature of the methodology also means that it can be
applied to other catchments in any part of Italy or other parts of the world that are considering the implementation of a CO
for flood risk management purposes.

The evidence on the costs and benefits of COs for flood risk management generated by this case study can provide
insights that policy makers, authorities and emergency managers can use to make informed choices about the adoption of
480 COs for improving their respective flood risk management practices. In Italy, in general, citizen participation in flood risk
management has been relatively limited. The previous strategy in the Brenta-Bacchiglione catchment has focused on
structural flood mitigation measures, dealing with emergencies and optimizing resources for rapid response. The inclusion of
a CO on flood risk management has been a true innovation in the flood risk management strategies of this region. Future
research can focus on the application of the methodology in other catchments as well as to other fields of disaster

485 management beyond floods. Such applications will serve to generate a broader evidence base for using these types of cost-benefit methodologies.

However, there are also limitations associated with this approach. For example, the analysis presented here did not consider indirect costs, such as those incurred after the event takes place, or in places other than those where the flooding occurred (Merz et al., 2010). In accordance with other authors (e.g., van der Veen et al., 2003), all expenses related to

490 disaster response (e.g., costs for sandbagging, evacuation) are classified as indirect damage. However, the presence of the CO in this catchment does reduce the costs related to emergency services, securing infrastructure, sandbagging and evacuation, all of which can be substantial during a flood event. Therefore, an analysis that takes indirect costs into account could help to further convince policy makers of the feasibility of a CO solution. Similarly, intangible costs were not considered, i.e., the values lost due to an adverse natural event where monetary valuation is difficult because the impacts do

495 not have a corresponding market value (e.g., health effects). Furthermore, the vulnerability assessment of economic activities considers only water depth and flow velocity but not additional factors such as the dynamics of contamination propagation in surface waters during the flood or the duration of the flood event, all of which could be taken into account in estimating the structural damage and monetary losses in the residential, commercial and agricultural sectors.

Another limitation is that this methodology is built on many assumptions, i.e., the numerous coefficients, value

500 functions and weights used to estimate the exposure and vulnerability. Many of these values have been derived through expert consultation and experience and validated internally within AAWA or other Italian agencies. Value functions, in particular, are a way of capturing human judgement in way that can be quantified in situations of high uncertainty. We would argue that the expert consultations have not been undertaken lightly and have often resulted in conservative estimates in the values. Other values have been derived from the literature, all of which will have some uncertainties associated with their

505 derivation. We have not undertaken an uncertainty analysis or a sensitivity analysis. Although we might be able to demonstrate a range of costs and benefits through such an approach, the current benefits heavily outweigh the costs so tweaking individual parameters will be unlikely to have large effects. That said, this cost-benefit analysis is hypothetical because the CO for flood risk management is still being implemented. Hence the real benefits will only be realized once the CO is fully operational. At that stage it will be interesting to validate the assumptions about reductions in social vulnerability

510 and which indicators are the key to reducing flood risk.

Despite these various limitations, this analysis has highlighted the feasibility of a non-structural flood mitigation choice such as a CO for flood risk management compared to the implementation of much more expensive structural measures (e.g., retention areas) in terms of the construction costs and the cost of maintenance over time. By involving citizens in a two-way communication with local authorities through a CO, flood forecasting models can be improved, increased awareness of flood

515 hazard and flood preparedness can be achieved, and community resilience to flood risk can be bolstered.

**Acknowledgements**

The research has been partly funded by the FP7 WeSenseIt project (No. 308429) and the Horizon2020 WeObserve project (No. 776740).

520 **References**

Assumpção, T. H., Popescu, I., Jonoski, A. and Solomatine, D. P.: Citizen observations contributing to flood modelling: Opportunities and challenges, Hydrology and Earth System Sciences, 22(2), 1473–1489, doi:10.5194/hess-22-1473-2018, 2018.

Balbi, S., Giupponi, C., Gain, A., Mojtahed, V., Gallina, V., Torresan, S. and Marcomini, A.: The KULTURisk Framework
525 (KR-FWK): A conceptual framework for comprehensive assessment of risk prevention measures. Deliverable 1.6. KULTURisk Project 265280., 2012.

Beinat, E.: Value Functions for Environmental Management., Kluwer Academic Publishers, The Netherlands., 1997.

Biausque, V.: The Value of Statistical Life: A Meta-Analysis. ENV/EPOC/WPNEP(2010)9/FINAL. Working Party on National Environmental Policies. OECD, Paris., 2012.

530  Blaney, R. J. P., Philippe, A. C. V., Pocock, M. J. O. and Jones, G. D.: Citizen Science and Environmental Monitoring: Towards a Methodology for Evaluating Opportunities, Costs and Benefits. UK Environmental Observation Framework. Available at: http://www.ukeof.org.uk/resources/citizen-science-resources/Costbenefitcitizenscience.pdf, 2016.

Buytaert, W., Zulkafli, Z., Grainger, S., Acosta, L., Alemie, T. C., Bastiaensen, J., De Bièvre, B., Bhusal, J., Clark, J., Dewulf, A., Foggin, M., Hannah, D. M., Hergarten, C., Isaeva, A., Karpouzoglou, T., Pandeya, B., Paudel, D., Sharma,
535  K., Steenhuis, T., Tilahun, S., Van Hecken, G. and Zhumanova, M.: Citizen science in hydrology and water resources: opportunities for knowledge generation, ecosystem service management, and sustainable development, Front. Earth Sci, 2, 26, doi:10.3389/feart.2014.00026, 2014.

Chanson, H. and Brown, R.: Stability of individuals during urban inundations: What should we learn from field observations?, Geosciences, 8(9), 341, doi:10.3390/geosciences8090341, 2018.

540  Citeau, J.-M.: A New Control Concept in the Oise Catchment Area. Definition and Assessment of Flood Compatible Agricultural Activities, FIG working week, Paris, France., 2003.

Clausen, L. and Clark, P. B.: The development of criteria for predicting dambreak flood damages using modelling of historical dam failures, in Proceedings of the International Conference on River Flood Hydraulics. 17-20 September 1990., edited by W. R. White, pp. 369–380, John Wiley & Sons Ltd.  and Hydraulics Research Limited., 1990.

545  CRED: Natural Disasters 2018. Available at: https://reliefweb.int/report/world/natural-disasters-2018, 2019.

Davids, J. C., Devkota, N., Pandey, A., Prajapati, R., Ertis, B. A., Rutten, M. M., Lyon, S. W., Bogaard, T. A. and van de Giesen, N.: Soda Bottle Science—Citizen Science Monsoon Precipitation Monitoring in Nepal, Front. Earth Sci., 7, 46, doi:10.3389/feart.2019.00046, 2019.

DEFRA and UK Environment Agency: Flood and Coastal Defence R&D Program: Flood Risk to People, Phase 2,
550  FD2321/TR2 Guidance Document., 2006.

Etter, S., Strobl, B., Seibert, J. and van Meerveld, H. J. I.: Value of uncertain streamflow observations for hydrological modelling, Hydrol. Earth Syst. Sci., 22(10), 5243–5257, doi:10.5194/hess-22-5243-2018, 2018.

EU: Directive 2007/60/EC of the European Parliament and of the Council of 23 October 2007 on the assessment and management of flood risks. Available at: https://eur-lex.europa.eu/legal-content/EN/TXT/?uri=CELEX:32007L0060,
555  2007.

Ferri, M., Norbiato, D., Monego, M., Galli, A., Gualdi, S., Bucchignani, E. and Baruffi, F.: Impact of climate change on hydrological regimes and water resources in TRUST (Life+ 2007) project, in Proceedings of Hydropredict 2010, Prague, Czech Republic., 2010.

Goldstein, E. A., Lawton, C., Sheehy, E. and Butler, F.: Locating species range frontiers: a cost and efficiency comparison of
560  citizen science and hair-tube survey methods for use in tracking an invasive squirrel, Wildl. Res., 41(1), 64, doi:10.1071/WR13197, 2014.

Hadj-Hammou, J., Loiselle, S., Ophof, D. and Thornhill, I.: Getting the full picture: Assessing the complementarity of citizen science and agency monitoring data, edited by J.-F. Humbert, PLoS ONE, 12(12), e0188507, doi:10.1371/journal.pone.0188507, 2017.

565  Howe, J.: The rise of crowdsourcing, Wired Magazine, 14(6), 1–4, 2006.

Hsu, W.-K., Huang, P.-C., Chang, C.-C., Chen, C.-W., Hung, D.-M. and Chiang, W.-L.: An integrated flood risk assessment model for property insurance industry in Taiwan, Nat Hazards, 58(3), 1295–1309, doi:10.1007/s11069-011-9732-9, 2011.

Huizinga, H. J.: Flood damage functions for EU member states. Technical Report, HKV Consultants. Implemented in the
570  framework of the contract #382441-F1SC awarded by the European Commission - Joint Research Centre, 2007.

IPCC: Managing the Risks of Extreme Events and Disasters to Advance Climate Change Adaptation. Field, C.B., V. Barros, T.F. Stocker, D. Qin, D.J. Dokken, K.L. Ebi, M.D. Mastrandrea, K.J. Mach, G.-K. Plattner, S.K. Allen, M. Tignor, and P.M. Midgley (eds.), Cambridge University Press, Cambridge, UK., 2012.

ISPRA: Proposta metodologica per l'aggiornamento delle mappe di pericolosità e di rischio. Attuazione della Direttiva
575  2007/60/CE/ relative alla valutazione e alla gestione dei rischi da alluvioni (Decreto Legislativo n.49/2010). Istituto Superiore per la Protezione e la Ricerca Ambientale (ISPRA), Roma., 2012.

Lanfranchi, V., Wrigley, S., Ireson, N., Ciravegna, F. and Wehn, U.: Citizens' observatories for situation awareness in flooding., in Proceedings of the 11th International ISCRAM Conference (Information Systems for Crisis and Response Management), 18–21 May 2014. University Park, Pennsylvania, USA, edited by S. R. Hiltz, M. S. Pfaff, L. Plotnick,
580  and P. C. Shih, pp. 145–154., 2014.

Lechowska, E.: What determines flood risk perception? A review of factors of flood risk perception and relations between its basic elements, Nat Hazards, 94(3), 1341–1366, doi:10.1007/s11069-018-3480-z, 2018.

Lenzi, A. and Millo, G.: Regional Heterogeneity and Spatial Spillovers in the Italian Insurance Market. WP1/05. Assicurazaioni Generali, Trieste, Italy, 2005.

585  Levy, J. K. and Hall, J.: Advances in flood risk management under uncertainty, Stoch Environ Res Ris Assess, 19(6), 375–377, doi:10.1007/s00477-005-0005-6, 2005.

Liu, H.-Y., Kobernus, M., Broday, D. and Bartonova, A.: A conceptual approach to a citizens' observatory – supporting community-based environmental governance, Environmental Health, 13(1), doi:10.1186/1476-069X-13-107, 2014.

Maijala, T.: Rescdam: Development of rescue actions based on dam-break flood analysis. Final Report, Grant agreement no.
590  Subv 99/52623 Community Action Programme in the field of civil protection, Helsinki: Finnish Environment Institute., 2001.

Mazumdar, S., Lanfranchi, V., Ireson, N., Wrigley, S., Bagnasco, C., Wehn, U., McDonagh, R., Ferri, M., McCarthy, S., Huwald, H. and Ciravegna, F.: Citizens observatories for effective Earth observations: the WeSenseIt approach, Environmental Scientist, 25(2), 56–61, 2016.

595 Mazzoleni, M., Verlaan, M., Alfonso, L., Monego, M., Norbiato, D., Ferri, M. and Solomatine, D. P.: Can assimilation of crowdsourced data in hydrological modelling improve flood prediction?, Hydrol. Earth Syst. Sci., 21(2), 839–861, doi:10.5194/hess-21-839-2017, 2017.

Mazzoleni, M., Cortes Arevalo, V. J., Wehn, U., Alfonso, L., Norbiato, D., Monego, M., Ferri, M. and Solomatine, D. P.: Exploring the influence of citizen involvement on the assimilation of crowdsourced observations: a modelling study

600 based on the 2013 flood event in the Bacchiglione catchment (Italy), Hydrol. Earth Syst. Sci., 22(1), 391–416, doi:10.5194/hess-22-391-2018, 2018.

Merz, B., Hall, J., Disse, M. and Schumann, A.: Fluvial flood risk management in a changing world, Natural Hazards and Earth System Science, 10(3), 509–527, doi:10.5194/nhess-10-509-2010, 2010.

Meyer, V., Haase, D. and Scheuer, S.: GIS-based multicriteria analysis as decision support in flood risk management, UFZ

605 Discussion Paper, No. 6/2007, Helmholtz-Zentrum für Umweltforschung (UFZ), Leipzig. Available at: https://www.econstor.eu/bitstream/10419/45237/1/548359628.pdf, 2007.

Mojtahed, V., Giupponi, C., Biscaro, C., Gain, A. K. and Balbi, S.: Integrated Assesment of natural Hazards and Climate Change Adaptation: The SERRA Methodology. Università Cà Foscari of Venice, Dept of Economics Research Paper Series No. 07/WP/2013, 2013.

610 Müller, U.: Implementation of the flood risk management directive in selected European countries, Int J Disaster Risk Sci, 4(3), 115–125, doi:10.1007/s13753-013-0013-y, 2013.

National Research Council: Tying Flood Insurance to Flood Risk for Low-Lying Structures in the Floodplain, National Academies Press, Washington, D.C., 2015.

Njue, N., Stenfert Kroese, J., Gräf, J., Jacobs, S. R., Weeser, B., Breuer, L. and Rufino, M. C.: Citizen science in

615 hydrological monitoring and ecosystem services management: State of the art and future prospects, Science of The Total Environment, 693, 133531, doi:10.1016/j.scitotenv.2019.07.337, 2019.

Penning-Rowsell, E., Floyd, P., Ramsbottom, D. and Surendran, S.: Estimating Injury and Loss of Life in Floods: A Deterministic Framework, Natural Hazards, 36(1–2), 43–64, doi:10.1007/s11069-004-4538-7, 2005.

Provincia Autonoma di Trento: Piano Generale di Utilizzazione delle Acque Pubbliche, Parte IV, DPR 15/02/2006., 2006.

620 Ramsbottom, D. S., Wade, S., Bain, V., Hassan, M., Penning-Rowsell, E., Wilson, T., Fernandez, A., House, M. and Floyd, P.: R&D Outputs: Flood Risks to People. Phase 2. FD2321 / IR2. Department for the Environment, Food and Rural Affairs / Environment Agency, London, United Kingdom., 2004.

Reiter, P.: International methods of Risk Analysis, Damage evaluation and social impact studies concerning Dam-Break accidents. EU-Project RESCDAM. Helsinki PR Water Consulting., 2000.

625 Schiermeier, Q.: Increased flood risk linked to global warming, Nature, 470(7334), 316–316, doi:10.1038/470316a, 2011.

Schroter, K., Velasco, C., Nachtnebel, H. P., Kahl, B., Beyene, M., Rubin, C. and Gocht, M.: Effectiveness and Efficiency of Early Warning System for flash floods, CRUE Research Report No1-5, 2008.

Shand, T. D., Cox, R. J., Blacka, M. J. and Smith, G. P.: Australian Rainfall and Runoff Project 10: Appropriate Safety Criteria for Vehicles – Literature Review. Commonwealth of Australia (Geoscience Australia)., Stage 2 Report., 2011.

630 Sheldon, D. and Ashcroft, R.: Citizen Science – where has it come from?, Environmental Scientist, 25(3), 4–11, 2016.

Shirk, J. L., Ballard, H. L., Wilderman, C. C., Phillips, T., Wiggins, A., Jordan, R., McCallie, E., Minarchek, M., Lewenstein, B. V., Krasny, M. E. and Bonney, R.: Public participation in scientific research: A framework for deliberate design, Ecology and Society, 17(2), 29, doi:10.5751/ES-04705-170229, 2012.

Smith, P. J., Brown, S. and Dugar, S.: Community-based early warning systems for flood risk mitigation in Nepal, Nat.

635 Hazards Earth Syst. Sci., 17(3), 423–437, doi:10.5194/nhess-17-423-2017, 2017.

Steemans, C.: Coordination of Information on the Environment (CORINE), in Encyclopedia of Geographic Information Science, edited by K. Kemp, pp. 49–50, Sage Publications Inc., Thousand Oaks, CA., 2008.

Tanoue, M., Hirabayashi, Y. and Ikeuchi, H.: Global-scale river flood vulnerability in the last 50 years, Sci Rep, 6(1), 36021, doi:10.1038/srep36021, 2016.

640 Thistlethwaite, J., Henstra, D., Brown, C. and Scott, D.: How Flood Experience and Risk Perception Influences Protective Actions and Behaviours among Canadian Homeowners, Environmental Management, 61(2), 197–208, doi:10.1007/s00267-017-0969-2, 2018.

Thornhill, I., Loiselle, S., Lind, K. and Ophof, D.: The Citizen Science Opportunity for Researchers and Agencies, BioScience, 66(9), 720–721, doi:10.1093/biosci/biw089, 2016.

645 Torresan, S., Gallina, V., Critto, A., Zabeo, A., Semenzin, E. and Marcomini, A.: D.1.7. Part A. Development of a risk assessment methodology to estimate risk levels, KULTURisk Project 265280., 2012.

van der Veen, A., Steenge, A. E., Bockarjova, M. and Logtmeijer, C.: Structural economic effects of large scale inundation: A simulation of the Krimpen dike breakage, in The Role of Flood Impact Assessment in Flood Defence Policies, edited by A. Vrouwenvelder, pp. 1–50, Delft Cluster, TUD, Delft, The Netherlands., 2003.

650 Wehn, U. and Evers, J.: The social innovation potential of ICT-enabled citizen observatories to increase eParticipation in local flood risk management, Technology in Society, 42, 187–198, doi:10.1016/j.techsoc.2015.05.002, 2015.

Wehn, U., McCarthy, S., Lanfranchi, V. and Tapsell, S. M.: Citizen observatories as facilitators of change in water governance? Experiences from three European cases., Environmental Engineering and Management Journal, 14(9), 2073–2086, 2015.

655  Wehn, U., Gharesifard, M., Anema, K., Alfonso, L. and Mazzoleni, M.: Initial validation and socio-economic impacts report, Ground Truth 2.0 project deliverable D1.11. Delft, the Netherlands, September 2019, 2019.

Wehn, U., Gharesifard, M. and Bilbao, A.: Report on IA methods adapted to CS. MICS project deliverable D2.2. Delft, the Netherlands, 2020a.

Wehn, U., Pfeiffer, E., Gharesifard, M., Alfonso, L. and Anema, K.: Updated validation and socio-economic impacts report.
660  Ground Truth 2.0 project deliverable D1.12. Delft, the Netherlands, February 2020, 2020b.

Werner, M., Reggiani, P., Roo, A. D., Bates, P. and Sprokkereef, E.: Flood Forecasting and Warning at the River Basin and at the European Scale, Natural Hazards, 36(1–2), 25–42, doi:10.1007/s11069-004-4537-8, 2005.

WP7 WSI Team: Case studies methodology and Design. Deliverable 7.10. WeSenseit Project FP7/2007-2013-308429, 2013.